# Directional Convergence, Benign Overfitting of Gradient Descent in leaky ReLU two-layer Neural Networks

**Ichiro Hashimoto**
Department of Statistical Sciences
University of Toronto
Toronto, ON M5G 1Z5
`ichiro.hashimoto@mail.utoronto.ca`

## Abstract

In this paper, we provide sufficient conditions of benign overfitting of fixed width leaky ReLU two-layer neural network classifiers trained on mixture data via gradient descent. Our results are derived by establishing directional convergence of the network parameters and classification error bound of the convergent direction. Our classification error bound also lead to the discovery of a newly identified phase transition. Previously, directional convergence in (leaky) ReLU neural networks was established only for gradient flow. Due to the lack of directional convergence, previous results on benign overfitting were limited to those trained on nearly orthogonal data. All of our results hold on mixture data, which is a broader data setting than the nearly orthogonal data setting in prior work. We demonstrate our findings by showing that benign overfitting occurs with high probability in a much wider range of scenarios than previously known. Our results also allow us to characterize cases when benign overfitting provably fails even if directional convergence occurs. Our work thus provides a more complete picture of benign overfitting in leaky ReLU two-layer neural networks.

## 1 Introduction

The practical success of deep learning has revealed surprising theoretical phenomena. One such phenomenon, which has inspired intense theoretical research, is benign overfitting, where over-parametrized neural network models can achieve arbitrarily small test errors while perfectly interpolating training data—even in the presence of label noise (Zhang et al. (2017); Belkin et al. (2019)). Benign overfitting has attracted broad attention since it seems to conflict with the classical statistical understanding that there should be a trade-off between fitting the training data and generalization on the test data. This led to intense theoretical research which has revealed that benign overfitting also occurs in several classical statistical models, such as linear regression (Bartlett et al. (2020), Muthukumar et al. (2020), Mei and Montanari (2022), Hastie et al. (2022)), ridge regression (Tsigler and Bartlett (2024)), binary linear classification (Chatterji and Long (2021); Wang and Thrampoulidis (2022); Cao et al. (2021); Hashimoto et al. (2025)), and kernel-based estimators (Liang and Rakhlin (2020), Liang et al. (2020)), to name a few. However, our understanding of the phenomenon for neural networks remains limited.

It is widely believed that the implicit bias, which results from implicit regularization of gradient based optimization, is the key to understand generalization performance in the over-parametrized regime. In fact, for binary linear classification, it is known that linear classifiers trained under gradient descent with a loss function with a tight exponential tail, e.g. the exponential loss and logistic loss, converge in direction to the maximum margin classifier (Soudry et al. (2018)). Benign overfitting in binary linear classification has been studied thoroughly by analyzing the convergent direction (Wang and Thrampoulidis (2022); Cao et al. (2021); Hashimoto et al. (2025)).

In this work, we investigate benign overfitting in two-layer leaky ReLU neural networks with fixed width for binary classification whose network parameters are trained on mixture data by gradient

descent with exponential loss. We show that benign overfitting occurs in a much wider scenarios than previously known.

Our results are derived by establishing directional convergence of the network parameters with precise characterization of the convergent direction and by obtaining a classification error bound of that direction. Due to the difficulty in establishing directional convergence of gradient descent for ReLU type networks, prior work for leaky ReLU networks were limited to either gradient flow (Frei et al. (2023a)), smoothed approximation of leaky ReLU (Frei et al. (2022)), or nearly orthogonal data regime (Xu and Gu (2023)), i.e. the predictors $\{\boldsymbol{x}_i\}_{i=1,\ldots,n} \subset \mathbb{R}^p$ satisfy $\|\boldsymbol{x}_i\|^2 \gg \max_{i \neq k} |\langle \boldsymbol{x}_i, \boldsymbol{x}_k \rangle|$.

The precise characterization of convergent direction not only allows us to prove benign overfitting beyond nearly orthogonal data regime but also leads to a novel lower bound of the classification error in the case of Gaussian mixture. The lower bound shows our bound is tight and allows us to characterize cases when benign overfitting provably fails even if directional convergence occurs.

Moreover, sufficient conditions of directional convergence and classification error bound are obtained in a deterministic manner, which allows us to show that benign overfitting occurs with high probability on polynomially tailed mixture models while the prior results are limited to sub-Gaussian mixture.

## 2 RELATED WORK

**Benign overfitting in neural networks for binary classification.** As noted earlier, benign overfitting in neural networks remains poorly understood. In fact, very few theoretical results are established beyond the so called neural tangent kernel (NTK) regime. Cao et al. (2022) studied benign overfitting for two-layer convolutional neural networks with polynomial ReLU activation ($\text{ReLU}^q, q > 2$). Their results were strengthened by Kou et al. (2023a) by relaxing the polynomial ReLU assumption. Both Cao et al. (2022) and Kou et al. (2023a) are not applicable to our setting since our work considers fully connected neural networks.

More closely related work is Frei et al. (2022), which studied benign overfitting in smoothed leaky ReLU two-layer neural networks on mixture data. Their work was extended by Xu and Gu (2023) to non-smooth leaky ReLU like activation. While Xu and Gu (2023)'s work is not limited to (leaky) ReLU, their work is limited to the nearly orthogonal data regime and only holds for isotropic mixture with bounded log-Sobolev constant, which is a stronger assumption than sub-Gaussian mixture. Our work establishes benign overfitting beyond the nearly orthogonal data regime and can be applied to a wider class of mixtures than the prior work such as polynomially tailed mixture. Moreover, we also establish a novel lower bound of the classification error in the case of Gaussian mixture, which shows tightness of our result and helps characterize cases when benign overfitting provably fails. Even with gradient flow, benign overfitting in leaky ReLU neural networks is limited to nearly orthogonal data setting and sub-Gaussian mixture model (Frei et al. (2023a)).

Lastly, our result is shown for fixed width two-layer neural networks, which relaxes Xu and Gu (2023) who required $m$ to grow as $n$, and thus is also beyond the NTK regime or the lazy training regime (Jacot et al. (2018); Arora et al. (2019)).

**Implicit bias in neural networks for binary classification.** Here, we focus on works on leaky ReLU homogeneous neural networks which are closely related to ours. We refer readers to Vardi (2022) for a comprehensive survey on implicit bias in neural networks.

Most prior work on the implicit bias of (leaky) ReLU neural networks for binary classification has considered gradient flow. For gradient flow, Lyu and Li (2020) showed that any limit point of the network parameter of homogeneous neural networks trained with exponential type loss is a KKT point of a constrained optimization problem and that margin maximization occurs. Ji and Telgarsky (2020) further established directional convergence of the parameter of homogeneous neural networks trained with the exponential type loss. Taking advantage of the directional convergence of gradient flow, Lyu et al. (2021); Bui Thi Mai and Lampert (2021); Sarussi et al. (2021); Frei et al. (2023b); Min et al. (2024) obtained detailed characterizations of the convergent direction of the parameters of (leaky) ReLU two-layer neural networks under further assumptions on the training data.

In contrast to gradient flow, directional convergence of network parameters for ReLU type neural networks was not proven for gradient descent with exponential type loss, even under strong assumptions.

As a result, only weaker forms of implicit bias have been obtained for gradient descent. Lyu and Li (2020) showed that margin maximization also occurs for the network parameters of a homogeneous network trained by gradient descent, but their assumptions include smoothness of the network, and hence rule out (leaky) ReLU networks. Frei et al. (2023b) showed that the stable rank of network parameters stays bounded by a constant for two-layer neural networks, but their result is limited to the smoothed approximation of leaky ReLU. Kou et al. (2023b) proved similar results for (leaky) ReLU two-layer neural networks assuming that the number of positive and negative neurons are equal. Moreover, Frei et al. (2023b); Kou et al. (2023b) are limited to the rather specialized scenario of nearly orthogonal data regime. Cai et al. (2025) showed directional convergence of gradient descent, but their result is not applicable to ReLU type neural networks since their result requires networks to be twice differentiable with respect to the parameters. Schechtman and Schreuder (2025) extended Lyu and Li (2020) by removing twice differentiability but their result does not guarantee directional convergence and assumes that training is already in the late-stage.

To the best of our knowledge, our work is the first to show directional convergence of gradient descent in ReLU type neural networks with precise characterization of the convergent direction.

We also note that Karhadkar et al. (2024) studied benign overfitting in leaky ReLU networks using hinge loss instead of exponential type loss.

A concise summary of related prior work is given in Table 1.

Table 1: Summary of Existing Studies of leaky ReLU two-layer neural networks

|  | Directional Convergence | | Benign Overfitting | |
| --- | --- | --- | --- | --- |
|  | Nearly Orthogonal Data | Other regime | Nearly Orthogonal Data | Other regime |
| Gradient Flow | Ji and Telgarsky (2020) | | Frei et al. (2023a) | NA |
| Gradient Descent | NA* | | Frei et al. (2022)**, Xu and Gu (2023) | NA |

\* Cai et al. (2025) showed directional convergence assuming twice differentiability
\*\* Frei et al. (2022) studied smoothed approximation of leaky ReLU instead.

## 3 PRELIMINARIES

We consider two-layer neural networks

$$f(\boldsymbol{x}; W) = \sum_{j=1}^{m} a_j \phi(\langle \boldsymbol{x}, \boldsymbol{w}_j \rangle), \tag{1}$$

where $\boldsymbol{x} \in \mathbb{R}^p$, $\boldsymbol{w}_j \in \mathbb{R}^p$, $W = (\boldsymbol{w}_1, \boldsymbol{w}_2, \ldots, \boldsymbol{w}_m) \in \mathbb{R}^{p \times m}$, $a_j = \pm \frac{1}{\sqrt{m}}$ and $\phi$ is $\gamma-$leaky ReLU activation, i.e. $\phi(x) = \max\{x, \gamma x\}$ for $\gamma \in (0, 1)$. Letting $\zeta(x) = 1$ if $x \geq 0$ and $\zeta(x) = \gamma$ if $x < 0$, we may conveniently write $\phi(x) = \zeta(x)x$.

Letting $J = \{1, 2, \ldots, m\}$, $J_+ = \{j \in J : a_j = \frac{1}{\sqrt{m}}\}$, and $J_- = J \setminus J_+$, we can rewrite the network (1) as

$$f(\boldsymbol{x}; W) = \sum_{j \in J_+} \frac{1}{\sqrt{m}} \phi(\langle \boldsymbol{x}, \boldsymbol{w}_j \rangle) - \sum_{j \in J_-} \frac{1}{\sqrt{m}} \phi(\langle \boldsymbol{x}, \boldsymbol{w}_j \rangle).$$

We consider the exponential loss $\ell(u) = \exp(-u)$ and define the empirical loss of the dataset $\{(\boldsymbol{x}_i, y_i)\}_{i \in I}$, where $I := \{1, 2, \ldots, n\}$, by

$$\mathcal{L}(W) = \sum_{i=1}^{n} \ell(y_i f(\boldsymbol{x}_i; W)) = \sum_{i=1}^{n} \exp(-y_i f(\boldsymbol{x}_i; W)).$$

We train the first layer weights $\boldsymbol{w}_j$ by gradient descent initialized at $W^{(0)}$, i.e.

$$W^{(t+1)} = W^{(t)} - \alpha \nabla_W \mathcal{L}(W^{(t)}), \quad t = 0, 1, 2, \ldots, \tag{2}$$

where $\alpha > 0$ is a fixed constant and we assume $\phi'(0)$ is any number in $[\gamma, 1]$. The second layer weights $a_j$ are fixed throughout the training. Fixing the second layer weights $a_j \in \{\pm 1/\sqrt{m}\}$ and only optimizing the first layer weights $w_j$ is standard in the existing literature, see e.g. Arora et al. (2019); Cao et al. (2022); Frei et al. (2022); Xu and Gu (2023); Kou et al. (2023b). We do not consider random initialization of $W^{(0)}$ since only the magnitude of $W^{(0)}$ plays a role in our analysis.

Assume that we observe i.i.d. copies $\{(\boldsymbol{x}_i, y_i)\}_{i=1,\ldots,n}$ from a mixture distribution on $\mathbb{R}^p \times \{\pm 1\}$ which is defined as follows

(M) The observations consist of $n$ i.i.d copies $(\boldsymbol{x}_i, y_i), i = 1, \ldots, n$ of the pair $(\boldsymbol{x}, y)$. Here for a random variable $y \in \{-1, 1\}$ satisfying $P(y = 1) = P(y = -1) = 1/2$, a random vector $\boldsymbol{z} \in \mathbb{R}^p$ independent of $y$, and a deterministic $\boldsymbol{\mu} \in \mathbb{R}^p$, we have

$$\boldsymbol{x} = y\boldsymbol{\mu} + \boldsymbol{z}. \tag{3}$$

The sub-Gaussian norm of a random variable $X$, say $\|X\|_{\psi_2}$, is defined as

$$\|X\|_{\psi_2} := \inf\{t > 0 : \mathbb{E} \exp(X^2/t^2) \leq 2\}.$$

The sub-Gaussian norm of a random vector $\boldsymbol{X} \in \mathbb{R}^p$ is denoted by $\|\boldsymbol{X}\|_{\psi_2}$ and defined as

$$\|\boldsymbol{X}\|_{\psi_2} := \sup_{\boldsymbol{v} \in S^{p-1}} \|\langle \boldsymbol{v}, \boldsymbol{X} \rangle\|_{\psi_2}.$$

As special case of the model (M), we consider sub-Gaussian mixture model (sG) and polynomially tailed mixture model (PM):

(sG) Suppose in model (M) $\boldsymbol{z} = \Sigma^{\frac{1}{2}}\boldsymbol{\xi}$, where $\boldsymbol{\xi} \in \mathbb{R}^p$ has independent entries $\xi_j$ that have mean zero, unit variance, and satisfy $\|\xi_j\|_{\psi_2} \leq L$ for all $j = 1, \ldots, p$ for $L < \infty$,

(PM) Suppose in model (M) $\boldsymbol{z} = \Sigma^{\frac{1}{2}}\boldsymbol{\xi}$, where $\boldsymbol{\xi} \in \mathbb{R}^p$ has independent entries $\xi_j$ that have mean zero, unit variance, and satisfy $\mathbb{E}|\xi_j|^r \leq K$ for all $j = 1, \ldots, p$ for $r \in (2, 4]$ and $K < \infty$.

The mixture model (sG) have been studied for linear classifiers by Chatterji and Long (2021); Wang and Thrampoulidis (2022); Hashimoto et al. (2025) and for two-layer neural networks with $\Sigma = I_p$ by Frei et al. (2022); Xu and Gu (2023). The model (PM) has been studied for linear classifiers by Hashimoto et al. (2025).

We denote by $\|A\|$ and $\|A\|_F$ the spectral norm and the Frobenius norm of a matrix $A$, respectively.

For $W \in \mathbb{R}^{p \times m}$, we say benign overfitting occurs if the network $f(\cdot; W)$ classifies the training data perfectly (interpolation) while arbitrarily small test error is achieved for sufficiently large $n$.

## 4 DIRECTIONAL CONVERGENCE OF GRADIENT DESCENT

We start from our main result on directional convergence of gradient descent in leaky ReLU two-layer neural networks trained on mixture data. Our result provides precise characterization of the convergent direction, which is a key for our analysis on benign overfitting. The key to prove directional convergence is neuron activation. We say that $j$-th neuron $\phi(\langle \cdot, \boldsymbol{w}_j \rangle)$ is activated if $a_j y_i \phi(\langle \boldsymbol{x}_i, \boldsymbol{w}_j \rangle) > 0$ holds for any $i \in I$, which can be equivalently written as $a_j y_i \langle \boldsymbol{x}_i, \boldsymbol{w}_j \rangle > 0$ for any $i \in I$.

In the following, we let $R_{max}(\boldsymbol{z}) := \max_i \|\boldsymbol{z}_i\|$ and $R_{min}(\boldsymbol{z}) := \min_i \|\boldsymbol{z}_i\|$, and define an event that capture deterministic conditions the training data need to satisfy for the directional convergence:

$$E(\theta_1, \theta_2) = \left\{ \max_{i \neq k} |\langle \frac{\boldsymbol{z}_i}{\|\boldsymbol{z}_i\|}, \frac{\boldsymbol{z}_k}{\|\boldsymbol{z}_k\|} \rangle| \leq \theta_1 \quad \text{and} \quad \max_i |\langle \frac{\boldsymbol{z}_i}{\|\boldsymbol{z}_i\|}, \frac{\boldsymbol{\mu}}{\|\boldsymbol{\mu}\|} \rangle| \leq \theta_2 \right\}, \tag{4}$$

for some $\theta_1, \theta_2 \in [0, 1]$ (take $\theta_1 = 0$ if $\boldsymbol{\mu} = 0$ or $R_{max}(\boldsymbol{z}) = 0$ and take $\theta_2 = 0$ if $\boldsymbol{\mu} = 0$ or $R_{max}(\boldsymbol{z}) = 0$). Event $E$ allows us to control $\|\boldsymbol{x}_i\|$ and $\langle \boldsymbol{x}_i, \boldsymbol{x}_k \rangle, i \neq k$ in terms of $\|\boldsymbol{\mu}\|, R_{min}(\boldsymbol{z}), R_{max}(\boldsymbol{z}), \theta_1,$ and $\theta_2$.

In addition, we denote by $\sigma := \max_j \|\boldsymbol{w}_j\|$ the size of initialization. It is also convenient to let $\rho := \sigma \sqrt{m(1 + \theta_2)\{\|\boldsymbol{\mu}\|^2 + R_{max}^2(\boldsymbol{z})\}}$ to keep the assumptions of our main result concise.

We present sufficient conditions of directional convergence for the following two cases:

**Case 1:** This is the case when $\langle y_i \boldsymbol{x}_i, y_k \boldsymbol{x}_k \rangle \geq 0$ is guaranteed for any $i \neq k$. In this case, we make the following assumptions:

**Assumption 4.1** (Positive Correlation)**.**

$$\|\boldsymbol{\mu}\|^2 \geq 2\theta_2 \|\boldsymbol{\mu}\| R_{max}(\boldsymbol{z}) + \theta_1 R_{max}^2(\boldsymbol{z}).$$

**Assumption 4.2** (Small Initialization)**.**

$$\alpha > \frac{\rho \exp(\rho)}{\gamma(1 - \theta_2)\{\|\boldsymbol{\mu}\|^2 + R_{min}^2(\boldsymbol{z})\}}$$

**Assumption 4.3** (Small Step Size)**.**

$$\alpha(n\|\boldsymbol{\mu}\|^2 + R_{max}(\boldsymbol{z})^2) < 1.$$

The intuition behind these assumptions is as follows: It can be shown that Assumption 4.1 ensures $\langle y_i \boldsymbol{x}_i, y_k \boldsymbol{x}_k \rangle \geq 0$ for any $i \neq k$. Assumption 4.2 ensures the step size sufficiently larger than initialization so that neuron activation occurs immediately after one step of (2). Assumption 4.3 ensures the step size is small enough for directional convergence to hold.

**Case 2:** This is when $\langle y_k \boldsymbol{x}_k, y_i \boldsymbol{x}_i \rangle, k \neq i$ could be negative. The assumptions we make are:

**Assumption 4.4** (Near Orthogonality)**.**

$$2\theta_2 \|\boldsymbol{\mu}\| R_{max}(\boldsymbol{z}) + \theta_1 R_{max}^2(\boldsymbol{z}) \leq \frac{\varepsilon_1 \gamma(1 - \theta_2) R_{min}^2(\boldsymbol{z})}{n \exp(2\rho)} \quad \text{for some } \varepsilon_1 \in [0, 1].$$

**Assumption 4.5** (Small Initialization)**.**

$$\alpha \geq \frac{\rho \exp(\rho)}{\varepsilon_2 \gamma(1 - \theta_2) R_{min}^2(\boldsymbol{z})} \quad \text{for some } \varepsilon_2 \in (0, 1].$$

**Assumption 4.6** (Weak Signal)**.**

$$n\|\boldsymbol{\mu}\|^2 \leq \varepsilon_3 R_{min}^2(\boldsymbol{z}) \quad \text{for some } \varepsilon_3 \in [0, 1).$$

**Assumption 4.7** (Small Step Size)**.**

$$6\alpha R_{max}^2(\boldsymbol{z}) \exp(\rho) < 1.$$

Assumption 4.4 and 4.6 allow us to carefully control $\langle y_k \boldsymbol{x}_k, y_i \boldsymbol{x}_i \rangle$. Small $\varepsilon_1, \varepsilon_3$ ensure that the training data are nearly orthogonal. By taking $\boldsymbol{\mu} = 0$, this case includes the nearly orthogonal data regime studied in the prior work (Frei et al. (2023b); Kou et al. (2023b)). The other assumptions are analogous to Case 1.

We are now ready to state our main result on directional convergence, which not only establishes directional convergence but also provides detailed characterization of the convergent direction:

**Theorem 4.8** (Directional Convergence on Mixture Data)**.** *Suppose event $E$ holds under one of the following conditions:*

- *(i) Assumptions 4.1, 4.2 and 4.3 with $\theta_2 < 1$,*

- *(ii) Assumption 4.4, 4.5, 4.6, and 4.7 with $\varepsilon_1 + \varepsilon_2 + \varepsilon_3 < 1$, $C_w \varepsilon_i \leq \frac{1}{2}, i = 1, 3$, and $\theta_2 \leq \frac{1}{2}$, where $C_w := 24 \exp(1) \gamma^{-2} R_{max}^2(\boldsymbol{z}) R_{min}^{-2}(\boldsymbol{z})$.*

*Then, the gradient descent iterate (2) keeps all the neurons activated for $t \geq 1$, satisfies $\mathcal{L}(W^{(t)}) = O(t^{-1})$, and converges in direction.*

*Furthermore, the convergent direction $\{\hat{\boldsymbol{w}}_j\}_{j \in J}$ can also be given by $\hat{\boldsymbol{w}}_j = \boldsymbol{w}_+$ for $j \in J_+$ and $\hat{\boldsymbol{w}}_j = \boldsymbol{w}_-$ for $j \in J_-$, which is the unique solution to the following optimization problem:*

$$Minimize \quad |J_+|\|\boldsymbol{w}_+\|^2 + |J_-|\|\boldsymbol{w}_-\|^2$$

$$s.t. \quad \frac{|J_+|}{\sqrt{m}}\langle \boldsymbol{x}_i, \boldsymbol{w}_+ \rangle - \frac{\gamma|J_-|}{\sqrt{m}}\langle \boldsymbol{x}_i, \boldsymbol{w}_- \rangle \geq 1, \forall i \in I_+, \tag{5}$$

$$\frac{|J_-|}{\sqrt{m}}\langle \boldsymbol{x}_i, \boldsymbol{w}_- \rangle - \frac{\gamma|J_+|}{\sqrt{m}}\langle \boldsymbol{x}_i, \boldsymbol{w}_+ \rangle \geq 1, \forall i \in I_-.$$

*Lastly, the resulting network $f(\cdot; \hat{W})$ has the linear decision boundary defined by $\bar{\boldsymbol{w}} := \frac{|J_+|}{\sqrt{m}}\boldsymbol{w}_+ - \frac{|J_-|}{\sqrt{m}}\boldsymbol{w}_-$, i.e. $\mathrm{sign}(f(\boldsymbol{x} : \hat{W})) = \mathrm{sign}(\langle \boldsymbol{x}, \bar{\boldsymbol{w}} \rangle)$ for any $\boldsymbol{x} \in \mathbb{R}^p$.*

We note that condition (i) goes beyond nearly orthogonal data regime studied in the existing literature since there is no upper bound to the magnitude of the signal $\boldsymbol{\mu}$, while condition (ii) contains the nearly orthogonal data regime studied in Frei et al. (2023b); Kou et al. (2023b). As discussed in Introduction, directional convergence of gradient descent in ReLU type neural networks was not established in the existing literature even under nearly orthogonal data regime. In addition, the characterizations of the convergent direction in Theorem 4.8 were obtained only for gradient flow trained on nearly orthogonal data by Frei et al. (2023b).

We also note that the optimization problem (5) has a unique solution since the objective function is strictly convex.

We show in Section 6 that these deterministic conditions on training data can be achieved with high probability under sufficient over-parametrization not only by (sG) but also by (PM).

The proof of Theorem 4.8 is given in Appendix B.

## 5 CLASSIFICATION ERROR OF THE CONVERGENT DIRECTION

In this section, we discuss the classification performance of the convergent direction $\hat{W}$.

Near orthogonality of $\boldsymbol{z}_i$'s to each other and to $\boldsymbol{\mu}$ is again the key to obtain the closed form expression and to establish an error bound. Letting $Z = (\boldsymbol{z}_1, \ldots, \boldsymbol{z}_n)^\top \in \mathbb{R}^{n \times p}$, $n_+ = |\{i \in I : y_i = 1\}|$, and $n_- = n - n_+$, the near orthogonality condition is characterized by the parameters from the following events which are defined for $R \geq 0, \tilde{\varepsilon}_1 \geq 0, \tilde{\varepsilon}_2 \geq 0$, and $\tilde{\varepsilon}_3 \in [0, 1]$:

$$\tilde{E}_1(\tilde{\varepsilon}_1, R) := \left\{ \|ZZ^\top - RI_n\| \leq \frac{\tilde{\varepsilon}_1}{3}R \right\}, \tag{6}$$

$$\tilde{E}_2(\tilde{\varepsilon}_2, R) := \left\{ \|Z\boldsymbol{\mu}\| \leq \frac{\tilde{\varepsilon}_2}{3}R \right\}, \tag{7}$$

$$\tilde{E}_3(\tilde{\varepsilon}_3) := \left\{ |n_+ - \tfrac{n}{2}| \leq \tilde{\varepsilon}_3 \tfrac{n}{2} \right\}. \tag{8}$$

Event $\tilde{E}_1$ captures near orthogonality of $\boldsymbol{z}_i$'s to each other and uniformity of their norms, and event $\tilde{E}_2$ captures near orthogonality of $\boldsymbol{z}_i$'s to $\boldsymbol{\mu}$. Although we could use the same parameters as in the analysis of directional convergence, these events $\tilde{E}_1, \tilde{E}_2$ makes the analysis of $\hat{W}$ simpler.

With $\tilde{\varepsilon} := \max\{\tilde{\varepsilon}_1, \sqrt{n}\tilde{\varepsilon}_2\}$, $q_+ = |J_+|/m$, $q_- = |J_-|/m$, and $q_\gamma = \min\{q_+ + \gamma^2 q_-, q_- + \gamma^2 q_+\}$, we can now present our main result on the classification performance of $\hat{W}$:

**Theorem 5.1** (Classification Error Bounds). *Suppose event $\bigcap_{i=1,2,3} \tilde{E}_i$ holds with $\tilde{\varepsilon} \leq \frac{q_\gamma}{2}, \tilde{\varepsilon}_3 \leq \frac{1}{2}$, and further assume one of the following condition holds for a constant $C = C(\gamma, q_+)$:*

*(i) $Cn\|\boldsymbol{\mu}\|^2 \leq R$, $C\tilde{\varepsilon} \leq n^{-\frac{1}{2}}$, and $C\tilde{\varepsilon}_2 \leq \frac{\sqrt{n}\|\boldsymbol{\mu}\|^2}{R}$,*

*(ii) $q_\gamma > \gamma$, $n\|\boldsymbol{\mu}\|^2 \geq CR$, $C\tilde{\varepsilon} \leq \frac{R}{n^{\frac{3}{2}}\|\boldsymbol{\mu}\|^2}$.*

*Then, there exists constants $c, N$ which depend on $\gamma, q_+$ such that $n \geq N$ implies*

$$\mathbb{P}_{(\boldsymbol{x},y)}(yf(x; \hat{W}) < 0) \leq \begin{cases} \exp\left(-c\frac{n\|\boldsymbol{\mu}\|^4}{\|\boldsymbol{z}\|_{\psi_2}^2\{n\|\boldsymbol{\mu}\|^2+R\}}\right) & \textit{for model (sG)}, \\ c\|\Sigma\|\left(\frac{1}{\|\boldsymbol{\mu}\|^2} + \frac{R}{n\|\boldsymbol{\mu}\|^4}\right) & \textit{for model (PM)}. \end{cases}$$

*If $\boldsymbol{z} \sim \mathcal{N}(0, \Sigma)$, then we have instead*

$$\kappa \left\{ c^{-1} \beta_{min}^{-\frac{1}{2}} \left( \frac{n\|\boldsymbol{\mu}\|^4}{n\|\boldsymbol{\mu}\|^2 + R} \right)^{\frac{1}{2}} \right\} \leq \mathbb{P}_{(\boldsymbol{x},y)}(yf(\boldsymbol{x}; \hat{W}) < 0) \leq \kappa \left\{ c\beta_{max}^{-\frac{1}{2}} \left( \frac{n\|\boldsymbol{\mu}\|^4}{n\|\boldsymbol{\mu}\|^2 + R} \right)^{\frac{1}{2}} \right\},$$

*where $\kappa(t) = \mathbb{P}(\xi_1 \geq t)$ and $\beta_{min}$ is the smallest eigenvalue of $\Sigma$ and $\beta_{max} = \|\Sigma\|$.*

This is a concise version for the purpose of presentation. A detailed version of the theorem and the proof are given in Theorem C.13, Appendix C.

Theorem 5.1 shows that there is a phase transition between the weak signal regime, i.e. $n\|\boldsymbol{\mu}\|^2 \lesssim R$ and the strong signal regime, i.e. $n\|\boldsymbol{\mu}\|^2 \gtrsim R$. The weak signal regime corresponds to the nearly orthogonal data regime studied in prior work under specific distributional settings (Frei et al. (2022); Xu and Gu (2023)). The lower bound in the case of Gaussian mixture implies that the phase transition is indeed a feature of the model. Previously, this type of phase transition was not shown even with gradient flow. Moreover, it was only known to occur for binary linear classification (Cao et al. (2021); Wang and Thrampoulidis (2022); Hashimoto et al. (2025)).

Moreover, the assumptions of Theorem 5.1 are given in a deterministic manner. We confirm in the next section that these assumptions are satisfied with high probability under model (sG) and (PM), and conjecture that they can also be verified in other settings.

Lastly, we note that the condition $q_\gamma > \gamma$ in (iii) is necessary to ensure the equivalence of $\hat{\boldsymbol{w}}$ and the minimum norm least square estimator. We can even prove that the equivalence fails if $q_\gamma < \gamma$, which can be viewed as another form of the phase transition in the behavior of the convergent direction. Additional details are provided in Appendix C.

## 6 BENIGN OVERFITTING ON MIXTURE MODELS

We here demonstrate that the results of Theorem 4.8 and 5.1 hold with high probability in model (sG) and (PM) under sufficient over-parametrization. For (sG) we make the following assumptions:

**Assumption 6.1.** *One of the following conditions holds for a constant $C$*

*(a)* $n \geq C$, $\alpha \leq \frac{1}{8\mathrm{tr}(\Sigma)}$, $\sigma \leq 0.1\gamma\alpha\sqrt{\frac{\mathrm{tr}(\Sigma)}{m}}$, $\|\boldsymbol{\mu}\|^2 \geq C\|\Sigma^{\frac{1}{2}}\boldsymbol{\mu}\|$,

$$\mathrm{tr}(\Sigma) \geq C \max\left\{ n\|\boldsymbol{\mu}\|^2, n\|\Sigma\|_F, n^{\frac{3}{2}}\|\Sigma\|, n^{\frac{3}{2}}\|\Sigma^{\frac{1}{2}}\boldsymbol{\mu}\|, n\frac{\|\Sigma^{\frac{1}{2}}\boldsymbol{\mu}\|^2}{\|\boldsymbol{\mu}\|^2} \right\},$$

*and further suppose one of the following:*

*(i)* $\|\boldsymbol{\mu}\|^2 \geq C \max\left\{ \sqrt{n}\|\Sigma^{\frac{1}{2}}\boldsymbol{\mu}\|, \sqrt{n}\|\Sigma\|_F, n\|\Sigma\| \right\}$,

*(ii)* $\mathrm{tr}(\Sigma) \geq C \max\left\{ n^{\frac{3}{2}}\|\Sigma^{\frac{1}{2}}\boldsymbol{\mu}\|, n^{\frac{3}{2}}\|\Sigma\|_F, n^2\|\Sigma\| \right\}$

*(b)* $q_\gamma > \gamma$, $n \geq C$, $\alpha \leq \frac{1}{2.4n\|\boldsymbol{\mu}\|^2}$, $\sigma \leq 0.2\gamma\alpha\sqrt{\frac{\|\boldsymbol{\mu}\|^2 + \mathrm{tr}(\Sigma)}{m}}$,

$$\mathrm{tr}(\Sigma) \geq C \max\left\{ n\|\boldsymbol{\mu}\|\|\Sigma\|_F^{\frac{1}{2}}, n^{5/4}\|\boldsymbol{\mu}\|\|\Sigma\|^{\frac{1}{2}}, n^{\frac{5}{4}}\|\boldsymbol{\mu}\|\|\Sigma^{\frac{1}{2}}\boldsymbol{\mu}\|^{\frac{1}{2}}, n\frac{\|\Sigma^{\frac{1}{2}}\boldsymbol{\mu}\|^2}{\|\boldsymbol{\mu}\|^2}, \sqrt{n}\|\Sigma\|_F, n\|\Sigma\| \right\},$$

$$\|\boldsymbol{\mu}\|^2 \geq C \max\left\{ \frac{\mathrm{tr}(\Sigma)}{n}, \sqrt{n}\|\Sigma^{\frac{1}{2}}\boldsymbol{\mu}\|, \sqrt{n}\|\Sigma\|_F, n\|\Sigma\| \right\}.$$

We note that condition (a) above corresponds to the weak signal regime and (b) to the strong signal regime in Theorem 5.1. We are now ready to present our main classification error bound on (sG):

**Theorem 6.2.** *Consider model (sG). Let $\delta \in (0, \frac{1}{4})$. Suppose Assumption 6.1 holds for a sufficiently large constant $C$ which depends only on $\delta, \gamma, q_+$. Then, we have with probably at least $1 - 3\delta$: the gradient descent iterate (2) keeps all the neurons activated for $t \geq 1$, satisfies $\mathcal{L}(W^{(t)}) = O(t^{-1})$, converges in direction, and the convergent direction $\hat{W}$ attains*

$$\mathbb{P}_{(\boldsymbol{x},y)}(yf(x; \hat{W}) < 0) \leq \exp\left( -\frac{c}{L^2} \frac{n\|\boldsymbol{\mu}\|^4}{\|\Sigma\|\{n\|\boldsymbol{\mu}\|^2 + \mathrm{tr}(\Sigma)\}} \right),$$

*where $c$ is a constant which depends only on $\gamma$ and $q_+$.*

*If $\boldsymbol{z} \sim \mathcal{N}(0, \Sigma)$, then we have instead*

$$\kappa \left\{ c^{-1} \beta_{min}^{-\frac{1}{2}} \left( \frac{n\|\boldsymbol{\mu}\|^4}{n\|\boldsymbol{\mu}\|^2 + \operatorname{tr}(\Sigma)} \right)^{\frac{1}{2}} \right\} \le \mathbb{P}_{(\boldsymbol{x},y)}(yf(\boldsymbol{x}; \hat{W}) < 0) \le \kappa \left\{ c\beta_{max}^{-\frac{1}{2}} \left( \frac{n\|\boldsymbol{\mu}\|^4}{n\|\boldsymbol{\mu}\|^2 + \operatorname{tr}(\Sigma)} \right)^{\frac{1}{2}} \right\}.$$

We note that $\hat{W}$ perfectly classifies the training data since the training loss $\mathcal{L}(t)$ is driven to zero by gradient descent. Thus, Theorem 6.2 implies that benign overfitting occurs if $n\|\boldsymbol{\mu}\|^4 \gg \|\Sigma\|\operatorname{tr}(\Sigma)$ under (a) or $\|\boldsymbol{\mu}\|^2 \gg \|\Sigma\|$ under (b) is satisfied. For Gaussian mixture model, we see that benign overfitting provably fails even though directional convergence occurs if $n\|\boldsymbol{\mu}\|^4 \lesssim \beta_{min}\operatorname{tr}(\Sigma)$ under (a) or $\|\boldsymbol{\mu}\|^2 \lesssim \beta_{min}$ under (b).

We note that Theorem 6.2 under Assumption 6.1 (a) (i) and (b) are novel regimes which were not identified in the existing literature (Frei et al. (2022); Xu and Gu (2023)). Only the result under Assumption 6.1 (a) (ii) with $\Sigma = I_p$ was previously studied. A more detailed comparison with the prior work is given in Appendix A.

The Bayes error rate is given by $\kappa(\|\Sigma^{-1/2}\boldsymbol{\mu}\|)$ if $\boldsymbol{z} \sim \mathcal{N}(0, \Sigma)$ and $\Sigma$ is intvertible (Section 1.2.1, Wainwright (2019)). Therefore, our bounds in the case of isotropic Gaussian mixture is tight up to a constant factor in $\kappa$ in the strong signal regime, i.e. the implicit bias of gradient descent leads to the Bayes optimal.

We also note that Karhadkar et al. (2024) studied implicit bias and benign overfitting in leaky ReLU networks in a somewhat different setting. Since their work studied hinge loss instead of exponential type loss and label-flipping noise is added, it is difficult to directly compare their result with ours. Nevertheless, their distributional setting can be viewed as a special case of ours. In fact, their distributional setting (ignoring the label-flipping noise) is essentially a Gaussian mixture with $\|\boldsymbol{\mu}\| \approx 1$ and $\operatorname{tr}(\Sigma) \approx 1$, which falls into the strong signal regime in our work.

Moreover, Theorem 4.8 and 5.1 allow us to extend the above result to a heavier tailed model (PM). This is a significant distributional relaxation from the existing results which assumed $\boldsymbol{z}$ is sub-Gaussian and strongly log-concave (Frei et al. (2022)) or $\boldsymbol{z}$ has a bounded log-Sobolev constant (Xu and Gu (2023)). For brevity, we only present a result for $\Sigma = I_p$:

**Theorem 6.3.** *Consider model (PM) with $\Sigma = I_p$. Then, benign overfitting occurs with the convergent direction $\hat{W}$ with high probability if either one of the following holds:*

*(i)* $\|\boldsymbol{\mu}\| \gtrsim \sqrt{n}$ *and* $\max\{n\|\boldsymbol{\mu}\|^2, n^{\frac{r+4}{2r-4}}, n^{\frac{8}{r}+1}\} \lesssim p \lesssim \min\left\{ \frac{\|\boldsymbol{\mu}\|^4}{n^{\frac{8}{r}}}, \frac{\|\boldsymbol{\mu}\|^r}{n} \right\}$,

*(ii)* $\|\boldsymbol{\mu}\| \gtrsim \sqrt{n}$ *and*

$$\max\{n^{\frac{5}{4}}\|\boldsymbol{\mu}\|^{\frac{3}{2}}, n^{\frac{8}{3r}+1}\|\boldsymbol{\mu}\|^{\frac{4}{3}}, n^{\frac{3r+4}{3r-4}}\|\boldsymbol{\mu}\|^{\frac{4r}{3r-4}}\} \lesssim p \lesssim \min\left\{ n\|\boldsymbol{\mu}\|^2, \frac{\|\boldsymbol{\mu}\|^4}{n^{\frac{8}{r}}}, \frac{\|\boldsymbol{\mu}\|^r}{n} \right\},$$

*(iii)* $\max\{n\|\boldsymbol{\mu}\|^2, n^{\frac{r+4}{2r-4}}, n^{\frac{8}{r}+2}\} \lesssim p \ll n\|\boldsymbol{\mu}\|^4$.

Detailed versions of the theorems and the proofs are given in Appendix D.

# 7 PROOF SKETCH

In this section, we provide the proof sketch of our main theorems. The key to establish the classification error bound in Theorem 5.1 is the precise characterization of the convergent direction in Theorem 4.8. The precise characterization allows us to obtain both upper and lower bounds of the key quantity $\frac{\|\bar{\boldsymbol{w}}\|}{\langle\bar{\boldsymbol{w}}, \boldsymbol{\mu}\rangle}$ in classification error bounds (98), which results in the novel lower bound in the case of Gaussian mixture. We also note that the conditions in Theorem 4.8 and 5.1 are given in a deterministic manner. The separation of deterministic and distributional arguments allows us to substantially relax distributional assumptions made in prior work. In fact, Theorem 6.2 and 6.3 are proved by showing that all the conditions in Theorem 4.8 and 5.1 are achieved with high probability under (sG) and (PM) with sufficient overparametrization.

**Proof sketch of Theorem 4.8** As noted in Section 4, the key to show directional convergence of the gradient descent is activation of neurons.

The next lemma shows that if all the neurons stay activated after some step of gradient descent (2), directional convergence occurs towards the maximum margin vector $\hat{\boldsymbol{w}} := \text{vec}(\hat{W})$ in a transformed sample space, i.e. $\hat{W}$ is the solution to the following optimization problem:

$$\text{Minimize} \quad \|\boldsymbol{w}\|^2 \quad \text{s.t.} \quad \langle y_i \tilde{\boldsymbol{x}}_i, \boldsymbol{w} \rangle \geq 1 \quad \text{for all } i \in I, \text{ where } \boldsymbol{w} = \text{vec}(W), \tag{9}$$

where $\tilde{\boldsymbol{x}}_i^\top = (\tilde{\boldsymbol{x}}_{i,1}^\top, \tilde{\boldsymbol{x}}_{i,2}^\top, \ldots, \tilde{\boldsymbol{x}}_{i,m}^\top) \in \mathbb{R}^{1 \times mp}$ with $\tilde{\boldsymbol{x}}_{i,j} = a_j \zeta(a_j y_i) \boldsymbol{x}_i$. Letting $\sigma_{max}(\tilde{X})$ be the maximum singular value of $\tilde{X} \in \mathbb{R}^{n \times mp}$ whose rows consist of $\tilde{\boldsymbol{x}}_i^\top$'s, we have

**Lemma 7.1.** *Suppose all the neurons are activated for $t \geq T$ and $\alpha \sigma_{max}^2(\tilde{X}) < 2$. Then, the gradient descent iterate (2) satisfies $\mathcal{L}(W^{(t)}) = O(t^{-1})$ and $W^{(t)}$ converges in the direction characterized by Theorem 4.8.*

The proof is done by reducing the argument to that of binary linear classification, i.e. classifiers of the form $\hat{y} = \text{sgn}(\langle \boldsymbol{x}, \boldsymbol{w} \rangle)$. In fact, if all the neurons are activated for all $(\boldsymbol{x}_i, y_i), i = 1, \ldots, n$ at step $t$, then we have

$$y_i f(\boldsymbol{x}_i; W^{(t)}) = \langle y_i \tilde{\boldsymbol{x}}_i, \boldsymbol{w}^{(t)} \rangle,$$

where $\boldsymbol{w}^{(t)} = \text{vec}(W^{(t)})$. Therefore, if all the neurons stay activated after $t \geq T$, gradient descent iterate (2) becomes that of linear classifier in a transformed space, and the directional convergence follows from Theorem 3, Soudry et al. (2018).

To establish sufficient conditions for the neuron activation, observe that, for any $i, j$, we have

$$\begin{aligned}
&a_j \langle y_i \boldsymbol{x}_i, \boldsymbol{w}_j^{(t+1)} - \boldsymbol{w}_j^{(t)} \rangle \\
&= \frac{\alpha}{m} \Big[ \zeta(\langle \boldsymbol{x}_i, \boldsymbol{w}_j^{(t)} \rangle) \|\boldsymbol{x}_i\|^2 \exp(-y_i f(\boldsymbol{x}_i; W^{(t)})) + \\
&\quad + \sum_{k \neq i} \zeta(\langle \boldsymbol{x}_k, \boldsymbol{w}_j^{(t)} \rangle) \langle y_k \boldsymbol{x}_k, y_i \boldsymbol{x}_i \rangle \exp(-y_k f(\boldsymbol{x}_k; W^{(t)})) \Big].
\end{aligned} \tag{10}$$

The only source of negativity in (10) are $\langle y_i \boldsymbol{x}_i, y_k \boldsymbol{x}_k \rangle, i \neq k$. We note that event $E$ implies

$$\left| \langle y_i \boldsymbol{x}_i, y_k \boldsymbol{x}_k \rangle - \|\boldsymbol{\mu}\|^2 \right| \leq 2\theta_2 \|\boldsymbol{\mu}\| R_{max}(\boldsymbol{z}) + \theta_1 R_{max}^2(\boldsymbol{z}), \forall i \neq k. \tag{11}$$

So, by keeping $2\theta_2 \|\boldsymbol{\mu}\| R_{max}(\boldsymbol{z}) + \theta_1 R_{max}^2(\boldsymbol{z})$ small (Assumption 4.1 and 4.4), the update (10) should stay positive. Then, it should suffice to make all the neurons activated at step $t = 1$, which can be done by taking initialization $\sigma$ sufficiently smaller than the step size $\alpha$ (Assumption 4.2 and 4.5).

**Proof sketch of Theorem 5.1.** The detailed characterizations of the convergent direction given by Theorem 4.8 allows us to establish the desired classification error bound. This is done by obtaining a closed form expression of $\hat{\boldsymbol{w}} = \text{vec}(\hat{W})$, which is $\hat{\boldsymbol{w}} = \tilde{X}^\top (\tilde{X}\tilde{X}^\top)^{-1} \boldsymbol{y}$, where $\boldsymbol{y} = (y_1, \ldots, y_n)^\top \in \mathbb{R}^n$. To establish $\hat{\boldsymbol{w}} = \tilde{X}^\top (\tilde{X}\tilde{X}^\top)^{-1} \boldsymbol{y}$, it is necessary to understand the behavior of $\boldsymbol{y}^\top (\tilde{X}\tilde{X}^\top)^{-1}$. Letting $X = (\boldsymbol{x}_1^\top, \boldsymbol{x}_2^\top, \ldots, \boldsymbol{x}_n^\top)^\top \in R^{n \times p}$ and recalling $\tilde{\boldsymbol{x}}_{ij} = a_j \zeta(y_i a_j) \boldsymbol{x}_i$, we can write

$$\tilde{X} = \begin{pmatrix} a_1 B_1 X & a_2 B_2 X & \cdots & a_m B_m X \end{pmatrix}, \tag{12}$$

where $B_j \in \mathbb{R}^{n \times n}$ is the diagonal matrix whose diagonal entry $(B_j)_{kk} = \zeta(a_j y_k)$. Then we have

$$\tilde{X}\tilde{X}^\top = \sum_{j \in J} a_j^2 B_j X X^\top B_j. \tag{13}$$

Notice that $B_j$'s are the same among $j \in J_+$ and $j \in J_-$, respectively. So, we can write $B_+ = B_j, j \in J_+$ and $B_- = B_j, j \in J_-$. With $a_j^2 = \frac{1}{m}$, we have

$$\tilde{X}\tilde{X}^\top = \frac{|J_+|}{m} B_+ X X^\top B_+ + \frac{|J_-|}{m} B_- X X^\top B_-. \tag{14}$$

The analysis of $\boldsymbol{y}(\tilde{X}\tilde{X}^\top)^{-1}$ will be done through $XX^\top = \|\boldsymbol{\mu}\|^2\boldsymbol{y}\boldsymbol{y}^\top + \boldsymbol{y}(Z\boldsymbol{\mu})^\top + Z\boldsymbol{\mu}\boldsymbol{y}^\top + ZZ^\top \approx \|\boldsymbol{\mu}\|^2\boldsymbol{y}\boldsymbol{y}^\top + RI_n$ under sufficient near orthogonality of $\boldsymbol{z}_i$'s to each other and to $\boldsymbol{\mu}$ (events $\tilde{E}_1, \tilde{E}_2$).

The rest of the argument is split into the strong signal regime and the weak signal regime. The latter is easier since we have $XX^\top \approx RI_n$, which further implies $\tilde{X}\tilde{X}^\top \approx R(q_+B_+^2 + q_-B_-^2)$, which is a diagonal matrix. The strong signal case is more difficult to analyze, and relies on a careful characterization of the eigenvectors of $\|\boldsymbol{\mu}\|^2(q_+B_+\boldsymbol{y}\boldsymbol{y}^\top B_+ + q_-B_-\boldsymbol{y}\boldsymbol{y}^\top B_-) + R(q_+B_+^2 + q_-B_-^2)$.

## 8 DISCUSSION

In this paper, we have established sufficient conditions of benign overfitting of gradient descent in leaky ReLU two-layer neural networks. Here we discuss directions for future research other than the obvious direction of extending to deep neural networks:

Not only showing directional convergence in more generality but also obtaining detailed characterization of the convergent direction remains an important open question. Even for two-layer neural networks we studied, our setting is rather restricted considering a more general result obtained by Ji and Telgarsky (2020) for gradient flow.

In this study, we did not introduce noise to the label $y$ as was done in some prior work on binary classification (Chatterji and Long (2021); Wang and Thrampoulidis (2022); Frei et al. (2022); Xu and Gu (2023); Hashimoto et al. (2025)). We conjecture that introducing noise will not change the result for the weak signal regime since in this regime the signal $\boldsymbol{\mu}$ is negligible. However, the situation is likely to change substantially in the strong signal regime with noisy data as was observed in Hashimoto et al. (2025) for binary linear classification.

### ACKNOWLEDGMENTS

We thank Stanislav Volgushev and Piotr Zwiernik for helpful discussions and guidance. The author is further grateful to the anonymous reviewers and the area chair of ICLR for constructive feedback that improved the manuscript. We would also like to thank Xingyu Xu for helpful discussions regarding the work Xu and Gu (2023).

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

## A    COMPARISON WITH THE EXISTING RESULTS

In this section, we provide more detailed comparison of our main results with the existing results (Frei et al. (2022); Xu and Gu (2023)). Since Frei et al. (2022) only considers smoothed approximation of leaky ReLU, our comparison is mostly with Xu and Gu (2023).

Since only isotropic sub-Gaussian mixture was studied in the existing literature, we present here sufficient conditions of benign overfitting derived from Theorem 6.2 in the case of $\Sigma = I_P$ for the purpose of direct comparison:

**Corollary A.1.** *Consider model (sG) with $\Sigma = I_p$. Then, for sufficiently large $n$, benign overfitting occurs with the convergent direction $\hat{W}$ with high probability if either one of the following holds:*

*(i)*  $\|\boldsymbol{\mu}\| \gtrsim \sqrt{n}$ *and* $n\|\boldsymbol{\mu}\|^2 \lesssim p \lesssim \frac{\|\boldsymbol{\mu}\|^4}{n}$,

*(ii)*  $\|\boldsymbol{\mu}\| \gtrsim \sqrt{n}$ *and* $n^{\frac{5}{4}}\|\boldsymbol{\mu}\|^{\frac{3}{2}} \lesssim p \lesssim \min\{n\|\boldsymbol{\mu}\|^2, \frac{\|\boldsymbol{\mu}\|^4}{n}\}$,

*(iii)*  $\max\{n\|\boldsymbol{\mu}\|^2, n^3\} \lesssim p \ll n\|\boldsymbol{\mu}\|^4$.

We first note that conditions (i) and (ii) are newly identified regimes on which benign overfitting occurs, and only condition (iii) is directly comparable to the results studied in the prior work (Frei et al. (2022); Xu and Gu (2023)). As shown in Theorem 6.3, we have further extended the result in the case of (PM).

**Data Regime:** Our work considers both weak and strong signal regimes and identified the surprising phase transition in classification error occurring at $n\|\boldsymbol{\mu}\|^2 \approx \text{tr}(\Sigma)$. Frei et al. (2022); Xu and Gu (2023) only studied nearly orthogonal data regime, which corresponds to the weak signal regime in this paper, and thus did not identify the phase transition. Even in the weak signal regime, condition (i) in Corollary A.1 was not studied.

**Distributional setting:** Our results significantly relax distributional conditions in prior work by extending to anisotropic polynomially tailed mixture (PM). Both Frei et al. (2022); Xu and Gu (2023) considered mixtures even stronger than (sG). Specifically, Frei et al. (2022) assumed $\boldsymbol{z}$ is not only sub-Gaussian but also strongly log-concave. Xu and Gu (2023) assumed $\boldsymbol{z}$ has a bounded log-Sobolev constant, which is also stronger than just assuming $\boldsymbol{z}$ is sub-Gaussian. Moreover, both work only considered isotropic mixture.

Comparing condition (iii) with that of Xu and Gu (2023), we have a better upper bound on $p$ (Xu and Gu (2023) requires $p\log(mp) \lesssim n\|\boldsymbol{\mu}\|^4$). We allow the width $m$ to be arbitrary while Xu and Gu (2023) requires $m \gtrsim \log n$. We note that Xu and Gu (2023)'s result holds for non-smooth leaky ReLU like activation, which is not limited to (leaky) ReLU. We also note that our lower bound on $p$ is slightly stronger than theirs which requires $p \gtrsim \max\{n\|\boldsymbol{\mu}\|^2, n^2 \log n\}$.

## B    PROOF OF DIRECTIONAL CONVERGENCE

### B.1    PROOF OF LEMMA 7.1

**Lemma B.1.** *Consider (M). If all the neurons are activated at step $t$, i.e. $a_j\langle \boldsymbol{w}_j^{(t)}, y_i\boldsymbol{x}_i\rangle > 0$ holds for all $i \in I$ and $j \in J$, then*

$$\boldsymbol{w}_j^{(t+1)} - \boldsymbol{w}_j^{(t)} = \alpha \sum_{i\in I} \exp(-y_i f(\boldsymbol{x}_i; W^{(t)})) a_j \zeta(a_j y_i) y_i \boldsymbol{x}_i.$$

*Proof.* By (2), we have

$$
\begin{aligned}
\boldsymbol{w}_j^{(t+1)} - \boldsymbol{w}_j^{(t)} &= -\alpha \sum_i \exp(-y_i f(\boldsymbol{x}_i; W^{(t)}))(-y_i) \nabla_{\boldsymbol{w}_j} \left\{ \sum_k a_k \phi(\langle \boldsymbol{w}_k, \boldsymbol{x}_i \rangle) \right\}_{W=W^{(t)}} \\
&= \alpha \sum_i \exp(-y_i f(\boldsymbol{x}_i; W^{(t)})) y_i a_j \nabla_{\boldsymbol{w}_j} \phi(\langle \boldsymbol{w}_j, \boldsymbol{x}_i \rangle)_{\boldsymbol{w}_j = \boldsymbol{w}_j^{(t)}} \\
&= \alpha \sum_i \exp(-y_i f(\boldsymbol{x}_i; W^{(t)})) a_j \zeta(a_j y_i) y_i \boldsymbol{x}_i,
\end{aligned}
$$

where the last equality is due to the assumption $a_j y_i \langle \boldsymbol{w}_j^{(t)}, \boldsymbol{x}_i \rangle > 0$ which implies $\operatorname{sgn}(\langle \boldsymbol{w}_j^{(t)}, \boldsymbol{x}_i \rangle) = \operatorname{sgn}(a_j y_i)$. □

*Remark.* Lemma B.1 implies that the gradient descent update is same within $j \in J_+$ and $j \in J_-$, respectively.

**Lemma B.2.** *Suppose $W = (\boldsymbol{w}_1, \ldots, \boldsymbol{w}_m)$ has $\boldsymbol{w}_j = \boldsymbol{w}_+$ if $j \in J_+$ and $\boldsymbol{w}_j = \boldsymbol{w}_-$ if $j \in J_-$. Then, $f(\boldsymbol{x}; W)$ has the linear decision boundary defined by $\bar{\boldsymbol{w}} := \frac{|J_+|}{\sqrt{m}} \boldsymbol{w}_+ - \frac{|J_-|}{\sqrt{m}} \boldsymbol{w}_-$, i.e. $\operatorname{sign}(f(\boldsymbol{x}; W)) = \operatorname{sign}(\langle \boldsymbol{x}, \bar{\boldsymbol{w}} \rangle)$.*

*Proof.* We consider several cases separately:

**Case $\langle \boldsymbol{x}, \boldsymbol{w}_+ \rangle \geq 0$ and $\langle \boldsymbol{x}, \boldsymbol{w}_- \rangle \geq 0$:** we have

$$
f(\boldsymbol{x}; \hat{W}) = \frac{|J_+|}{\sqrt{m}} \langle \boldsymbol{x}, \boldsymbol{w}_+ \rangle - \frac{|J_-|}{\sqrt{m}} \langle \boldsymbol{x}, \boldsymbol{w}_- \rangle = \langle \boldsymbol{x}, \bar{\boldsymbol{w}} \rangle.
$$

**Case $\langle \boldsymbol{x}, \boldsymbol{w}_+ \rangle \geq 0$ and $\langle \boldsymbol{x}, \boldsymbol{w}_- \rangle < 0$:** we have

$$
f(\boldsymbol{x}; \hat{W}) = \frac{|J_+|}{\sqrt{m}} \langle \boldsymbol{x}, \boldsymbol{w}_+ \rangle - \frac{\gamma |J_-|}{\sqrt{m}} \langle \boldsymbol{x}, \boldsymbol{w}_- \rangle \geq \frac{|J_+|}{\sqrt{m}} \langle \boldsymbol{x}, \boldsymbol{w}_+ \rangle - \frac{|J_-|}{\sqrt{m}} \langle \boldsymbol{x}, \boldsymbol{w}_- \rangle = \langle \boldsymbol{x}, \bar{\boldsymbol{w}} \rangle \geq 0.
$$

**Case $\langle \boldsymbol{x}, \boldsymbol{w}_+ \rangle < 0$ and $\langle \boldsymbol{x}, \boldsymbol{w}_- \rangle \geq 0$:** we have

$$
0 > \langle \boldsymbol{x}, \bar{\boldsymbol{w}} \rangle = \frac{|J_+|}{\sqrt{m}} \langle \boldsymbol{x}, \boldsymbol{w}_+ \rangle - \frac{|J_-|}{\sqrt{m}} \langle \boldsymbol{x}, \boldsymbol{w}_- \rangle \geq \frac{\gamma |J_+|}{\sqrt{m}} \langle \boldsymbol{x}, \boldsymbol{w}_+ \rangle - \frac{|J_-|}{\sqrt{m}} \langle \boldsymbol{x}, \boldsymbol{w}_- \rangle = f(\boldsymbol{x}; \hat{W}).
$$

**Case $\langle \boldsymbol{x}, \boldsymbol{w}_+ \rangle < 0$ and $\langle \boldsymbol{x}, \boldsymbol{w}_- \rangle < 0$:** we have

$$
f(\boldsymbol{x}; \hat{W}) = \gamma \left( \frac{|J_+|}{\sqrt{m}} \langle \boldsymbol{x}, \boldsymbol{w}_+ \rangle - \frac{|J_-|}{\sqrt{m}} \langle \boldsymbol{x}, \boldsymbol{w}_- \rangle \right) = \gamma \langle \boldsymbol{x}, \bar{\boldsymbol{w}} \rangle.
$$

Therefore, we have confirmed $\operatorname{sign}(f(\boldsymbol{x} : \hat{W})) = \operatorname{sign}(\langle \boldsymbol{x}, \bar{\boldsymbol{w}} \rangle)$ holds for all the cases above. □

*Proof of Lemma 7.1.* By the assumption that all the neurons stay activated after $t \geq T$, the directional convergence follows from Corollary 8 of Soudry et al. (2018).

For completeness and to show the remaining claims, we provide the complete proof of the directional convergence here.

First note that $a_j \phi(\langle \boldsymbol{x}_i, \boldsymbol{w}_j^{(t)} \rangle) = a_j \zeta(a_j y_i) \langle \boldsymbol{x}_i, \boldsymbol{w}_j^{(t)} \rangle$ holds for all $i \in I, j \in J, t \geq T$ by the assumption that all the neurons are activated after $t \geq T$. Then, we have for $t \geq T$,

$$
\begin{aligned}
y_i f(\boldsymbol{x}_i; W^{(t)}) &= \sum_j a_j y_i \phi(\langle \boldsymbol{x}_i, \boldsymbol{w}_j^{(t)} \rangle) \\
&= \sum_j a_j y_i \zeta(a_j y_i) \langle \boldsymbol{x}_i, \boldsymbol{w}_j^{(t)} \rangle \\
&= \sum_j \langle a_j \zeta(a_j y_i) y_i \boldsymbol{x}_i, \boldsymbol{w}_j^{(t)} \rangle \\
&= \sum_j \langle y_i \tilde{\boldsymbol{x}}_{ij}, \boldsymbol{w}_j^{(t)} \rangle \\
&= \langle y_i \tilde{\boldsymbol{x}}_i, \boldsymbol{w}^{(t)} \rangle,
\end{aligned}
$$

where the fourth equality follows from the definition of $\tilde{\boldsymbol{x}}_i$ and $\boldsymbol{w}^{(t)} = \text{vec}(W^{(t)})$.

With this, the gradient descent updates in Lemma B.1 can be rewritten as

$$
\boldsymbol{w}_j^{(t+1)} - \boldsymbol{w}_j^{(t)} = \alpha \sum_i \exp\left( -\langle y_i \tilde{\boldsymbol{x}}_i, \boldsymbol{w}^{(t)} \rangle \right) a_j \zeta(a_j y_i) y_i \boldsymbol{x}_i.
$$

Therefore, we have

$$
\boldsymbol{w}^{(t+1)} - \boldsymbol{w}^{(t)} = \alpha \sum_{i \in I} \exp(-\langle y_i \tilde{\boldsymbol{x}}_i, \boldsymbol{w}^{(t)} \rangle) y_i \tilde{\boldsymbol{x}}_i, \tag{15}
$$

which can be viewed as the gradient descent iterates of binary linear classifier of the dataset $\{(\tilde{\boldsymbol{x}}_i, y_i)\}_{i \in I} \subset \mathbb{R}^{mp} \times \{\pm 1\}$.

Therefore, gradient descent iterates $\boldsymbol{w}^{(t)}$ converges in direction if the dataset $\{(\tilde{\boldsymbol{x}}_i, y_i)\}_{i \in I}$ is linearly separable and $\alpha$ is sufficiently small due to Theorem 3 of Soudry et al. (2018). We now provide additional details verifying that this result is applicable in our setting.

It is easy to see linear separability of $\{(\tilde{\boldsymbol{x}}_i, y_i)\}_{i \in I}$ by expanding the following inner product

$$
\langle y_i \tilde{\boldsymbol{x}}_i, \boldsymbol{w}^{(t)} \rangle = \sum_{j \in J} \langle y_i a_j \zeta(a_j y_i) \boldsymbol{x}_i, \boldsymbol{w}_j^{(t)} \rangle = \sum_{j \in J} \zeta(a_j y_i) a_j \langle y_i \boldsymbol{x}_i, \boldsymbol{w}_j^{(t)} \rangle. \tag{16}
$$

Now it is clear that the inner product is positive by the assumption that all the neurons are activated at time $t \geq T$.

In addition, $u \mapsto e^{-u}$ is 1-Lipschitz on $(0, \infty)$ and (16) implies that $\langle y_i \tilde{\boldsymbol{x}}_i, \boldsymbol{w}^{(t)} \rangle$ stays in this region. Therefore, Theorem 3 of Soudry et al. (2018) can be used with $\beta = 1$.

Since $\{(\tilde{\boldsymbol{x}}_i, y_i)\}_{i \in I}$ is linearly separable and satisfies $\alpha \sigma_{max}^2(\tilde{X}) < 2$, $\lim_{t \to \infty} \|\boldsymbol{w}^{(t)}\| = \infty$ follows from Lemma 1 of Soudry et al. (2018), $\mathcal{L}(W^{(t)}) = O(t^{-1})$ follows from Theorem 5 of Soudry et al. (2018), and the directional convergence follows from Theorem 3 of Soudry et al. (2018), respectively.

Now we turn to the characterization of the convergent direction. First, we show that not only $\|\boldsymbol{w}^{(t)}\|$ but also $\|\boldsymbol{w}_j^{(t)}\|$ diverges to infinity for all $j \in J$. To see this, note that by Lemma B.1, we have for $t \geq T$

$$
\boldsymbol{w}_j^{(t)} - \boldsymbol{w}_j^{(T)} = \alpha \sum_i \left( \sum_{s=T}^{t-1} \exp(-y_i f(\boldsymbol{x}_i; W^{(s)})) \right) \zeta(a_j y_i) a_j y_i \boldsymbol{x}_i. \tag{17}
$$

Since $\|\boldsymbol{w}^{(t)}\|$ is diverging to infinity, we must have $\|\boldsymbol{w}_j^{(t)}\|$ diverging to infinity for some $j$, and hence $\sum_{s=T}^{t-1} \exp(-y_i f(\boldsymbol{x}_i; W^{(s)})) \to \infty$ as $t \to \infty$ for some $i$. Denote such $i$ as $i_*$.

Again by Lemma B.1, we have for any $j$,

$$
\begin{aligned}
\langle \boldsymbol{w}_j^{(t)} - \boldsymbol{w}_j^{(T)}, \boldsymbol{w}_j^{(T)} \rangle &= \alpha \sum_i \left( \sum_{s=T}^{t-1} \exp(-y_i f(\boldsymbol{x}_i; W^{(s)})) \right) \zeta(a_j y_i) a_j \langle y_i \boldsymbol{x}_i, \boldsymbol{w}_j^{(T)} \rangle \\
&\geq \alpha \sum_{s=T}^{t-1} \exp(-y_{i_*} f(\boldsymbol{x}_{i_*}; W^{(s)})) \zeta(a_j y_{i_*}) a_j \langle y_{i_*} \boldsymbol{x}_{i_*}, \boldsymbol{w}_j^{(T)} \rangle \to \infty,
\end{aligned} \tag{18}
$$

where the inequality is due to the assumption that all the neurons are activated at $s \geq T$ and the divergence is due to $\sum_{s=T}^{t-1} \exp(-y_{i_*} f(\boldsymbol{x}_{i_*}; W^{(s)})) \to \infty$ and the fact that all neurons are activated at time $T$, implying $\langle y_{i_*} \boldsymbol{x}_{i_*}, \boldsymbol{w}_j^{(T)} \rangle \neq 0$. Since this holds for any $j$, we must have $\|\boldsymbol{w}_j\|$ diverging to infinity for all $j \in J$.

By Theorem 3 of Soudry et al. (2018), the convergent direction is equal to that of the solution to the following optimization problem:

$$\text{Minimize} \quad \|\boldsymbol{w}\|^2 \quad \text{s.t.} \quad \langle y_i \tilde{\boldsymbol{x}}_i, \boldsymbol{w} \rangle \geq 1, \forall i \in I, \tag{19}$$

which is equivalently written as

$$\text{Minimize} \quad \sum_{j \in J} \|\boldsymbol{w}_j\|^2 \quad \text{s.t.} \quad \sum_{j \in J} \langle a_j y_i \zeta(a_j y_i) \boldsymbol{x}_i, \boldsymbol{w}_j \rangle \geq 1, \forall i \in I. \tag{20}$$

Recalling that the gradient descent updates are same within $j \in J_+$ and $j \in J_-$, respectively, and that $\|\boldsymbol{w}_j^{(t)}\|$ diverges to infinity for all $j \in J$, the direction that $\boldsymbol{w}_j^{(t)}$ converges to must be the same for all $j \in J_+$ and the same for all $j \in J_-$. Thus the solution to the optimization problem (20) must have $\boldsymbol{w}_j = \boldsymbol{w}_+$ for all $j \in J_+$ and $\boldsymbol{w}_j = \boldsymbol{w}_-$ for all $j \in J_-$. With this, (20) can be further rewritten as the desired optimization problem (5).

Finally, the claim about the linear decision boundary follows from Lemma B.2 $\qquad \square$

Before proceeding to the subsequent subsections where we prove Theorem 4.8, we introduce some ancillary results:

We first note that event $E$ implies (11) and the following inequality:
$$(1 - \theta_2)\{\|\boldsymbol{\mu}\|^2 + R_{min}^2(\boldsymbol{z})\} \leq \|y_i \boldsymbol{x}_i\|^2 \leq (1 + \theta_2)\{\|\boldsymbol{\mu}\|^2 + R_{max}^2(\boldsymbol{z})\}, \forall i. \tag{21}$$

We also note that the definition of $\rho$ implies
$$\max_i |f(\boldsymbol{x}_i; W^{(0)})| \leq \rho. \tag{22}$$

By (11) and (21), we obtain the following lemma, which is going to be useful to establish sufficient conditions for $\alpha \sigma_{max}^2(\tilde{X}) < 2$:

**Lemma B.3.**
$$\sigma_{max}^2(\tilde{X}) \leq (1 + \theta_2)\{\|\boldsymbol{\mu}\|^2 + R_{max}^2(\boldsymbol{z})\} + (n-1)\left[\|\boldsymbol{\mu}\|^2 + \{2\theta_2\|\boldsymbol{\mu}\|R_{max}(\boldsymbol{z}) + \theta_1 R_{max}^2(\boldsymbol{z})\}\right]$$

*Proof.* Note that

$$\begin{aligned}
|\langle \tilde{\boldsymbol{x}}_i, \tilde{\boldsymbol{x}}_k \rangle| &= \left| \sum_j a_j^2 \zeta(a_j y_i) \zeta(a_j y_k) \langle \boldsymbol{x}_i, \boldsymbol{x}_k \rangle \right| \\
&\leq \sum_j \frac{\zeta(a_j y_i) \zeta(a_j y_k)}{m} |\langle \boldsymbol{x}_i, \boldsymbol{x}_k \rangle| \\
&\leq |\langle \boldsymbol{x}_i, \boldsymbol{x}_k \rangle|,
\end{aligned} \tag{23}$$

where the last inequality is due to $\gamma^2 \leq \zeta(a_j y_i) \zeta(a_j y_k) \leq 1$.

With this, (11), and (21), for any $\boldsymbol{v} \in S^{n-1}$, we have
$$\begin{aligned}
\|\tilde{X}^\top \boldsymbol{v}\|^2 &= \sum_i v_i^2 \|\tilde{\boldsymbol{x}}_i\|^2 + \sum_{i,k: i \neq k} v_i v_k \langle \tilde{\boldsymbol{x}}_i, \tilde{\boldsymbol{x}}_k \rangle \\
&\leq (1 + \theta_2)\{\|\boldsymbol{\mu}\|^2 + R_{max}^2(\boldsymbol{z})\} \sum_i v_i^2 \\
&\quad + \left[\|\boldsymbol{\mu}\|^2 + \{2\theta_2\|\boldsymbol{\mu}\|R_{max}(\boldsymbol{z}) + \theta_1 R_{max}^2(\boldsymbol{z})\}\right] \sum_{i,k: i \neq k} |v_i v_k| \\
&\leq (1 + \theta_2)\{\|\boldsymbol{\mu}\|^2 + R_{max}^2(\boldsymbol{z})\} \\
&\quad + (n-1)\left[\|\boldsymbol{\mu}\|^2 + \{2\theta_2\|\boldsymbol{\mu}\|R_{max}(\boldsymbol{z}) + \theta_1 R_{max}^2(\boldsymbol{z})\}\right],
\end{aligned}$$

where in the last inequality we used $\sum_{i,k:i\neq k}|v_iv_k|\leq n-1$, which follows from the Cauchy-Schwartz inequality. $\qquad\square$

## B.2 PROOF OF THEOREM 4.8 UNDER CONDITION (I)

**Lemma B.4.** *Suppose event $E$ holds with $\|\boldsymbol{\mu}\|^2 \geq 2\theta_2\|\boldsymbol{\mu}\|R_{max}(\boldsymbol{z}) + \theta_1 R_{max}^2(\boldsymbol{z})$ and Assumption 4.2. Then, all the neurons stay activated for $t \geq 1$.*

*Proof of Lemma B.4.* Under $\|\boldsymbol{\mu}\|^2 \geq 2\theta_2\|\boldsymbol{\mu}\|R_{max}(\boldsymbol{z}) + \theta_1 R_{max}^2(\boldsymbol{z})$, the gradient descent updates (10) stays positive. Therefore, it is enough to confirm all the neurons are activated at $t = 1$.

Again by (10), (21), (22), $\|\boldsymbol{\mu}\|^2 \geq 2\theta_2\|\boldsymbol{\mu}\|R_{max}(\boldsymbol{z}) + \theta_1 R_{max}^2(\boldsymbol{z})$,

$$
\begin{aligned}
a_j &\langle y_i\boldsymbol{x}_i, \boldsymbol{w}_j^{(1)}\rangle \\
&\geq \frac{1}{m}\Big[\alpha\gamma(1-\theta_2)\{\|\boldsymbol{\mu}\|^2 + R_{min}^2(\boldsymbol{z})\} \\
&\qquad\qquad\qquad \times \exp\{-\sqrt{m}\sigma\max_i\|\boldsymbol{x}_i\|\} - \sqrt{m}\sigma\max_i\|\boldsymbol{x}_i\|\Big] \\
&\geq \frac{1}{m}\Big[\alpha\gamma(1-\theta_2)\{\|\boldsymbol{\mu}\|^2 + R_{min}^2(\boldsymbol{z})\}\exp(-\rho) - \rho\Big] > 0.
\end{aligned}
\tag{24}
$$

$\qquad\square$

*Proof of Theorem 4.8 under condition (i).* By Lemma B.4, all the neurons stay activated after $t \geq 1$.

By Assumption 4.3, it suffices to show $\sigma_{max}^2(\tilde{X}) \leq 2(n\|\boldsymbol{\mu}\|^2 + R_{max}^2(\boldsymbol{z}))$ to ensure $\alpha\sigma_{max}^2(\tilde{X}) < 2$.

By Lemma B.3 and $\|\boldsymbol{\mu}\|^2 \geq 2\theta_2\|\boldsymbol{\mu}\|R_{max}(\boldsymbol{z}) + \theta_1 R_{max}^2(\boldsymbol{z})$ we have

$$
\begin{aligned}
\sigma_{max}^2(\tilde{X}) &\leq (1+\theta_2)(\|\boldsymbol{\mu}\|^2 + R_{max}^2(\boldsymbol{z})) + 2(n-1)\|\boldsymbol{\mu}\|^2 \\
&\leq 2(\|\boldsymbol{\mu}\|^2 + R_{max}^2(\boldsymbol{z})) + 2(n-1)\|\boldsymbol{\mu}\|^2 \\
&\leq 2(n\|\boldsymbol{\mu}\|^2 + R_{max}^2(\boldsymbol{z})).
\end{aligned}
\tag{25}
$$

Therefore, the directional convergence follows by Lemma 7.1. $\qquad\square$

## B.3 PROOF OF THEOREM 4.8 UNDER CONDITION (II)

First, it is useful to recognize the following inequality which follows from (10), (11), and (21):

$$
\begin{aligned}
\frac{\alpha}{m}&\Big[\gamma(1-\theta_2)\{\|\boldsymbol{\mu}\|^2 + R_{min}^2(\boldsymbol{z})\}\exp(-y_if(\boldsymbol{x}_i; W^{(t)})) \\
&\quad + \sum_{k:k\neq i}\zeta(\langle\boldsymbol{x}_k, \boldsymbol{w}_j^{(t)}\rangle)\{\|\boldsymbol{\mu}\|^2 - 2\theta_2\|\boldsymbol{\mu}\|R_{max}(\boldsymbol{z}) - \theta_1 R_{max}^2(\boldsymbol{z})\}\exp(-y_kf(\boldsymbol{x}_k; W^{(t)}))\Big] \\
&\leq a_j\langle y_i\boldsymbol{x}_i, \boldsymbol{w}_j^{(t+1)} - \boldsymbol{w}_j^{(t)}\rangle \\
&\leq \frac{\alpha}{m}\Big[(1+\theta_2)\{\|\boldsymbol{\mu}\|^2 + R_{max}^2(\boldsymbol{z})\}\exp(-y_if(\boldsymbol{x}_i; W^{(t)})) \\
&\quad + \sum_{k:k\neq i}\{\|\boldsymbol{\mu}\|^2 + 2\theta_2\|\boldsymbol{\mu}\|R_{max}(\boldsymbol{z}) + \theta_1 R_{max}^2(\boldsymbol{z})\}\exp(-y_kf(\boldsymbol{x}_k; W^{(t)}))\Big].
\end{aligned}
\tag{26}
$$

We start from establishing neuron activation at $t = 1$.

**Lemma B.5.** *Suppose event $E$, Assumption 4.4 and 4.5 with $\varepsilon_1 + \varepsilon_2 < 1$. Then, we have for any $i, j$,*

$$
a_j\langle y_i\boldsymbol{x}_i, \boldsymbol{w}_j^{(1)}\rangle > 0,
$$

*and hence all the neurons are activated at $t = 1$.*

*Proof.* By (22), (26), and the assumptions, we have

$$a_j \langle y_i \boldsymbol{x}_i, \boldsymbol{w}_j^{(1)} \rangle \geq \frac{1}{m} \Big[ \alpha \Big\{ \gamma (1 - \theta_2) R_{min}^2(\boldsymbol{z}) \exp(-\rho)$$
$$- n \{ 2\theta_2 \|\boldsymbol{\mu}\| R_{max}(\boldsymbol{z}) + \theta_1 R_{max}^2(\boldsymbol{z}) \} \exp(\rho) \Big\} - \rho \Big]$$
$$\geq \frac{1}{m} \left[ (1 - \varepsilon_1) \alpha \gamma (1 - \theta_2) R_{min}^2(\boldsymbol{z}) \exp(-\rho) - \rho \right]$$
$$\geq \frac{1}{m} (1 - \varepsilon_1 - \varepsilon_2) \alpha \gamma (1 - \theta_2) R_{min}^2(\boldsymbol{z}) \exp(-\rho) > 0.$$

Thus, all the neurons are activated at $t = 1$ if $\varepsilon_1 + \varepsilon_2 < 1$. $\qquad \square$

**Lemma B.6.** *Suppose event E, Assumption 4.4, 4.5, 4.6, and 4.7 with $\varepsilon_1 + \varepsilon_2 + \varepsilon_3 < 1$. Then we have*

$$\max_{k \neq i} \frac{\exp(-y_i f(\boldsymbol{x}_i; W^{(1)}))}{\exp(-y_k f(\boldsymbol{x}_k; W^{(1)}))} \leq \exp(1) \quad and \quad \alpha \sigma_{max}^2(\tilde{X}) < 2.$$

*Proof.* By (22), (26), and the assumptions, we have

$$a_j \langle y_i \boldsymbol{x}_i, \boldsymbol{w}_j^{(1)} \rangle \leq \frac{1}{m} \Big[ \alpha \Big\{ (1 + \theta_2) \{ n \|\boldsymbol{\mu}\|^2 + R_{max}^2(\boldsymbol{z}) \} \exp(\rho)$$
$$+ n \{ 2\theta_2 \|\boldsymbol{\mu}\| R_{max}(\boldsymbol{z}) + \theta_1 R_{max}^2(\boldsymbol{z}) \} \exp(\rho) \Big\} + \rho \Big]$$
$$\leq \frac{1}{m} \Big[ \alpha \Big\{ (1 + \theta_2)(1 + \varepsilon_3) R_{max}^2(\boldsymbol{z}) + \varepsilon_1 R_{max}^2(\boldsymbol{z}) \Big\} \exp(\rho) + \rho \Big]$$
$$\leq \frac{\alpha}{m} (1 + \theta_2)(1 + \varepsilon_1 + \varepsilon_2 + \varepsilon_3) R_{max}^2(\boldsymbol{z}) \exp(\rho)$$
$$\leq \frac{4\alpha R_{max}^2(\boldsymbol{z}) \exp(\rho)}{m}$$
$$\leq \frac{1}{m},$$

where the second inequality is due to Assumption 4.4 and 4.6, the third is due to Assumption 4.5, the fourth is due to $\varepsilon_1 + \varepsilon_2 < 1$, and the last inequality is due to Assumption 4.7.

Therefore, with Lemma B.5, we have $0 < y_i f(\boldsymbol{x}_i; W^{(1)}) \leq 1$, and hence

$$\max_{k \neq i} \frac{\exp(-y_i f(\boldsymbol{x}_i; W^{(1)}))}{\exp(-y_k f(\boldsymbol{x}_k; W^{(1)}))} \leq \max_k \exp(y_k f(\boldsymbol{x}_k; W^{(1)})) \leq \exp(1).$$

We now show $\alpha \sigma_{max}^2(\tilde{X}) < 2$. By Lemma B.3, we have

$$\alpha \sigma_{max}^2(\tilde{X}) \leq \alpha \left\{ (1 + \theta_2) \{ n \|\boldsymbol{\mu}\|^2 + R_{max}^2(\boldsymbol{z}) \} + \varepsilon_1 (1 + \theta_2) R_{max}^2(\boldsymbol{z}) \right\}$$
$$\leq \alpha (1 + \theta_2)(1 + \varepsilon_1 + \varepsilon_3) R_{max}^2(\boldsymbol{z})$$
$$\leq 4\alpha R_{max}^2(\boldsymbol{z}) < 2,$$

where the first inequality is due to Assumption 4.4, the second inequality is due to Assumption 4.6, and the last line is due to $\varepsilon_1 + \varepsilon_3 < 1$ and Assumption 4.7.

$\qquad \square$

Finally, we prove that all the neurons stay activated after $t \geq 1$.

**Lemma B.7.** *Suppose event E, Assumption 4.4, 4.5, 4.6, and 4.7 with $\varepsilon_1 + \varepsilon_2 + \varepsilon_3 < 1$, $C_w \varepsilon_i < \frac{1}{2}, i = 1, 3$, and $\theta_2 \leq \frac{1}{2}$. Then we have for any $i, j$*

$$a_j \langle y_i \boldsymbol{x}_i, \boldsymbol{w}_j^{(t)} \rangle > 0$$

*and*

$$\max_{k \neq i} \frac{\exp(-y_i f(\boldsymbol{x}_i; W^{(t)}))}{\exp(-y_k f(\boldsymbol{x}_k; W^{(t)}))} \leq C_w. \tag{27}$$

*Proof.* We prove both claims simultaneously by induction. By Lemma B.5 and B.6, both claims hold at step $t = 1$. Suppose both claims hold for $t \geq 1$.

We first show the first claim also holds for $t + 1$. By (26), Assumption 4.4, and the induction hypothesis, we have

$$
\begin{aligned}
a_j \langle y_i \boldsymbol{x}_i, \boldsymbol{w}_j^{(t+1)} - \boldsymbol{w}_j^{(t)} \rangle \geq & \frac{\alpha}{m} \exp(-y_i f(\boldsymbol{x}_i; W^{(t)})) \\
& \times \left\{ \gamma(1 - \theta_2) R_{min}^2(\boldsymbol{z}) - C_w \varepsilon_1 \gamma(1 - \theta_2) R_{min}^2(\boldsymbol{z}) \right\} \\
= & \frac{1}{m} \gamma (1 - C_w \varepsilon_1)(1 - \theta_2) \alpha R_{min}^2(\boldsymbol{z}) \exp(-y_i f(\boldsymbol{x}_i; W^{(t)})) \\
\geq & \frac{1}{4m} \gamma \alpha R_{min}^2(\boldsymbol{z}) \exp(-y_i f(\boldsymbol{x}_i; W^{(t)})) > 0.
\end{aligned}
\tag{28}
$$

where the last line is due to $C_w \varepsilon_1 \leq \frac{1}{2}$ and $\theta_2 \leq \frac{1}{2}$. Therefore, all the neurons are active at step $t + 1$.

Next, we prove (27). By (26), the assumptions, and the induction hypothesis, we have

$$
\begin{aligned}
& a_j \langle y_i \boldsymbol{x}_i, \boldsymbol{w}_j^{(t+1)} - \boldsymbol{w}_j^{(t)} \rangle \\
\leq & \frac{\alpha}{m} \exp(-y_i f(\boldsymbol{x}_i; W^{(t)})) \\
& \times \left[ (1 + \theta_2)\left(1 + \frac{\varepsilon_3}{n}\right) R_{max}^2(\boldsymbol{z}) + C_w \varepsilon_3 R_{max}^2(\boldsymbol{z}) + C_w \varepsilon_1 (1 + \theta_2) R_{max}^2(\boldsymbol{z}) \right] \\
\leq & \frac{1}{m} \left\{ 1 + \frac{\varepsilon_3}{n} + C_w \varepsilon_3 + C_w \varepsilon_1 \right\} (1 + \theta_2) \alpha R_{max}^2(\boldsymbol{z}) \exp(-y_i f(\boldsymbol{x}_i; W^{(t)})) \\
\leq & \frac{1}{m} 6 \alpha R_{max}^2(\boldsymbol{z}) \exp(-y_i f(\boldsymbol{x}_i; W^{(t)})),
\end{aligned}
\tag{29}
$$

where the first inequality is due to Assumption 4.4 and 4.6, the last is due to the assumptions on $\varepsilon_i$.

Since all the neurons are activated at both $t$ and $t + 1$, we have

$$
a_j y_i \phi(\langle \boldsymbol{x}_i, \boldsymbol{w}_j^{(t+1)} \rangle) - a_j y_i \phi(\langle \boldsymbol{x}_i, \boldsymbol{w}_j^{(t)} \rangle) = \zeta(a_j y_i) a_j \langle y_i \boldsymbol{x}_i, \boldsymbol{w}_j^{(t+1)} - \boldsymbol{w}_j^{(t)} \rangle.
$$

With this, (28) and (29), we have

$$
\begin{aligned}
\frac{\gamma^2}{4} \alpha R_{min}^2(\boldsymbol{z}) \exp(-y_i f(\boldsymbol{x}_i; W^{(t)})) & \\
\leq y_i f(\boldsymbol{x}_i; W^{(t+1)}) & - y_i f(\boldsymbol{x}_i; W^{(t)}) \\
& \leq 6 \alpha R_{max}^2(\boldsymbol{z}) \exp(-y_i f(\boldsymbol{x}_i; W^{(t)})).
\end{aligned}
\tag{30}
$$

Now we fix $i \neq k$ and let $A_t = \frac{\exp(-y_i f(\boldsymbol{x}_i; W^{(t)}))}{\exp(-y_k f(\boldsymbol{x}_k; W^{(t)}))}$. By induction hypothesis, we have $A_t \leq C_w$. By (30), we have

$$
\begin{aligned}
A_{t+1} \leq A_t \exp \Big[ & - \frac{\gamma^2}{4} \alpha R_{min}^2(\boldsymbol{z}) \exp(-y_i f(\boldsymbol{x}_i; W^{(t)})) \\
& + 6 \alpha R_{max}^2(\boldsymbol{z}) \exp(-y_k f(\boldsymbol{x}_k; W^{(t)})) \Big]
\end{aligned}
\tag{31}
$$

$$
= A_t \exp \Big[ - \frac{\gamma^2}{4} \alpha R_{min}^2(\boldsymbol{z}) \exp(-y_k f(\boldsymbol{x}_k; W^{(t)})) \Big\{ A_t - \frac{24 R_{max}^2(\boldsymbol{z})}{\gamma^2 R_{min}^2(\boldsymbol{z})} \Big\} \Big].
\tag{32}
$$

If $A_t \geq \frac{24 R_{max}^2(\boldsymbol{z})}{\gamma^2 R_{min}^2(\boldsymbol{z})}$, then $A_{t+1} \leq A_t$ immediately follows from (32).

If $A_t < \frac{24 R_{max}^2(\boldsymbol{z})}{\gamma^2 R_{min}^2(\boldsymbol{z})}$, then (31) and the fact that all neurons are activated at time $t$, yielding $y_i f(x_i; W^{(t)}) > 0$ for all $i$, implies

$$
\begin{aligned}
A_{t+1} & \leq \frac{24 R_{max}^2(\boldsymbol{z})}{\gamma^2 R_{min}^2(\boldsymbol{z})} \exp \Big[ 6 \alpha R_{max}^2(\boldsymbol{z}) \Big] \\
& \leq \frac{24 R_{max}^2(\boldsymbol{z})}{\gamma^2 R_{min}^2(\boldsymbol{z})} \exp(1) \leq C_w,
\end{aligned}
$$

where the last line holds from Assumption 4.7. This concludes the induction argument proving the desired claims. $\qquad\square$

*Proof of Theorem 4.8 under condition (ii).* The conclusion now follows immediately from Lemma B.6, B.7, and Lemma 7.1. $\qquad\square$

## C   PROOF OF THEOREM 5.1

The proof of Theorem 5.1 is done by first establishing the equivalence of the maximum margin vector and the minimum norm least square estimator,i.e. $\hat{\boldsymbol{w}} = \tilde{X}^\top(\tilde{X}\tilde{X}^\top)^{-1}\boldsymbol{y}$, and carefully analyzing the behavior.

To prove $\hat{\boldsymbol{w}} = \tilde{X}^\top(\tilde{X}\tilde{X}^\top)^{-1}\boldsymbol{y}$, we rely on the "proliferation of support vector" phenomenon due to Lemma 1, Hsu et al. (2021).

**Lemma C.1.** *If $\boldsymbol{y}^\top(\tilde{X}\tilde{X}^\top)^{-1}y_i\boldsymbol{e}_i > 0$ for all $i$, then*

$$\hat{\boldsymbol{w}} = \tilde{X}^\top(\tilde{X}\tilde{X}^\top)^{-1}\boldsymbol{y}.$$

*Proof.* By Lemma 1, Hsu et al. (2021), the assumption implies all $(\tilde{\boldsymbol{x}}_i, y_i), i = 1, 2, \ldots, n$ are support vectors. Under this condition, the optimization problem defining $\hat{\boldsymbol{w}}$ is equivalent to

$$\hat{\boldsymbol{w}} = \arg\min \|\boldsymbol{w}\|^2, \quad \text{subject to } \langle \boldsymbol{w}, y_i\tilde{\boldsymbol{x}}_i \rangle = 1,$$

and is also equivalent to the optimization problem defining the least square estimator. $\qquad\square$

As noted in Section 7, the analysis is done by establishing $\tilde{X}\tilde{X}^\top \approx \|\boldsymbol{\mu}\|^2(q_+B_+\boldsymbol{y}\boldsymbol{y}^\top B_+ + q_-B_-\boldsymbol{y}\boldsymbol{y}^\top B_-) + R(q_+B_+^2 + q_-B_-^2)$.

**Lemma C.2.** *If event $\tilde{E}_1 \cap \tilde{E}_2$ holds, then the following inequality holds*

$$\left\|XX^\top - \left(\|\boldsymbol{\mu}\|^2\boldsymbol{y}\boldsymbol{y}^\top + RI_n\right)\right\| \le \tilde{\varepsilon}R,$$

*where $\tilde{\varepsilon} := \max\{\tilde{\varepsilon}_1, \sqrt{n}\tilde{\varepsilon}_2\}$.*

*Proof.* Noting event $\tilde{E}_2$ implies $\|\boldsymbol{y}(Z\boldsymbol{\mu})^\top\| \le \frac{\tilde{\varepsilon}_2}{3}R$ and $XX^\top = \|\boldsymbol{\mu}\|^2\boldsymbol{y}\boldsymbol{y}^\top + \boldsymbol{y}(Z\boldsymbol{\mu})^\top + (Z\boldsymbol{\mu})\boldsymbol{y}^\top + ZZ^\top$, we have

$$\left\|XX^\top - \left(\|\boldsymbol{\mu}\|^2\boldsymbol{y}\boldsymbol{y}^\top + RI_n\right)\right\| \le \|\boldsymbol{y}(Z\boldsymbol{\mu})^\top\| + \|(Z\boldsymbol{\mu})\boldsymbol{y}^\top\| + \|ZZ^\top - RI_n\|$$
$$\le \tfrac{\tilde{\varepsilon}}{3}R + \tfrac{\tilde{\varepsilon}}{3}R + \tfrac{\tilde{\varepsilon}}{3}R = \tilde{\varepsilon}R.$$

$\qquad\square$

With (14), this establishes $\tilde{X}\tilde{X}^\top \approx \|\boldsymbol{\mu}\|^2(q_+B_+\boldsymbol{y}\boldsymbol{y}^\top B_+ + q_-B_-\boldsymbol{y}\boldsymbol{y}^\top B_-) + R(q_+B_+^2 + q_-B_-^2)$. It is convenient to introduce $B^2 = q_+B_+^2 + q_-B_-^2$ where $B \in \mathbb{R}^{n \times n}$ is a diagonal matrix with

$$B_{ii} = \begin{cases} \sqrt{q_+ + \gamma^2 q_-} & \text{if } i \in I_+ \\ \sqrt{q_- + \gamma^2 q_+} & \text{if } i \in I_-. \end{cases}$$

**Lemma C.3.** *If event $\tilde{E}_1 \cap \tilde{E}_2$ holds, then we have*

$$\left\|B^{-1}\tilde{X}\tilde{X}^\top B^{-1} - A\right\| \le \tilde{\varepsilon}q_\gamma^{-1}R$$

*where*

$$A := \|\boldsymbol{\mu}\|^2 B^{-1}(q_+B_+\boldsymbol{y}\boldsymbol{y}^\top B_+ + q_-B_-\boldsymbol{y}\boldsymbol{y}^\top B_-)B^{-1} + RI_n.$$

To obtain approximation of $(\tilde{X}\tilde{X}^\top)^{-1}$ from Lemma C.3, we need the following ancillary lemma:

**Lemma C.4.** *If $\|U - V\| \le sL$ and $V \succeq LI_n$ for some $L > 0$ and $s \in [0, \frac{1}{2}]$, then*

$$\|U^{-1} - V^{-1}\| \le 2sL^{-1}.$$

*Proof.* Since $V \succeq LI_n$ for $L > 0$, $V$ is invertible and $\|V^{-1}\| \leq L^{-1}$.

Then we have

$$\|I_n - V^{-1}U\| \leq \|V^{-1}\|\|U - V\| \leq s.$$

Letting $S = I_n - V^{-1}U$, we have $\|S\| \leq s \leq \frac{1}{2}$. So, $U$ is invertible and

$$\|(V^{-1}U)^{-1} - I_n\| \leq \|S\| + \|S\|^2 + \cdots \leq \frac{s}{1-s} \leq 2s.$$

Combining with $\|V^{-1}\| \leq L^{-1}$, we conclude

$$\|U^{-1} - V^{-1}\| \leq \|V^{-1}\|\|(V^{-1}U)^{-1} - I_n\| \leq 2sL^{-1}.$$

$\square$

Now we are ready to obtain the following approximation of $(\tilde{X}\tilde{X}^\top)^{-1}$, which our remaining arguments heavily relies on:

**Lemma C.5.** *If event $\tilde{E}_1 \cap \tilde{E}_2$ holds with $\tilde{\varepsilon} \leq \frac{q_\gamma}{2}$, then we have*

$$\|B(\tilde{X}\tilde{X}^\top)^{-1}B - A^{-1}\| \leq \frac{2q_\gamma^{-1}\tilde{\varepsilon}}{R}.$$

*Proof.* Note that the additional assumption implies $\tilde{\varepsilon}q_\gamma^{-1} \leq \frac{1}{2}$. Then, the desired conclusion follows from Lemma C.3 and C.4 by taking $U = B^{-1}\tilde{X}\tilde{X}^\top B^{-1}, V = A, s = \tilde{\varepsilon}q_\gamma^{-1}$, and $L = R$. $\square$

By Lemma C.1 and C.5, the rest of our arguments is reduced to the analysis of $(B^{-1}\boldsymbol{y})^\top A^{-1}$. Here, we present technical results regarding this quantity which are used frequently later.

**Lemma C.6.** *Let $\boldsymbol{y} = \mathbf{1}_+ - \mathbf{1}_-$, where $(\mathbf{1}_+)_i = 1$ if $i \in I_+$ and $(\mathbf{1}_+)_i = 0$ if $i \in I_-$ and $(\mathbf{1}_-)_i = 1$ if $i \in I_-$ and $(\mathbf{1}_-)_i = 0$ if $i \in I_+$. We have the following equalities:*

1. *if $q_+ = 1, q_- = 0$,*

$$(B^{-1}\boldsymbol{y})^\top A^{-1} = R^{-1} \left\{ \left(1 - \frac{n\|\boldsymbol{\mu}\|^2}{n\|\boldsymbol{\mu}\|^2 + R} \frac{n_+ + \gamma^{-1}n_-}{n}\right) \mathbf{1}_+ \right.$$
$$\left. - \left(\gamma^{-1} - \frac{n\|\boldsymbol{\mu}\|^2}{n\|\boldsymbol{\mu}\|^2 + R} \frac{n_+ + \gamma^{-1}n_-}{n}\right) \mathbf{1}_- \right\}^\top, \quad (33)$$

2. *if $q_+ = 0, q_- = 1$,*

$$(B^{-1}\boldsymbol{y})^\top A^{-1} = R^{-1} \left\{ \left(\gamma^{-1} - \frac{n\|\boldsymbol{\mu}\|^2}{n\|\boldsymbol{\mu}\|^2 + R} \frac{\gamma^{-1}n_+ + n_-}{n}\right) \mathbf{1}_+ \right.$$
$$\left. - \left(1 - \frac{n\|\boldsymbol{\mu}\|^2}{n\|\boldsymbol{\mu}\|^2 + R} \frac{\gamma^{-1}n_+ + n_-}{n}\right) \mathbf{1}_- \right\}^\top, \quad (34)$$

3. *if $q_+q_- \neq 0$,*

$$(B^{-1}\boldsymbol{y})^\top A^{-1}$$
$$= \{d(q_+ + \gamma^2 q_-)(q_- + \gamma^2 q_+)\}^{-1}$$
$$\times \left[\sqrt{q_+ + \gamma^2 q_-}\{(q_- + \gamma^2 q_+ - \gamma)n_-\|\boldsymbol{\mu}\|^2 + (q_- + \gamma^2 q_+)R\}\mathbf{1}_+ \right. \quad (35)$$
$$\left. - \sqrt{q_- + \gamma^2 q_+}\{(q_+ + \gamma^2 q_- - \gamma)n_+\|\boldsymbol{\mu}\|^2 + (q_+ + \gamma^2 q_-)R\}\mathbf{1}_-\right]^\top,$$

*where*

$$d(q_+ + \gamma^2 q_-)(q_- + \gamma^2 q_+)$$
$$= (1 - \gamma^2)^2 q_+ q_- n_+ n_- \|\boldsymbol{\mu}\|^4 + (q_+ + \gamma^2 q_-)(q_- + \gamma^2 q_+)\{n\|\boldsymbol{\mu}\|^2 + R\}R. \quad (36)$$

*Proof.* Let $A = \sum_{i=1}^{n} s_i \boldsymbol{u}_i \boldsymbol{u}_i^\top$ be the singular value decomposition of $A$ with $s_1 \geq s_2 \geq \ldots \geq s_n$. We split in cases.

We first consider the case either $q_+$ or $q_- = 0$, i.e. either $|J_+|$ or $|J_-| = 0$. Then we have $B = B_+$ if $q_- = 0$ and $B = B_-$ if $q_+ = 0$, and hence $A = \|\boldsymbol{\mu}\|^2 \boldsymbol{y}\boldsymbol{y}^\top + R I_n$ either way. Then, we have $s_1 = n\|\boldsymbol{\mu}\|^2 + R$, $s_i = R, i \geq 2$, and can take $\boldsymbol{u}_1 = \frac{\boldsymbol{y}}{\sqrt{n}}$. Thus,

$$
\begin{aligned}
A^{-1} &= [n\|\boldsymbol{\mu}\|^2 + R]^{-1} \frac{\boldsymbol{y}}{\sqrt{n}} \frac{\boldsymbol{y}^\top}{\sqrt{n}} + R^{-1} \sum_{i \geq 2} \boldsymbol{u}_i \boldsymbol{u}_i^\top \\
&= R^{-1} \sum_{i=1}^{n} \boldsymbol{u}_i \boldsymbol{u}_i^\top - \frac{n\|\boldsymbol{\mu}\|^2}{[n\|\boldsymbol{\mu}\|^2 + R]R} \frac{\boldsymbol{y}}{\sqrt{n}} \frac{\boldsymbol{y}^\top}{\sqrt{n}} \\
&= R^{-1} \left( I_n - \frac{n\|\boldsymbol{\mu}\|^2}{n\|\boldsymbol{\mu}\|^2 + R} \frac{\boldsymbol{y}}{\sqrt{n}} \frac{\boldsymbol{y}^\top}{\sqrt{n}} \right).
\end{aligned}
\tag{37}
$$

**Case $q_+ = 1, q_- = 0$:** In this case, we have $B = B_+$ as noted earlier. Since $\boldsymbol{y} = \mathbf{1}_+ - \mathbf{1}_-$ and $B_+^{-1}\boldsymbol{y} = \mathbf{1}_+ - \gamma^{-1}\mathbf{1}_-$, we have that

$$
\begin{aligned}
&(B_+^{-1}\boldsymbol{y})^\top A^{-1} \\
=&R^{-1} \left\{ B_+^{-1}\boldsymbol{y} - \frac{n\|\boldsymbol{\mu}\|^2}{n\|\boldsymbol{\mu}\|^2 + R} \frac{\langle B_+^{-1}\boldsymbol{y}, \boldsymbol{y}\rangle}{n} \boldsymbol{y} \right\}^\top \\
=&R^{-1} \left\{ \left( 1 - \frac{n\|\boldsymbol{\mu}\|^2}{n\|\boldsymbol{\mu}\|^2 + R} \frac{n_+ + \gamma^{-1}n_-}{n} \right) \mathbf{1}_+ \right. \\
&\left. - \left( \gamma^{-1} - \frac{n\|\boldsymbol{\mu}\|^2}{n\|\boldsymbol{\mu}\|^2 + R} \frac{n_+ + \gamma^{-1}n_-}{n} \right) \mathbf{1}_- \right\}^\top.
\end{aligned}
$$

**Case $q_+ = 0, q_- = 1$:** In this case, we have $B = B_-$ as noted earlier. Since $\boldsymbol{y} = \mathbf{1}_+ - \mathbf{1}_-$ and $B_-^{-1}\boldsymbol{y} = \gamma^{-1}\mathbf{1}_+ - \mathbf{1}_-$, we have that

$$
\begin{aligned}
&(B_-^{-1}\boldsymbol{y})^\top A^{-1} \\
=&R^{-1} \left\{ B_-^{-1}\boldsymbol{y} - \frac{n\|\boldsymbol{\mu}\|^2}{n\|\boldsymbol{\mu}\|^2 + R} \frac{\langle B_-^{-1}\boldsymbol{y}, \boldsymbol{y}\rangle}{n} \boldsymbol{y} \right\}^\top \\
=&R^{-1} \left\{ \left( \gamma^{-1} - \frac{n\|\boldsymbol{\mu}\|^2}{n\|\boldsymbol{\mu}\|^2 + R} \frac{\gamma^{-1}n_+ + n_-}{n} \right) \mathbf{1}_+ \right. \\
&\left. - \left( 1 - \frac{n\|\boldsymbol{\mu}\|^2}{n\|\boldsymbol{\mu}\|^2 + R} \frac{\gamma^{-1}n_+ + n_-}{n} \right) \mathbf{1}_- \right\}^\top.
\end{aligned}
$$

**Case $q_+ q_- \neq 0$:** In this case, we have the span of $\{B^{-1}B_+\boldsymbol{y}, B^{-1}B_-\boldsymbol{y}\}$ equal to the span of $\{\boldsymbol{u}_1, \boldsymbol{u}_2\}$, and $s_i = R$ for $i \geq 3$.

To determine $\sum_{i=1,2} s_i^{-1} \boldsymbol{u}_i \boldsymbol{u}_i^\top$, it is convenient to use isomorphism between the span of $\{B^{-1}B_+\boldsymbol{y}, B^{-1}B_-\boldsymbol{y}\}$ and $\mathbb{R}^2$ by $\mathbf{1}_+ \leftrightarrow \sqrt{n_+}\boldsymbol{e}_1$ and $\mathbf{1}_- \leftrightarrow \sqrt{n_-}\boldsymbol{e}_2$. Under this isomorphism, we have

$$
\begin{aligned}
B^{-1}B_+\boldsymbol{y} &= \frac{1}{\sqrt{q_+ + \gamma^2 q_-}}\mathbf{1}_+ - \frac{\gamma}{\sqrt{q_- + \gamma^2 q_+}}\mathbf{1}_- \leftrightarrow \frac{\sqrt{n_+}}{\sqrt{q_+ + \gamma^2 q_-}}\boldsymbol{e}_1 - \frac{\gamma\sqrt{n_-}}{\sqrt{q_- + \gamma^2 q_+}}\boldsymbol{e}_2, \\
B^{-1}B_-\boldsymbol{y} &= \frac{\gamma}{\sqrt{q_+ + \gamma^2 q_-}}\mathbf{1}_+ - \frac{1}{\sqrt{q_- + \gamma^2 q_+}}\mathbf{1}_- \leftrightarrow \frac{\gamma\sqrt{n_+}}{\sqrt{q_+ + \gamma^2 q_-}}\boldsymbol{e}_1 - \frac{\sqrt{n_-}}{\sqrt{q_- + \gamma^2 q_+}}\boldsymbol{e}_2.
\end{aligned}
\tag{38}
$$

Thus, $\sum_{i=1,2} s_i \boldsymbol{u}_i \boldsymbol{u}_i^\top$ is regarded as a $2 \times 2$ matrix

$$
\begin{pmatrix}
n_+\|\boldsymbol{\mu}\|^2 + R & -\dfrac{\gamma\sqrt{n_+ n_-}\|\boldsymbol{\mu}\|^2}{\sqrt{(q_+ + \gamma^2 q_-)(q_- + \gamma^2 q_+)}} \\[4mm]
-\dfrac{\gamma\sqrt{n_+ n_-}\|\boldsymbol{\mu}\|^2}{\sqrt{(q_+ + \gamma^2 q_-)(q_- + \gamma^2 q_+)}} & n_-\|\boldsymbol{\mu}\|^2 + R
\end{pmatrix}.
\tag{39}
$$

Thus, $\sum_{i=1,2} s_i^{-1} \boldsymbol{u}_i \boldsymbol{u}_i^\top$ is regarded as

$$
\begin{pmatrix}
n_+ \|\boldsymbol{\mu}\|^2 + R & -\dfrac{\gamma\sqrt{n_+ n_-}\|\boldsymbol{\mu}\|^2}{\sqrt{(q_+ + \gamma^2 q_-)(q_- + \gamma^2 q_+)}} \\
-\dfrac{\gamma\sqrt{n_+ n_-}\|\boldsymbol{\mu}\|^2}{\sqrt{(q_+ + \gamma^2 q_-)(q_- + \gamma^2 q_+)}} & n_- \|\boldsymbol{\mu}\|^2 + R
\end{pmatrix}^{-1}
$$
$$
= d^{-1}
\begin{pmatrix}
n_- \|\boldsymbol{\mu}\|^2 + R & \dfrac{\gamma\sqrt{n_+ n_-}\|\boldsymbol{\mu}\|^2}{\sqrt{(q_+ + \gamma^2 q_-)(q_- + \gamma^2 q_+)}} \\
\dfrac{\gamma\sqrt{n_+ n_-}\|\boldsymbol{\mu}\|^2}{\sqrt{(q_+ + \gamma^2 q_-)(q_- + \gamma^2 q_+)}} & n_+ \|\boldsymbol{\mu}\|^2 + R
\end{pmatrix},
\tag{40}
$$

where

$$
d = \{n_+ \|\boldsymbol{\mu}\|^2 + R\}\{n_- \|\boldsymbol{\mu}\|^2 + R\} - \frac{\gamma^2 n_+ n_- \|\boldsymbol{\mu}\|^4}{(q_+ + \gamma^2 q_-)(q_- + \gamma^2 q_+)}.
$$

Since $B^{-1}\boldsymbol{y} \leftrightarrow \frac{\sqrt{n_+}}{\sqrt{q_+ + \gamma^2 q_-}}\boldsymbol{e}_1 - \frac{\sqrt{n_-}}{\sqrt{q_- + \gamma^2 q_+}}\boldsymbol{e}_2$ and $B^{-1}\boldsymbol{y} \perp \boldsymbol{u}_i$ for all $i \geq 3$, we have

$$
\begin{aligned}
& (B^{-1}\boldsymbol{y})^\top A^{-1} \\
=& d^{-1}\left[ \left\{ \frac{n_- \|\boldsymbol{\mu}\|^2 + R}{\sqrt{q_+ + \gamma^2 q_-}} - \frac{\gamma n_- \|\boldsymbol{\mu}\|^2}{\sqrt{q_+ + \gamma^2 q_-}(q_- + \gamma^2 q_+)} \right\} \boldsymbol{1}_+ \right. \\
& \left. - \left\{ \frac{n_+ \|\boldsymbol{\mu}\|^2 + R}{\sqrt{q_- + \gamma^2 q_+}} - \frac{\gamma n_+ \|\boldsymbol{\mu}\|^2}{\sqrt{q_- + \gamma^2 q_+}(q_+ + \gamma^2 q_-)} \right\} \boldsymbol{1}_- \right]^\top \\
=& \{d(q_+ + \gamma^2 q_-)(q_- + \gamma^2 q_+)\}^{-1} \\
& \times \left[ \sqrt{q_+ + \gamma^2 q_-}\{(q_- + \gamma^2 q_+ - \gamma)n_- \|\boldsymbol{\mu}\|^2 + (q_- + \gamma^2 q_+)R\} \boldsymbol{1}_+ \right. \\
& \left. - \sqrt{q_- + \gamma^2 q_+}\{(q_+ + \gamma^2 q_- - \gamma)n_+ \|\boldsymbol{\mu}\|^2 + (q_+ + \gamma^2 q_-)R\} \boldsymbol{1}_- \right]^\top.
\end{aligned}
\tag{41}
$$

$\square$

Before proceeding to the next lemma, we note that the following inequality holds under $\tilde{E}_3$:

$$
(1 - \tilde{\varepsilon}_3^2)\frac{n^2}{4} \leq n_+ n_- \leq \frac{n^2}{4}.
\tag{42}
$$

The next lemma establishes useful bounds on $d$.

**Lemma C.7.** *We have*

$$
\begin{aligned}
d(q_+ + \gamma^2 q_-)(q_- + \gamma^2 q_+) &\leq \left\{ \gamma^2 + \frac{5(1 - \gamma^2)^2}{4}q_+ q_- \right\} (n\|\boldsymbol{\mu}\|^2 + R)^2 \\
&\leq \frac{5q_\gamma}{4}(n\|\boldsymbol{\mu}\|^2 + R)^2.
\end{aligned}
\tag{43}
$$

*If additionally $\tilde{E}_3(\tilde{\varepsilon}_3)$ holds for some $\tilde{\varepsilon}_3 \in [0, 1]$, then we also have*

$$
d(q_+ + \gamma^2 q_-)(q_- + \gamma^2 q_+) \geq \frac{(1 - \tilde{\varepsilon}_3^2)(1 - \gamma^2)^2 q_+ q_-}{8}(n\|\boldsymbol{\mu}\|^2 + R)^2 + \gamma^2 R^2.
\tag{44}
$$

*Proof.* We first note that

$$
q_\gamma \geq (q_+ + \gamma^2 q_-)(q_- + \gamma^2 q_+) = (1 + \gamma^4)q_+ q_- + \gamma^2(q_+^2 + q_-^2) = \gamma^2 + (1 - \gamma^2)^2 q_+ q_-.
\tag{45}
$$

With (45) and also noting that $n_+ n_- \leq \frac{n^2}{4}$, it is easy to see (43) follows from (36).

By (36), (45), and (42), we have

$$d(q_+ + \gamma^2 q_-)(q_- + \gamma^2 q_+)$$
$$\geq \frac{(1 - \tilde{\varepsilon}_3^2)(1 - \gamma^2)^2}{4} q_+ q_- \left(n\|\boldsymbol{\mu}\|^2\right)^2 + \left\{\gamma^2 + (1 - \gamma^2)^2 q_+ q_-\right\} R^2$$
$$\geq \frac{(1 - \tilde{\varepsilon}_3^2)(1 - \gamma^2)^2}{4} q_+ q_- \left\{\left(n\|\boldsymbol{\mu}\|^2\right)^2 + R^2\right\} + \gamma^2 R^2$$
$$\geq \frac{(1 - \tilde{\varepsilon}_3^2)(1 - \gamma^2)^2}{8} q_+ q_- \left(n\|\boldsymbol{\mu}\|^2 + R\right)^2 + \gamma^2 R^2,$$

where the last inequality is due to $a^2 + b^2 \geq \frac{(a+b)^2}{2}$.

$\square$

Now we are ready to establish sufficient conditions of $\hat{\boldsymbol{w}} = \tilde{X}(\tilde{X}\tilde{X}^\top)^{-1}\boldsymbol{y}$.

**Lemma C.8.** *Suppose event $\bigcap_{i=1,2,3} \tilde{E}_i$ holds with $\tilde{\varepsilon} \leq \frac{q_\gamma}{2}$ and further assume one of the following conditions holds:*

(i) $q_+ q_- = 0$, $n\|\boldsymbol{\mu}\|^2 < \frac{\gamma}{1 + (1+\gamma)\tilde{\varepsilon}_3} R$, *and* $\sqrt{n}\tilde{\varepsilon} \leq \frac{\gamma^3}{4}$,

(ii) $q_+ q_- \neq 0$, $n\|\boldsymbol{\mu}\|^2 \leq \lambda R$, *and* $\sqrt{n}\tilde{\varepsilon} \leq \frac{q_\gamma^2}{5(1+\lambda)^2}$ *for some* $\lambda \leq \frac{q_\gamma}{2\gamma(1-\gamma)}$,

(iii) $q_\gamma > \gamma$, $n\|\boldsymbol{\mu}\|^2 \geq \lambda R$, *and* $\sqrt{n}\tilde{\varepsilon} \leq \tilde{C}(\gamma, q_+, \tilde{\varepsilon}_3, \lambda)\frac{R}{n\|\boldsymbol{\mu}\|^2}$ *for some* $\lambda > 0$, *where*

$$\tilde{C}(\gamma, q_+, \tilde{\varepsilon}_3, \lambda) := \frac{(1 - \tilde{\varepsilon}_3)q_\gamma^2(q_\gamma - \gamma)}{5(1 + \lambda^{-1})^2}.$$

*Then, we have*

$$\hat{\boldsymbol{w}} = \tilde{X}(\tilde{X}\tilde{X}^\top)^{-1}y.$$

*Proof.* By Lemma C.1, it suffices to show $\boldsymbol{y}^\top(\tilde{X}\tilde{X}^\top)^{-1}y_i\boldsymbol{e}_i > 0$ for all $i$.

By Lemma C.5, we have

$$\boldsymbol{y}^\top(\tilde{X}\tilde{X}^\top)^{-1}y_i\boldsymbol{e}_i$$
$$=(B^{-1}\boldsymbol{y})^\top B(\tilde{X}\tilde{X}^\top)^{-1}BB^{-1}y_i\boldsymbol{e}_i$$
$$\geq(B^{-1}\boldsymbol{y})^\top A^{-1}B^{-1}y_i\boldsymbol{e}_i - \|B^{-1}\boldsymbol{y}\|B_{ii}^{-1}2\tilde{\varepsilon}q_\gamma^{-1}R^{-1}$$
$$\geq B_{ii}^{-1}(B^{-1}\boldsymbol{y})^\top A^{-1}y_i\boldsymbol{e}_i - 2B_{ii}^{-1}q_\gamma^{-1}\tilde{\varepsilon}R^{-1}\sqrt{\frac{n_+}{q_+ + \gamma^2 q_-} + \frac{n_-}{q_- + \gamma^2 q_+}}$$
$$\geq B_{ii}^{-1}\left\{(B^{-1}\boldsymbol{y})^\top A^{-1}y_i\boldsymbol{e}_i - 2q_\gamma^{-3/2}\tilde{\varepsilon}R^{-1}\sqrt{n}\right\},$$

(46)

where the last inequality follows from the definition of $q_\gamma$ and $n_+ + n_- = n$. The rest of the argument is split in cases.

**Case $q_+ = 1, q_- = 0$:** By (33), we have

$$(B_+^{-1}\boldsymbol{y})^\top A^{-1}B_+^{-1}y_i\boldsymbol{e}_i$$
$$=\begin{cases} R^{-1}\left(1 - \frac{n\|\boldsymbol{\mu}\|^2}{n\|\boldsymbol{\mu}\|^2+R}\frac{n_+ + \gamma^{-1}n_-}{n}\right), & \text{if } i \in I_+, \\ \gamma^{-1}R^{-1}\left(\gamma^{-1} - \frac{n\|\boldsymbol{\mu}\|^2}{n\|\boldsymbol{\mu}\|^2+R}\frac{n_+ + \gamma^{-1}n_-}{n}\right), & \text{if } i \in I_-. \end{cases}$$

(47)

Therefore, by the definition of $\tilde{E}_3$, (46) and (47), we have

$$\boldsymbol{y}^\top(\tilde{X}\tilde{X}^\top)^{-1}y_i\boldsymbol{e}_i$$
$$\geq\begin{cases} R^{-1}\left(1 - \frac{n\|\boldsymbol{\mu}\|^2}{n\|\boldsymbol{\mu}\|^2+R}\frac{(1+\gamma^{-1})(1+\tilde{\varepsilon}_3)}{2} - 2\gamma^{-3}\sqrt{n}\tilde{\varepsilon}\right), & \text{if } i \in I_+, \\ \gamma^{-1}R^{-1}\left(\gamma^{-1} - \frac{n\|\boldsymbol{\mu}\|^2}{n\|\boldsymbol{\mu}\|^2+R}\frac{(1+\gamma^{-1})(1+\tilde{\varepsilon}_3)}{2} - 2\gamma^{-3}\sqrt{n}\tilde{\varepsilon}\right), & \text{if } i \in I_-. \end{cases}$$

(48)

**Case** $q_+ = 0, q_- = 1$**:** By (34), we have

$$
\begin{aligned}
&(B_-^{-1}\boldsymbol{y})^\top A^{-1} B_-^{-1} y_i \boldsymbol{e}_i \\
&= \begin{cases} \gamma^{-1} R^{-1} \left( \gamma^{-1} - \frac{n\|\boldsymbol{\mu}\|^2}{n\|\boldsymbol{\mu}\|^2 + R} \frac{\gamma^{-1} n_+ + n_-}{n} \right), & \text{if } i \in I_+ \\ R^{-1} \left( 1 - \frac{n\|\boldsymbol{\mu}\|^2}{n\|\boldsymbol{\mu}\|^2 + R} \frac{\gamma^{-1} n_+ + n_-}{n} \right), & \text{if } i \in I_-. \end{cases}
\end{aligned}
\tag{49}
$$

Therefore, by the definition of $\tilde{E}_3$, (46) and (49), we have

$$
\begin{aligned}
&\boldsymbol{y}^\top (\tilde{X}\tilde{X}^\top)^{-1} y_i \boldsymbol{e}_i \\
&\geq \begin{cases} \gamma^{-1} R^{-1} \left( \gamma^{-1} - \frac{n\|\boldsymbol{\mu}\|^2}{n\|\boldsymbol{\mu}\|^2 + R} \frac{(1+\gamma^{-1})(1+\tilde{\varepsilon}_3)}{2} - 2\gamma^{-3}\sqrt{n}\tilde{\varepsilon} \right), & \text{if } i \in I_+, \\ R^{-1} \left( 1 - \frac{n\|\boldsymbol{\mu}\|^2}{n\|\boldsymbol{\mu}\|^2 + R} \frac{(1+\gamma^{-1})(1+\tilde{\varepsilon}_3)}{2} - 2\gamma^{-3}\sqrt{n}\tilde{\varepsilon} \right), & \text{if } i \in I_-. \end{cases}
\end{aligned}
\tag{50}
$$

Therefore, for both the cases above, it suffices to have

$$
\frac{n\|\boldsymbol{\mu}\|^2}{n\|\boldsymbol{\mu}\|^2 + R} \frac{(1+\gamma^{-1})(1+\tilde{\varepsilon}_3)}{2} < \frac{1}{2}, \quad \text{and} \quad 2\gamma^{-3}\sqrt{n}\tilde{\varepsilon} \leq \frac{1}{2},
$$

which follows from the condition (i).

**Case** $q_+ q_- \neq 0$**:** By (41) and (46), we have for $i \in I_+$

$$
\begin{aligned}
&\boldsymbol{y}^\top (\tilde{X}\tilde{X}^\top)^{-1} y_i \boldsymbol{e}_i \\
&\geq B_{ii}^{-1} \Big[ \{d(q_+ + \gamma^2 q_-)(q_- + \gamma^2 q_+)\}^{-1} \\
&\quad \times \Big\{ \sqrt{q_+ + \gamma^2 q_-} \{(q_- + \gamma^2 q_+ - \gamma) n_- \|\boldsymbol{\mu}\|^2 + (q_- + \gamma^2 q_+) R\} \\
&\quad - 2 q_\gamma^{-3/2} R^{-1} \sqrt{n}\tilde{\varepsilon} \Big],
\end{aligned}
\tag{51}
$$

and for $i \in I_-$

$$
\begin{aligned}
&\boldsymbol{y}^\top (\tilde{X}\tilde{X}^\top)^{-1} y_i \boldsymbol{e}_i \\
&\geq B_{ii}^{-1} \Big[ \{d(q_+ + \gamma^2 q_-)(q_- + \gamma^2 q_+)\}^{-1} \\
&\quad \times \Big\{ \sqrt{q_- + \gamma^2 q_+} \{(q_+ + \gamma^2 q_- - \gamma) n_+ \|\boldsymbol{\mu}\|^2 + (q_+ + \gamma^2 q_-) R\} \\
&\quad - 2 q_\gamma^{-3/2} R^{-1} \sqrt{n}\tilde{\varepsilon} \Big].
\end{aligned}
\tag{52}
$$

Under regime (ii), (43) implies

$$
d(q_+ + \gamma^2 q_-)(q_- + \gamma^2 q_+) \leq \frac{5 q_\gamma}{4} (1+\lambda)^2 R^2.
\tag{53}
$$

By $q_\gamma > \gamma^2$, we have

$$
\begin{aligned}
(q_- + \gamma^2 q_+ - \gamma) n_- \|\boldsymbol{\mu}\|^2 + (q_- + \gamma^2 q_+) R &> q_\gamma R - \gamma(1-\gamma) n_- \|\boldsymbol{\mu}\|^2 \\
&\geq q_\gamma R - \gamma(1-\gamma) n \|\boldsymbol{\mu}\|^2 \\
&\geq \frac{q_\gamma}{2} R,
\end{aligned}
\tag{54}
$$

where the last inequality is due to the assumption on $\lambda$.

Similarly, we have

$$
(q_+ + \gamma^2 q_- - \gamma) n_+ \|\boldsymbol{\mu}\|^2 + (q_+ + \gamma^2 q_-) R > \frac{q_\gamma}{2} R.
\tag{55}
$$

Therefore, for $i \in I_+$, (51), (53), and (54) imply

$$\boldsymbol{y}^\top (\tilde{X} \tilde{X}^\top)^{-1} y_i \boldsymbol{e}_i > B_{ii}^{-1} R^{-1} \left[ \frac{2 q_\gamma^{\frac{1}{2}}}{5(1+\lambda)^2} - 2 q_\gamma^{-\frac{3}{2}} \sqrt{n} \tilde{\varepsilon} \right] \geq 0, \tag{56}$$

where the last inequality is due to the assumption on $\sqrt{n} \tilde{\varepsilon}$.

The same conclusion analogously holds for $i \in I_-$ as well by (52), (53), and (54).

Under regime (iii), by (43), we have that

$$d(q_+ + \gamma^2 q_-)(q_- + \gamma^2 q_+) \leq \frac{5 q_\gamma}{4} (1 + \lambda^{-1})^2 \left( n \|\boldsymbol{\mu}\|^2 \right)^2. \tag{57}$$

With (51) and (57), we have for $i \in I_+$

$$\boldsymbol{y}^\top (\tilde{X} \tilde{X}^\top)^{-1} y_i \boldsymbol{e}_i$$
$$> B_{ii}^{-1} \left[ \frac{q_\gamma^{\frac{1}{2}} \{ (q_\gamma - \gamma)(1 - \tilde{\varepsilon}_3) \frac{n}{2} \|\boldsymbol{\mu}\|^2 \}}{\frac{5 q_\gamma}{4} (1 + \lambda^{-1})^2 (n \|\boldsymbol{\mu}\|^2)^2} - 2 q_\gamma^{-\frac{3}{2}} \sqrt{n} \tilde{\varepsilon} R^{-1} \right]$$
$$= (B_{ii} n \|\boldsymbol{\mu}\|^2)^{-1} q_\gamma^{-\frac{3}{2}} \left[ \frac{q_\gamma^2 (q_\gamma - \gamma)(1 - \tilde{\varepsilon}_3)}{5(1 + \lambda^{-1})^2} - 2 \sqrt{n} \tilde{\varepsilon} \frac{n \|\boldsymbol{\mu}\|^2}{R} \right] \geq 0. \tag{58}$$

Similarly, the same lower bound holds for $i \in I_-$ from (52).

$\square$

*Remark.* By similar argument we can show under (iii), an upper bound of the similar form as (58) for $i \in I_+$ whose dominant term is $(q_- + \gamma^2 q_+ - \gamma) n \|\boldsymbol{\mu}\|^2$. Similarly, for $i \in I_-$, we have an upper bound whose dominant term is $(q_+ + \gamma^2 q_- - \gamma) n \|\boldsymbol{\mu}\|^2$. If we have $q_\gamma < \gamma$, exactly one of $q_- + \gamma^2 q_+ - \gamma$ and $q_+ + \gamma^2 q_- - \gamma$ becomes negative and the other stays positive. Suppose $q_- + \gamma^2 q_+ - \gamma < 0$, then we have $\boldsymbol{y}^\top (\tilde{X} \tilde{X}^\top)^{-1} y_i \boldsymbol{e}_i < 0$ for all $i \in I_+$ when $n \|\boldsymbol{\mu}\|^2$ is sufficiently larger than $R$, which means the equivalence of $\hat{\boldsymbol{w}} = \tilde{X}^\top (\tilde{X} \tilde{X}^\top)^{-1} \boldsymbol{y}$ provably fails in such case. The same applies with the case $q_+ + \gamma^2 q_- - \gamma < 0$.

With $\hat{\boldsymbol{w}} = \tilde{X}^\top (\tilde{X} \tilde{X}^\top)^{-1} y$, by defining

$$\boldsymbol{w}_+ = \frac{1}{\sqrt{m}} X^\top B_+ (\tilde{X} \tilde{X}^\top)^{-1} \boldsymbol{y}$$
$$= \frac{1}{\sqrt{m}} X^\top B_+ \left( q_+ B_+ X X^\top B_+ + q_- B_- X X^\top B_- \right)^{-1} \boldsymbol{y},$$
$$\boldsymbol{w}_- = -\frac{1}{\sqrt{m}} X^\top B_- (\tilde{X} \tilde{X}^\top)^{-1} \boldsymbol{y}$$
$$= -\frac{1}{\sqrt{m}} X^\top B_- \left( q_+ B_+ X X^\top B_+ + q_- B_- X X^\top B_- \right)^{-1} \boldsymbol{y}, \tag{59}$$

we have $\hat{\boldsymbol{w}}_j = \boldsymbol{w}_+$ if $j \in J_+$ and $\hat{\boldsymbol{w}}_j = \boldsymbol{w}_-$ if $j \in J_-$.

By Lemma B.2, the network $f(x; \hat{W})$ has the linear decision boundary defined by

$$\bar{\boldsymbol{w}} = \frac{|J_+|}{\sqrt{m}} \boldsymbol{w}_+ - \frac{|J_-|}{\sqrt{m}} \boldsymbol{w}_-. \tag{60}$$

By (59) and (60), we have

$$\bar{\boldsymbol{w}} = X^\top \left( q_+ B_+ + q_- B_- \right) (\tilde{X} \tilde{X}^\top)^{-1} \boldsymbol{y}.$$

This implies

$$\|\bar{\boldsymbol{w}}\|^2 = \boldsymbol{y}^\top (\tilde{X} \tilde{X}^\top)^{-1} \left( q_+ B_+ + q_- B_- \right) X X^\top \left( q_+ B_+ + q_- B_- \right) (\tilde{X} \tilde{X}^\top)^{-1} \boldsymbol{y}, \tag{61}$$

$$\langle \bar{\boldsymbol{w}}, \boldsymbol{\mu} \rangle = \boldsymbol{y}^\top (\tilde{X} \tilde{X}^\top)^{-1} \left( q_+ B_+ + q_- B_- \right) X \boldsymbol{\mu}$$
$$= \boldsymbol{y}^\top (\tilde{X} \tilde{X}^\top)^{-1} \left( q_+ B_+ + q_- B_- \right) \left( \|\boldsymbol{\mu}\|^2 \boldsymbol{y} + Z \boldsymbol{\mu} \right). \tag{62}$$

We need to obtain bounds of $\|\bar{w}\|^2$ and $\langle \bar{w}, \mu \rangle$, respectively, in order to later establish a classification error bound of $\bar{w}$.

To obtain bounds of $\|\bar{w}\|^2$, the following inequality is useful.

**Lemma C.9.** *Suppose that the conclusions of Lemma C.2, C.5, and C.8 hold. Then, we have*

$$\left| \|\bar{w}\|^2 - (\|\mu\|^2 K_1 + R K_2) \right| \leq \tilde{\varepsilon} R K_2,$$

*and*

$$\left| \langle \bar{w}, \mu \rangle - \|\mu\|^2 \sqrt{K_1} \right| \leq \frac{q_\gamma^{-\frac{1}{2}}}{3} \tilde{\varepsilon}_2 \|(B^{-1}y)^\top A^{-1}\| R + \frac{2q_\gamma^{-2}}{3} \tilde{\varepsilon} \cdot \sqrt{n} \tilde{\varepsilon}_2,$$

*where*

$$
\begin{aligned}
K_1 &:= \left\{ y^\top (\tilde{X}\tilde{X}^\top)^{-1}(q_+ B_+ + q_- B_-) y \right\}^2 \\
&= \left\{ (B^{-1}y)^\top B(\tilde{X}\tilde{X}^\top)^{-1} B \cdot B^{-1}(q_+ B_+ + q_- B_-) y \right\}^2 \text{ and} \quad (63) \\
K_2 &:= \left\| (q_+ B_+ + q_- B_-)(\tilde{X}\tilde{X}^\top)^{-1} y \right\|^2 \\
&= \left\| (q_+ B_+ + q_- B_-) B^{-1} \cdot B(\tilde{X}\tilde{X}^\top)^{-1} B \cdot B^{-1} y \right\|^2. \quad (64)
\end{aligned}
$$

*Proof.* Letting $v = (q_+ B_+ + q_- B_-)(\tilde{X}\tilde{X}^\top)^{-1} y$, we have $\|\bar{w}\|^2 = v^\top X X^\top v$.

Then, by (61) and the conclusion of Lemma C.2, we have

$$\left| \|\bar{w}\|^2 - (\|\mu\|^2 \langle v, y \rangle^2 + R \|v\|^2) \right| \leq \tilde{\varepsilon} R \|v\|^2.$$

The first desired inequality now follows by noting $\langle v, y \rangle^2 = K_1$ and $\|v\|^2 = K_2$.

To show the second desired inequality, we note that (62) implies

$$
\begin{aligned}
\langle \bar{w}, \mu \rangle = \|\mu\|^2 \sqrt{K_1} &+ (B^{-1}y)^\top A^{-1} B^{-1}(q_+ B_+ + q_- B_-) Z\mu \\
&+ (B^{-1}y)^\top \left[ B(\tilde{X}\tilde{X}^\top)^{-1} - A^{-1} \right] B^{-1}(q_+ B_+ + q_- B_-) Z\mu.
\end{aligned}
\quad (65)
$$

By Lemma C.5 and (65), we have

$$
\begin{aligned}
\left| \langle \bar{w}, \mu \rangle - \|\mu\|^2 \sqrt{K_1} \right| &\leq \|(B^{-1}y)^\top A^{-1}\| \|B^{-1}\| \|q_+ B_+ + q_- B_-\| \|Z\mu\| \\
&\quad + \|B^{-1}\| \|y\| \frac{2q_\gamma^{-1}\tilde{\varepsilon}}{R} \|B^{-1}\| \|q_+ B_+ + q_- B_-\| \|Z\mu\| \\
&\leq \frac{q_\gamma^{-\frac{1}{2}}}{3} \tilde{\varepsilon}_2 \|(B^{-1}y)^\top A^{-1}\| R + \frac{2q_\gamma^{-2}}{3} \tilde{\varepsilon} \cdot \sqrt{n} \tilde{\varepsilon}_2,
\end{aligned}
$$

where the second inequality follows from the definition of $\tilde{\varepsilon}_2$, $\|B^{-1}\| \leq q_\gamma^{-\frac{1}{2}}$, and $\|q_+ B_+ + q_- B_-\| \leq 1$.

$\square$

**Lemma C.10.** *Suppose that all the assumptions of Lemma C.8 hold with $\tilde{\varepsilon}_3 \leq \frac{1}{2}$ in (ii) and (iii). Then there exists constant $\tilde{c}_1$ which depends only on $\gamma$ and $q_+$ such that*

$$\|\bar{w}\|^2 \leq \tilde{c}_1 \frac{n}{n\|\mu\|^2 + R}.$$

*Furthermore, there exist constants $\hat{c}_1$ and $N$ which only depends on $\gamma$, $q_+$ (and $\lambda$ under (ii) and (iii)) such that $n \geq N$ implies*

$$\|\bar{w}\|^2 \geq \hat{c}_1 \frac{n}{n\|\mu\|^2 + R}.$$

*Proof.* By Lemma C.9, the proof is reduced to bounding $K_i, i = 1, 2$.

By Lemma C.5, we have

$$
\begin{aligned}
&\left| (B^{-1}\boldsymbol{y})^\top B(\tilde{X}\tilde{X}^\top)^{-1}B \cdot B^{-1}(q_+B_+ + q_-B_-)\boldsymbol{y} - (B^{-1}\boldsymbol{y})^\top A^{-1}B^{-1}(q_+B_+ + q_-B_-)\boldsymbol{y} \right| \\
\leq& \frac{2q_\gamma^{-1}\tilde{\varepsilon}}{R} \|B^{-1}\boldsymbol{y}\| \|B^{-1}(q_+B_+ + q_-B_-)\boldsymbol{y}\| \\
\leq& \frac{2q_\gamma^{-1}\tilde{\varepsilon}}{R} \left\| \frac{1}{\sqrt{q_+ + \gamma^2 q_-}}\mathbf{1}_+ - \frac{1}{\sqrt{q_- + \gamma^2 q_+}}\mathbf{1}_- \right\| \cdot \left\| \frac{q_+ + \gamma q_-}{\sqrt{q_+ + \gamma^2 q_-}}\mathbf{1}_+ - \frac{q_- + \gamma q_+}{\sqrt{q_- + \gamma^2 q_+}}\mathbf{1}_- \right\| \\
\leq& \frac{2q_\gamma^{-1}\tilde{\varepsilon}}{R} \sqrt{\frac{1}{q_+ + \gamma^2 q_-}n_+ + \frac{1}{q_- + \gamma^2 q_+}n_-} \cdot \sqrt{\frac{(q_+ + \gamma q_-)^2}{q_+ + \gamma^2 q_-}n_+ + \frac{(q_- + \gamma q_+)^2}{q_- + \gamma^2 q_+}n_-} \\
\leq& \frac{2q_\gamma^{-1}\tilde{\varepsilon}}{R} \sqrt{\frac{n}{q_\gamma}} \cdot \sqrt{\frac{n}{q_\gamma}} \\
\leq& \frac{2q_\gamma^{-2}n\tilde{\varepsilon}}{R}.
\end{aligned}
$$

$$(66)$$

Similarly, we have

$$
\begin{aligned}
&\left| \|(q_+B_+ + q_-B_-)B^{-1} \cdot B(\tilde{X}\tilde{X}^\top)^{-1}B \cdot B^{-1}\boldsymbol{y}\| - \|(q_+B_+ + q_-B_-)B^{-1} \cdot A^{-1} \cdot B^{-1}\boldsymbol{y}\| \right| \\
\leq& \frac{2q_\gamma^{-1}\tilde{\varepsilon}}{R} \|(q_+B_+ + q_-B_-)B^{-1}\| \cdot \|B^{-1}\boldsymbol{y}\| \\
\leq& \frac{2q_\gamma^{-1}\tilde{\varepsilon}}{R} \max\left\{ \frac{q_+ + \gamma q_-}{\sqrt{q_+ + \gamma^2 q_-}}, \frac{q_- + \gamma q_+}{\sqrt{q_- + \gamma^2 q_+}} \right\} \cdot \sqrt{\frac{n}{q_\gamma}} \\
\leq& \frac{2q_\gamma^{-1}\tilde{\varepsilon}}{R} \frac{1}{\sqrt{q_\gamma}} \cdot \sqrt{\frac{n}{q_\gamma}} \\
\leq& \frac{2q_\gamma^{-2}\sqrt{n}\tilde{\varepsilon}}{R}.
\end{aligned}
$$

$$(67)$$

We split in cases.

**Regime (i):** We first note that $n\|\boldsymbol{\mu}\|^2 < \frac{\gamma}{1 + (1+\gamma)\tilde{\varepsilon}_3}R \leq \gamma R$ implies

$$
R^{-1} \leq (1 + \gamma)(n\|\boldsymbol{\mu}\|^2 + R)^{-1}. \tag{68}
$$

We first consider the case when $q_+ = 1, q_- = 0$. By (33), we have

$$
\begin{aligned}
&(B^{-1}\boldsymbol{y})^\top A^{-1}B^{-1}(q_+B_+ + q_-B_-)\boldsymbol{y} \\
=&(B_+^{-1}\boldsymbol{y})^\top A^{-1}(B_+^{-1}\boldsymbol{y}) \\
=&R^{-1}\left\{ \left( 1 - \frac{n\|\boldsymbol{\mu}\|^2}{n\|\boldsymbol{\mu}\|^2 + R}\frac{n_+ + \gamma^{-1}n_-}{n} \right)n_+ \right. \\
&\left. + \gamma^{-1}\left( \gamma^{-1} - \frac{n\|\boldsymbol{\mu}\|^2}{n\|\boldsymbol{\mu}\|^2 + R}\frac{n_+ + \gamma^{-1}n_-}{n} \right)n_- \right\} \\
\leq& \frac{n_+ + \gamma^{-2}n_-}{R} \\
\leq& (1 + \gamma)\gamma^{-2}\frac{n}{n\|\boldsymbol{\mu}\|^2 + R},
\end{aligned}
$$

$$(69)$$

where the last line is due to (68).

We also have

$$
\begin{aligned}
&(B^{-1}\boldsymbol{y})^{\top} A^{-1} B^{-1}(q_+ B_+ + q_- B_-)\boldsymbol{y} \\
\geq{}&R^{-1}\left\{\left(1 - \frac{n\|\boldsymbol{\mu}\|^2}{n\|\boldsymbol{\mu}\|^2 + R}\frac{(1+\tilde{\varepsilon}_3)(1+\gamma^{-1})}{2}\right) n_+\right. \\
&\left. + \gamma^{-1}\left(\gamma^{-1} - \frac{n\|\boldsymbol{\mu}\|^2}{n\|\boldsymbol{\mu}\|^2 + R}\frac{(1+\tilde{\varepsilon}_3)(1+\gamma^{-1})}{2}\right) n_-\right\} \\
\geq{}&R^{-1}\left\{\left(1 - \frac{1}{2}\right) n_+ + \gamma^{-1}\left(\gamma^{-1} - \frac{1}{2}\right) n_-\right\} \\
\geq{}&\{2(1+\gamma)\}^{-1}\frac{n}{n\|\boldsymbol{\mu}\|^2 + R},
\end{aligned}
\tag{70}
$$

where the first inequality is due to $\tilde{E}_3$, the second inequality is due to the assumption $n\|\boldsymbol{\mu}\|^2 < \frac{\gamma}{1+(1+\gamma)\tilde{\varepsilon}_3}R$, and the last inequality is due to $\gamma \in (0,1)$.

Analogously, we have the same results for $q_+ = 0, q_- = 1$.

With these, (66) with (68), and noting that $q_\gamma = \gamma^2$ when $q_+ q_- = 0$, we have under the regime (i)

$$
\begin{aligned}
\left\{\frac{3}{8(1+\gamma)}\right\}^2 \frac{n^2}{(n\|\boldsymbol{\mu}\|^2+R)^2} \leq{}& \left(\frac{1}{2(1+\gamma)} - 2(1+\gamma)\gamma^{-4}\tilde{\varepsilon}\right)^2 \frac{n^2}{(n\|\boldsymbol{\mu}\|^2+R)^2} \\
\leq{}& K_1 \leq (1+\gamma)^2 \left(\gamma^{-2} + 2\gamma^{-4}\tilde{\varepsilon}\right)^2 \frac{n^2}{(n\|\boldsymbol{\mu}\|^2+R)^2}
\end{aligned}
\tag{71}
$$

for sufficiently large $n$.

Similarly, (33) implies

$$
\begin{aligned}
&\left\|(q_+ B_+ + q_- B_-)B^{-1} \cdot A^{-1} \cdot B^{-1}\boldsymbol{y}\|\right\| \\
={}&\left\|A^{-1} \cdot B_+^{-1}\boldsymbol{y}\|\right\| \\
={}&R^{-1}\left\{\left(1 - \frac{n\|\boldsymbol{\mu}\|^2}{n\|\boldsymbol{\mu}\|^2+R}\frac{n_+ + \gamma^{-1}n_-}{n}\right)^2 n_+ \right. \\
&\left. + \gamma^{-2}\left(\gamma^{-1} - \frac{n\|\boldsymbol{\mu}\|^2}{n\|\boldsymbol{\mu}\|^2+R}\frac{n_+ + \gamma^{-1}n_-}{n}\right)^2 n_-\right\}^{\frac{1}{2}},
\end{aligned}
\tag{72}
$$

and so under the regime (i), we have

$$
\frac{1}{2(1+\gamma)}\frac{\sqrt{n}}{n\|\boldsymbol{\mu}\|^2+R} \leq \left\|(q_+ B_+ + q_- B_-)B^{-1} \cdot A^{-1} \cdot B^{-1}\boldsymbol{y}\|\right\| \leq (1+\gamma)\gamma^{-2}\frac{\sqrt{n}}{n\|\boldsymbol{\mu}\|^2+R}.
\tag{73}
$$

With (67), we have

$$
\begin{aligned}
\left\{\frac{3}{8(1+\gamma)}\right\}^2 \frac{n}{(n\|\boldsymbol{\mu}\|^2+R)^2} \leq{}& \left(\frac{1}{2(1+\gamma)} - 2(1+\gamma)\gamma^{-4}\tilde{\varepsilon}\right)^2 \frac{n}{R^2} \\
\leq{}& K_2 \leq (1+\gamma)^2 \left(\gamma^{-2} + 2\gamma^{-4}\tilde{\varepsilon}\right)^2 \frac{n}{(n\|\boldsymbol{\mu}\|^2+R)^2}
\end{aligned}
\tag{74}
$$

for sufficiently large $n$.

Thus, by Lemma C.9, (71), and (74), we have

$$
\begin{aligned}
\left[\left\{\frac{3}{8(1+\gamma)}\right\}^2 - (1+\gamma)^2 \left(\gamma^{-2} + 2\gamma^{-4}\tilde{\varepsilon}\right)^2 \tilde{\varepsilon}\right] \frac{n}{n\|\boldsymbol{\mu}\|^2 + R} \\
\leq \|\bar{\boldsymbol{w}}\|^2 \leq (1+\tilde{\varepsilon})(1+\gamma)^2 \left(\gamma^{-2} + 2\gamma^{-4}\tilde{\varepsilon}\right)^2 \frac{n}{n\|\boldsymbol{\mu}\|^2 + R},
\end{aligned}
$$

with the left hand side being positive for sufficiently large $n$.

**Case $q_+q_- \neq 0$:** By (35), we have

$$
\begin{aligned}
&(B^{-1}\boldsymbol{y})^\top A^{-1}B^{-1}(q_+B_+ + q_-B_-)\boldsymbol{y} \\
&= \{d(q_+ + \gamma^2 q_-)(q_- + \gamma^2 q_+)\}^{-1} \\
&\quad \times \Big[(q_+ + \gamma q_-)\{(q_- + \gamma^2 q_+ - \gamma)n_-\|\boldsymbol{\mu}\|^2 + (q_- + \gamma^2 q_+)R\}n_+ \\
&\qquad + (q_- + \gamma q_+)\{(q_+ + \gamma^2 q_- - \gamma)n_+\|\boldsymbol{\mu}\|^2 + (q_+ + \gamma^2 q_-)R\}n_-\Big].
\end{aligned}
\tag{75}
$$

Thus, we have with Lemma C.7

$$
\begin{aligned}
(B^{-1}\boldsymbol{y})^\top A^{-1}B^{-1}(q_+B_+ + q_-B_-)\boldsymbol{y} &\leq \{d(q_+ + \gamma^2 q_-)(q_- + \gamma^2 q_+)\}^{-1} n(n\|\boldsymbol{\mu}\|^2 + R) \\
&\leq \frac{8}{(1 - \tilde{\varepsilon}_3^2)(1 - \gamma^2)^2 q_+ q_-} \frac{n}{n\|\boldsymbol{\mu}\|^2 + R}.
\end{aligned}
\tag{76}
$$

Similarly, (35) implies

$$
\begin{aligned}
&\big\|(q_+B_+ + q_-B_-)B^{-1} \cdot A^{-1} \cdot B^{-1}\boldsymbol{y}\|\big\| \\
&= \{d(q_+ + \gamma^2 q_-)(q_- + \gamma^2 q_+)\}^{-1} \\
&\quad \times \Big[(q_+ + \gamma q_-)^2\{(q_- + \gamma^2 q_+ - \gamma)n_-\|\boldsymbol{\mu}\|^2 + (q_- + \gamma^2 q_+)R\}^2 n_+ \\
&\qquad + (q_- + \gamma q_+)^2\{(q_+ + \gamma^2 q_- - \gamma)n_+\|\boldsymbol{\mu}\|^2 + (q_+ + \gamma^2 q_-)R\}^2 n_-\Big]^{\frac{1}{2}}
\end{aligned}
\tag{77}
$$

Thus, we have with Lemma C.7

$$
\begin{aligned}
\big\|(q_+B_+ + q_-B_-)B^{-1} \cdot A^{-1} \cdot B^{-1}\boldsymbol{y}\|\big\| &\leq \{d(q_+ + \gamma^2 q_-)(q_- + \gamma^2 q_+)\}^{-1}\sqrt{n}(n\|\boldsymbol{\mu}\|^2 + R^2) \\
&\leq \frac{8}{(1 - \tilde{\varepsilon}_3^2)(1 - \gamma^2)^2 q_+ q_-} \frac{\sqrt{n}}{n\|\boldsymbol{\mu}\|^2 + R}.
\end{aligned}
\tag{78}
$$

We now consider regime (ii) and (iii) separately.

**Regime (ii):** By (66) and (76), we have

$$
K_1 \leq \left\{ \frac{8}{(1 - \tilde{\varepsilon}_3^2)(1 - \gamma^2)^2 q_+ q_-} + 2(1 + \lambda)q_\gamma^{-2}\tilde{\varepsilon} \right\}^2 \frac{n^2}{(n\|\boldsymbol{\mu}\|^2 + R)^2}.
\tag{79}
$$

Similarly, (67) and (78) imply

$$
K_2 \leq \left\{ \frac{8}{(1 - \tilde{\varepsilon}_3^2)(1 - \gamma^2)^2 q_+ q_-} + 2(1 + \lambda)q_\gamma^{-2}\tilde{\varepsilon} \right\}^2 \frac{n}{(n\|\boldsymbol{\mu}\|^2 + R)^2}.
\tag{80}
$$

(54) and (75) imply

$$
\begin{aligned}
&(B^{-1}\boldsymbol{y})^\top A^{-1}B^{-1}(q_+B_+ + q_-B_-)\boldsymbol{y} \\
&\geq \{d(q_+ + \gamma^2 q_-)(q_- + \gamma^2 q_+)\}^{-1}\frac{q_\gamma}{2}Rn \\
&\geq \{d(q_+ + \gamma^2 q_-)(q_- + \gamma^2 q_+)\}^{-1}\frac{q_\gamma^2}{2(1 + \lambda)}n(n\|\boldsymbol{\mu}\|^2 + R).
\end{aligned}
\tag{81}
$$

By Lemma C.7, (66), (81), and the assumption on $\tilde{\varepsilon}$, we have for sufficiently large $n$

$$
K_1 \geq \left\{ \frac{2q_\gamma}{5(1 + \lambda)} - 2(1 + \lambda)q_\gamma^{-2}\tilde{\varepsilon} \right\}^2 \frac{n^2}{(n\|\boldsymbol{\mu}\|^2 + R)^2} \geq \left\{ \frac{q_\gamma}{5(1 + \lambda)} \right\}^2 \frac{n^2}{(n\|\boldsymbol{\mu}\|^2 + R)^2}.
\tag{82}
$$

Similarly, (54) and (77) imply

$$
\left\| (q_+ B_+ + q_- B_-) B^{-1} \cdot A^{-1} \cdot B^{-1} \boldsymbol{y} \right\|
$$

$$
\geq \{ d(q_+ + \gamma^2 q_-)(q_- + \gamma^2 q_+) \}^{-1} \left[ q_\gamma^2 \left\{ \frac{q_\gamma}{2} R \right\}^2 n \right]^{\frac{1}{2}}
$$

$$
= \{ d(q_+ + \gamma^2 q_-)(q_- + \gamma^2 q_+) \}^{-1} \frac{q_\gamma^2}{2} \sqrt{n} R
$$

$$
\geq \{ d(q_+ + \gamma^2 q_-)(q_- + \gamma^2 q_+) \}^{-1} \frac{q_\gamma^2}{2(1+\lambda)} \sqrt{n} (n \|\boldsymbol{\mu}\|^2 + R). \tag{83}
$$

By Lemma C.7, (67), (83), and the assumption on $\tilde{\varepsilon}$, we have for sufficiently large $n$

$$
K_2 \geq \left\{ \frac{2 q_\gamma}{5(1+\lambda)} - 2(1+\lambda) q_\gamma^{-2} \tilde{\varepsilon} \right\}^2 \frac{n}{(n\|\boldsymbol{\mu}\|^2 + R)^2} \geq \left\{ \frac{q_\gamma}{5(1+\lambda)} \right\}^2 \frac{n^2}{(n\|\boldsymbol{\mu}\|^2 + R)^2}. \tag{84}
$$

By Lemma C.9, (79), (80), (82), and (84), we have

$$
\left[ \left\{ \frac{q_\gamma}{5(1+\lambda)} \right\}^2 - \left\{ \frac{8}{(1-\tilde{\varepsilon}_3^2)(1-\gamma^2)^2 q_+ q_-} + 2 q_\gamma^{-2} \tilde{\varepsilon} \right\}^2 \tilde{\varepsilon} \right] \frac{n}{n\|\boldsymbol{\mu}\|^2 + R}
$$

$$
\leq \|\bar{\boldsymbol{w}}\|^2 \leq (1+\tilde{\varepsilon}) \left\{ \frac{8}{(1-\tilde{\varepsilon}_3^2)(1-\gamma^2)^2 q_+ q_-} + 2 q_\gamma^{-2} \tilde{\varepsilon} \right\}^2 \frac{n}{n\|\boldsymbol{\mu}\|^2 + R}, \tag{85}
$$

where the left hand side is positive for sufficiently large $n$ due to the assumption on $\tilde{\varepsilon}$.

**Regime (iii):** By $n\|\boldsymbol{\mu}\|^2 \geq \lambda R$, $\sqrt{n} \tilde{\varepsilon} \leq \tilde{C} \frac{R}{n\|\boldsymbol{\mu}\|^2} \leq \frac{\tilde{C}}{\lambda}$, (66), and (76), we have

$$
K_1 \leq \left\{ \frac{8}{(1-\tilde{\varepsilon}_3^2)(1-\gamma^2)^2 q_+ q_-} + 2 q_\gamma^{-2} \left( 1 + \frac{n\|\boldsymbol{\mu}\|^2}{R} \right) \tilde{\varepsilon} \right\}^2 \frac{n^2}{(n\|\boldsymbol{\mu}\|^2 + R)^2}
$$

$$
\leq \left\{ \frac{8}{(1-\tilde{\varepsilon}_3^2)(1-\gamma^2)^2 q_+ q_-} + 2 q_\gamma^{-2} \tilde{C} \frac{\lambda^{-1} + 1}{\sqrt{n}} \right\}^2 \frac{n^2}{(n\|\boldsymbol{\mu}\|^2 + R)^2}. \tag{86}
$$

Similarly, $n\|\boldsymbol{\mu}\|^2 \geq \lambda R$, $\sqrt{n} \tilde{\varepsilon} \leq \tilde{C} \frac{R}{n\|\boldsymbol{\mu}\|^2} \leq \frac{\tilde{C}}{\lambda}$, (67), and (78) imply

$$
K_2 \leq \left\{ \frac{8}{(1-\tilde{\varepsilon}_3^2)(1-\gamma^2)^2 q_+ q_-} + \left( 1 + \frac{n\|\boldsymbol{\mu}\|^2}{R} \right) \tilde{\varepsilon} \right\}^2 \frac{n}{(n\|\boldsymbol{\mu}\|^2 + R)^2}
$$

$$
\leq \left\{ \frac{8}{(1-\tilde{\varepsilon}_3^2)(1-\gamma^2)^2 q_+ q_-} + 2 q_\gamma^{-2} \tilde{C} \frac{\lambda^{-1} + 1}{\sqrt{n}} \right\}^2 \frac{n}{(n\|\boldsymbol{\mu}\|^2 + R)^2}. \tag{87}
$$

By (42) and (75), we have

$$
(B^{-1} \boldsymbol{y})^\top A^{-1} B^{-1} (q_+ B_+ + q_- B_-) \boldsymbol{y}
$$

$$
\geq \{ d(q_+ + \gamma^2 q_-)(q_- + \gamma^2 q_+) \}^{-1} q_\gamma (q_\gamma - \gamma) \frac{1 - \tilde{\varepsilon}_3}{2} n(n_+ + n_-) \|\boldsymbol{\mu}\|^2
$$

$$
\geq \{ d(q_+ + \gamma^2 q_-)(q_- + \gamma^2 q_+) \}^{-1} \frac{(1 - \tilde{\varepsilon}_3) q_\gamma (q_\gamma - \gamma)}{2} n^2 \|\boldsymbol{\mu}\|^2. \tag{88}
$$

By Lemma C.7, (66), and (88), we have for sufficiently large $n$

$$
K_1 \geq \left\{ \frac{2(1 - \tilde{\varepsilon}_3)(q_\gamma - \gamma)}{5} \frac{n^2 \|\boldsymbol{\mu}\|^2}{(n\|\boldsymbol{\mu}\|^2 + R)^2} - 2 q_\gamma^{-2} \left( 1 + \frac{n\|\boldsymbol{\mu}\|^2}{R} \right) \tilde{\varepsilon} \frac{n}{n\|\boldsymbol{\mu}\|^2 + R} \right\}^2
$$

$$
\geq \left\{ \frac{2(1 - \tilde{\varepsilon}_3)(q_\gamma - \gamma)}{5(1 + \lambda^{-1})} - 2 q_\gamma^{-2} \tilde{C} \frac{\lambda^{-1} + 1}{\sqrt{n}} \right\}^2 \frac{n^2}{(n\|\boldsymbol{\mu}\|^2 + R)^2} \tag{89}
$$

$$
\geq \left\{ \frac{(1 - \tilde{\varepsilon}_3)(q_\gamma - \gamma)}{5(1 + \lambda^{-1})} \right\}^2 \frac{n^2}{(n\|\boldsymbol{\mu}\|^2 + R)^2}.
$$

Similarly, (42) and (77) imply

$$
\begin{aligned}
&\big\|(q_+B_+ + q_-B_-)B^{-1} \cdot A^{-1} \cdot B^{-1}\boldsymbol{y}\|\big| \\
&\geq \{d(q_+ + \gamma^2 q_-)(q_- + \gamma^2 q_+)\}^{-1}\big[q_\gamma^2(q_\gamma - \gamma)^2 n_+ n_-(n_+ + n_-)\|\boldsymbol{\mu}\|^4\big]^{\frac{1}{2}} \\
&\geq \{d(q_+ + \gamma^2 q_-)(q_- + \gamma^2 q_+)\}^{-1}q_\gamma(q_\gamma - \gamma)\sqrt{\frac{(1-\tilde{\varepsilon}_3^2)}{4}n^3}\|\boldsymbol{\mu}\|^2 \\
&\geq \{d(q_+ + \gamma^2 q_-)(q_- + \gamma^2 q_+)\}^{-1}q_\gamma(q_\gamma - \gamma)\frac{(1-\tilde{\varepsilon}_3)}{2}n^{\frac{3}{2}}\|\boldsymbol{\mu}\|^2
\end{aligned}
\tag{90}
$$

By Lemma C.7, (66), and (90), we have for sufficiently large $n$

$$
\begin{aligned}
K_2 &\geq \left\{\frac{2(1-\tilde{\varepsilon}_3)(q_\gamma - \gamma)}{5}\frac{n^{\frac{3}{2}}\|\boldsymbol{\mu}\|^2}{(n\|\boldsymbol{\mu}\|^2 + R)^2} - 2q_\gamma^{-2}\left(1 + \frac{n\|\boldsymbol{\mu}\|^2}{R}\right)\tilde{\varepsilon}\frac{\sqrt{n}}{n\|\boldsymbol{\mu}\|^2 + R}\right\}^2 \\
&\geq \left\{\frac{2(1-\tilde{\varepsilon}_3)(q_\gamma - \gamma)}{5(1+\lambda^{-1})}\frac{\sqrt{n}}{n\|\boldsymbol{\mu}\|^2 + R)} - 2q_\gamma^{-2}\tilde{C}\frac{\lambda^{-1}+1}{\sqrt{n}}\frac{\sqrt{n}}{n\|\boldsymbol{\mu}\|^2 + R}\right\}^2 \\
&= \left\{\frac{2(1-\tilde{\varepsilon}_3)(q_\gamma - \gamma)}{5(1+\lambda^{-1})} - 2q_\gamma^{-2}\tilde{C}\frac{\lambda^{-1}+1}{\sqrt{n}}\right\}^2\frac{n}{(n\|\boldsymbol{\mu}\|^2 + R)^2} \\
&\geq \left\{\frac{(1-\tilde{\varepsilon}_3)(q_\gamma - \gamma)}{5(1+\lambda^{-1})}\right\}^2\frac{n}{(n\|\boldsymbol{\mu}\|^2 + R)^2}.
\end{aligned}
\tag{91}
$$

By Lemma C.9, (79), (80), (89), and (91), we have

$$
\begin{aligned}
&\left[\left\{\frac{(1-\tilde{\varepsilon}_3)(q_\gamma - \gamma)}{5(1+\lambda^{-1})}\right\}^2 - \left\{\frac{8}{(1-\tilde{\varepsilon}_3^2)(1-\gamma^2)^2 q_+ q_-} + 2q_\gamma^{-2}\tilde{C}\frac{\lambda^{-1}+1}{\sqrt{n}}\right\}^2\tilde{\varepsilon}\right]\frac{n}{n\|\boldsymbol{\mu}\|^2 + R} \\
&\leq \|\bar{\boldsymbol{w}}\|^2 \leq (1+\tilde{\varepsilon})\left\{\frac{8}{(1-\tilde{\varepsilon}_3^2)(1-\gamma^2)^2 q_+ q_-} + 2q_\gamma^{-2}\tilde{C}\frac{\lambda^{-1}+1}{\sqrt{n}}\right\}^2\frac{n}{n\|\boldsymbol{\mu}\|^2 + R},
\end{aligned}
\tag{92}
$$

where the left hand side is positive for sufficiently large $n$.

$\square$

We now turn to obtain bounds of $\langle\bar{\boldsymbol{w}}, \boldsymbol{\mu}\rangle$.

**Lemma C.11.** *Suppose that all the assumptions of Lemma C.8 hold with $\tilde{\varepsilon}_3 \leq \frac{1}{2}$ in regime (ii) and (iii). Further suppose $\tilde{\varepsilon}_2 \leq \tilde{D}\frac{\sqrt{n}\|\boldsymbol{\mu}\|^2}{R}$ in (i) and (ii), where $\tilde{D}$ is a constant which depends only on $\gamma$, $q_+$ (and $\lambda$ under regime (ii)). Then, there exist constants $\hat{c}_2, \tilde{c}_2, N$ which depend only on $\gamma, q_+$ (and $\lambda$ under regime (ii) and (iii)) such that $n \geq N$ imply*

$$
\hat{c}_2 \frac{n\|\boldsymbol{\mu}\|^2}{n\|\boldsymbol{\mu}\|^2 + R} \leq \langle\bar{\boldsymbol{w}}, \boldsymbol{\mu}\rangle \leq \tilde{c}_2\frac{n\|\boldsymbol{\mu}\|^2}{n\|\boldsymbol{\mu}\|^2 + R}.
$$

*Proof.* By Lemma C.6 and C.7, we have

$$
\|(B^{-1}\boldsymbol{y})^\top A^{-1}\| \leq \begin{cases} \gamma^{-1}\dfrac{\sqrt{n}}{R}, & \text{if } q_+q_- = 0, \\ \dfrac{8q_\gamma^{\frac{1}{2}}}{(1-\tilde{\varepsilon}_3^2)(1-\gamma^2)^2 q_+ q_-}\dfrac{\sqrt{n}}{n\|\boldsymbol{\mu}\|^2 + R}, & \text{if } q_+q_- \neq 0. \end{cases}
\tag{93}
$$

The remaining argument is split in cases.

**Regime (i):**

We first note (71) and $n\|\boldsymbol{\mu}\|^2 \le \gamma R$ imply

$$\frac{9}{64(1+\gamma)^3}\frac{n\|\boldsymbol{\mu}\|^2}{R} \le \|\boldsymbol{\mu}\|^2\sqrt{K_1} \le (1+\gamma)^2\left(\gamma^{-2}+2\gamma^{-4}\tilde{\varepsilon}\right)\frac{n\|\boldsymbol{\mu}\|^2}{R}. \tag{94}$$

Furthermore, Lemma C.9 and (93) imply

$$\begin{aligned}
\left|\langle\bar{\boldsymbol{w}},\boldsymbol{\mu}\rangle - \|\boldsymbol{\mu}\|^2\sqrt{K_1}\right| &\le \frac{\gamma^{-1}q_\gamma^{-\frac{1}{2}}}{3}\sqrt{n}\tilde{\varepsilon}_2 + \frac{2q_\gamma^{-2}}{3}\tilde{\varepsilon}\cdot\sqrt{n}\tilde{\varepsilon}_2 \\
&= \frac{\gamma^{-1}q_\gamma^{-\frac{1}{2}} + 2q_\gamma^{-2}\tilde{\varepsilon}}{3}\sqrt{n}\tilde{\varepsilon}_2 \\
&= \frac{\gamma^{-2} + 2\gamma^{-4}\tilde{\varepsilon}}{3}\sqrt{n}\tilde{\varepsilon}_2 \\
&\le \frac{\gamma^{-2} + 2\gamma^{-4}\tilde{\varepsilon}}{3}\tilde{D}\frac{n\|\boldsymbol{\mu}\|^2}{R},
\end{aligned} \tag{95}$$

where the last line is due to the assumption on $\tilde{\varepsilon}_2$.

By (94) and (95), it suffices to choose $\tilde{D}$ so that

$$\frac{\gamma^{-2} + 2\gamma^{-4}\tilde{\varepsilon}}{3}\tilde{D} \le \frac{1}{8(1+\gamma)^3}.$$

Then, we have

$$\begin{aligned}
&\frac{1}{64(1+\gamma)^3}\frac{n\|\boldsymbol{\mu}\|^2}{n\|\boldsymbol{\mu}\|^2+R} \le \frac{1}{64(1+\gamma)^3}\frac{n\|\boldsymbol{\mu}\|^2}{R} \le \langle\bar{\boldsymbol{w}},\boldsymbol{\mu}\rangle \\
&\le \left\{\gamma^{-2}+2\gamma^{-4}\tilde{\varepsilon}+\frac{1}{8(1+\gamma)^3}\right\}\frac{n\|\boldsymbol{\mu}\|^2}{R} \le (1+\gamma)\left\{\gamma^{-2}+2\gamma^{-4}\tilde{\varepsilon}+\frac{1}{8(1+\gamma)^3}\right\}\frac{n\|\boldsymbol{\mu}\|^2}{n\|\boldsymbol{\mu}\|^2+R}.
\end{aligned}$$

**Regime (ii):** We note that under regime (ii), we have $\|\boldsymbol{\mu}\|^2\sqrt{K_1} \approx \frac{n\|\boldsymbol{\mu}\|^2}{n\|\boldsymbol{\mu}\|^2+R}$ by (79) and (82). On the other hand, Lemma C.9 and (93) imply

$$\begin{aligned}
\left|\langle\bar{\boldsymbol{w}},\boldsymbol{\mu}\rangle - \|\boldsymbol{\mu}\|^2\sqrt{K_1}\right| &\le \frac{8}{3(1-\tilde{\varepsilon}_3^2)(1-\gamma^2)^2q_+q_-}\tilde{\varepsilon}_2\frac{\sqrt{n}R}{n\|\boldsymbol{\mu}\|^2+R} + \frac{2q_\gamma^{-2}}{3}\tilde{\varepsilon}\cdot\sqrt{n}\tilde{\varepsilon}_2 \\
&= \frac{8}{3(1-\tilde{\varepsilon}_3^2)(1-\gamma^2)^2q_+q_-}\frac{\tilde{\varepsilon}_2R}{\sqrt{n}\|\boldsymbol{\mu}\|^2}\frac{n\|\boldsymbol{\mu}\|^2}{n\|\boldsymbol{\mu}\|^2+R} + \frac{2q_\gamma^{-2}}{3}\tilde{\varepsilon}\cdot\sqrt{n}\tilde{\varepsilon}_2 \\
&\le \frac{8\tilde{D}}{3(1-\tilde{\varepsilon}_3^2)(1-\gamma^2)^2q_+q_-}\frac{n\|\boldsymbol{\mu}\|^2}{n\|\boldsymbol{\mu}\|^2+R} + + \frac{2q_\gamma^{-2}}{3}\tilde{\varepsilon}\tilde{D}\frac{n\|\boldsymbol{\mu}\|^2}{n\|\boldsymbol{\mu}\|^2+R} \\
&= \left\{\frac{8}{3(1-\tilde{\varepsilon}_3^2)(1-\gamma^2)^2q_+q_-} + \frac{2q_\gamma^{-2}}{3}\tilde{\varepsilon}\right\}\tilde{D}\frac{n\|\boldsymbol{\mu}\|^2}{n\|\boldsymbol{\mu}\|^2+R}.
\end{aligned} \tag{96}$$

Therefore, choosing an appropriate $\tilde{D}$ suffices to establish the desired bounds.

Specifically, under regime (ii), we have by (82)

$$\|\boldsymbol{\mu}\|^2\sqrt{K_1} \ge \frac{q_\gamma}{5(1+\lambda)}\frac{n\|\boldsymbol{\mu}\|^2}{n\|\boldsymbol{\mu}\|^2+R},$$

and hence it suffices to choose $\tilde{D}$ so that

$$\left\{\frac{8}{3(1-\tilde{\varepsilon}_3^2)(1-\gamma^2)^2q_+q_-} + \frac{2q_\gamma^{-2}}{3}\tilde{\varepsilon}\right\}\tilde{D} \le \frac{q_\gamma}{10(1+\lambda)}.$$

**Regime (iii):** Similarly, we note that under regime (iii), we have $\|\boldsymbol{\mu}\|^2\sqrt{K_1} \approx \frac{n\|\boldsymbol{\mu}\|^2}{n\|\boldsymbol{\mu}\|^2+R} \approx 1$ by (86) and (89). On the other hand, Lemma C.9 and (93) imply

$$
\begin{aligned}
&\left|\langle\bar{\boldsymbol{w}},\boldsymbol{\mu}\rangle - \|\boldsymbol{\mu}\|^2\sqrt{K_1}\right|\\
\leq& \frac{8}{3(1-\tilde\varepsilon_3^2)(1-\gamma^2)^2 q_+ q_-}\tilde\varepsilon_2 \frac{\sqrt{n}R}{n\|\boldsymbol{\mu}\|^2+R} + \frac{2q_\gamma^{-2}}{3}\tilde\varepsilon\cdot\sqrt{n}\tilde\varepsilon_2\\
=& \frac{8}{3(1-\tilde\varepsilon_3^2)(1-\gamma^2)^2 q_+ q_-}\frac{\tilde\varepsilon_2 R}{\sqrt{n}\|\boldsymbol{\mu}\|^2}\frac{n\|\boldsymbol{\mu}\|^2}{n\|\boldsymbol{\mu}\|^2+R} + \frac{2q_\gamma^{-2}}{3}\tilde\varepsilon\cdot\sqrt{n}\tilde\varepsilon_2\\
\leq& \frac{8}{3(1-\tilde\varepsilon_3^2)(1-\gamma^2)^2 q_+ q_-}\frac{\tilde\varepsilon R}{n\|\boldsymbol{\mu}\|^2}\frac{n\|\boldsymbol{\mu}\|^2}{n\|\boldsymbol{\mu}\|^2+R} + \frac{2q_\gamma^{-2}}{3}\tilde\varepsilon^2\\
\leq& \frac{8}{3(1-\tilde\varepsilon_3^2)(1-\gamma^2)^2 q_+ q_-}\lambda^{-1}\tilde\varepsilon\frac{n\|\boldsymbol{\mu}\|^2}{n\|\boldsymbol{\mu}\|^2+R} + \frac{2q_\gamma^{-2}}{3}\tilde\varepsilon^2(1+\lambda^{-1})^{-1}\frac{n\|\boldsymbol{\mu}\|^2}{n\|\boldsymbol{\mu}\|^2+R}\\
=& \left\{\frac{8}{3(1-\tilde\varepsilon_3^2)(1-\gamma^2)^2 q_+ q_-}\lambda^{-1} + \frac{2q_\gamma^{-2}}{3}\tilde\varepsilon(1+\lambda^{-1})^{-1}\right\}\tilde\varepsilon\frac{n\|\boldsymbol{\mu}\|^2}{n\|\boldsymbol{\mu}\|^2+R}.
\end{aligned}
\tag{97}
$$

By (86), (89), and (97), we have the desired bound for sufficiently large $n$.

$\square$

**Lemma C.12.** *Consider model (M). If $\boldsymbol{w}\in\mathbb{R}^p$ is such that $\langle\boldsymbol{w},\boldsymbol{\mu}\rangle > 0$, Then, we have*

$$
\mathbb{P}(\langle\boldsymbol{w},y\boldsymbol{x}\rangle \leq 0) \leq \exp\left\{-c\left(\frac{\langle\boldsymbol{w},\boldsymbol{\mu}\rangle}{\|\boldsymbol{w}\|\|\boldsymbol{z}\|_{\psi_2}}\right)^2\right\},
$$

*where $c$ is a universal constant and $\|\boldsymbol{z}\|_{\psi_2}$ is the sub-Gaussian norm of $\boldsymbol{z}$.*

*If $\boldsymbol{z} = \Sigma^{\frac{1}{2}}\boldsymbol{\xi}$ with $\boldsymbol{\xi}\sim\mathcal{N}(0,I_p)$, then we further have*

$$
\kappa\left(\beta_{min}^{-\frac{1}{2}}\frac{\langle\boldsymbol{w},\boldsymbol{\mu}\rangle}{\|\boldsymbol{w}\|}\right) \leq \mathbb{P}(\langle\boldsymbol{w},y\boldsymbol{x}\rangle \leq 0) = \kappa\left(\frac{\langle\boldsymbol{w},\boldsymbol{\mu}\rangle}{\|\Sigma^{\frac{1}{2}}\boldsymbol{w}\|}\right) \leq \kappa\left(\beta_{max}^{-\frac{1}{2}}\frac{\langle\boldsymbol{w},\boldsymbol{\mu}\rangle}{\|\boldsymbol{w}\|}\right),
\tag{98}
$$

*where $\kappa(t) = \mathbb{P}(\xi_1 \geq t)$ and $\beta_{min}$ is the smallest eigenvalue of $\Sigma$ and $\beta_{max} = \|\Sigma\|$.*

*Proof.* Since $y\boldsymbol{x} = \boldsymbol{\mu} + y\boldsymbol{z}$, we have

$$
\begin{aligned}
\mathbb{P}(\langle\boldsymbol{w},y\boldsymbol{x}\rangle \leq 0) &= \mathbb{P}(y\langle\boldsymbol{w},\boldsymbol{z}\rangle \leq -\langle\boldsymbol{w},\boldsymbol{\mu}\rangle)\\
&= \frac{1}{2}\mathbb{P}(\langle\boldsymbol{w},\boldsymbol{z}\rangle \leq -\langle\boldsymbol{w},\boldsymbol{\mu}\rangle) + \frac{1}{2}\mathbb{P}(-\langle\boldsymbol{w},\boldsymbol{z}\rangle \leq -\langle\boldsymbol{w},\boldsymbol{\mu}\rangle)\\
&= \frac{1}{2}\mathbb{P}(|\langle\boldsymbol{w},\boldsymbol{z}\rangle| > \langle\boldsymbol{w},\boldsymbol{\mu}\rangle).
\end{aligned}
$$

Then the desired conclusion follows from the basic property of sub-Gaussian distributions (see Proposition 2.5.2, Vershynin (2018)).

The second result follows since $\langle\boldsymbol{w},\boldsymbol{z}\rangle = \langle\boldsymbol{w},\Sigma^{\frac{1}{2}}\boldsymbol{\xi}\rangle = \langle\Sigma^{\frac{1}{2}}\boldsymbol{w},\boldsymbol{\xi}\rangle \sim \mathcal{N}(0,\boldsymbol{w}^\top\Sigma\boldsymbol{w})$ if $\boldsymbol{\xi}\sim\mathcal{N}(0,\Sigma)$.
$\square$

**Theorem C.13** (Detailed version of Theorem 5.1). *Suppose event $\bigcap_{i=1,2,3}\tilde{E}_i$ holds with one of the following conditions:*

(i) $q_+ q_- = 0$, $n\|\boldsymbol{\mu}\|^2 < \frac{\gamma}{1+(1+\gamma)\tilde\varepsilon_3}R$, and $\sqrt{n}\tilde\varepsilon \leq \frac{\gamma^3}{4}$,

(ii) $q_+ q_- \neq 0$, $n\|\boldsymbol{\mu}\|^2 \leq \frac{q_\gamma}{2\gamma(1-\gamma)}R$, $\sqrt{n}\tilde\varepsilon \leq \frac{q_\gamma^2}{20}$, and $\tilde\varepsilon_3 \leq \frac{1}{2}$,

(iii) $q_\gamma > \gamma$, $n\|\boldsymbol{\mu}\|^2 \geq \frac{q_\gamma}{2\gamma(1-\gamma)}R$, $\tilde\varepsilon_3 \leq \frac{1}{2}$, and $\sqrt{n}\tilde\varepsilon \leq \frac{q_\gamma^2(q_\gamma-\gamma)}{40}\frac{R}{n\|\boldsymbol{\mu}\|^2}$

*Further suppose $\tilde{\varepsilon}_2 \leq \tilde{D}\frac{\sqrt{n}\|\boldsymbol{\mu}\|^2}{R}$ in (i) and (ii), where $\tilde{D}$ is a constant which depends only on $\gamma$, $q_+$.*
*Then, there exist constants $c$ and $N$, which depend on $\gamma$ and $q_+$, such that $n \geq N$ imply*

$$\mathbb{P}_{(\boldsymbol{x},y)}(yf(\boldsymbol{x};\hat{W}) < 0) \leq \exp\left(-c\frac{n\|\boldsymbol{\mu}\|^4}{\|\boldsymbol{z}\|_{\psi_2}^2\{n\|\boldsymbol{\mu}\|^2 + R\}}\right).$$

*If $\boldsymbol{z} = \Sigma^{\frac{1}{2}}\boldsymbol{\xi}$ with $\boldsymbol{\xi} \sim \mathcal{N}(0, I_p)$, then we have instead*

$$\kappa\left\{c^{-1}\beta_{min}^{-\frac{1}{2}}\left(\frac{n\|\boldsymbol{\mu}\|^4}{n\|\boldsymbol{\mu}\|^2 + R}\right)^{\frac{1}{2}}\right\} \leq \mathbb{P}_{(\boldsymbol{x},y)}(yf(\boldsymbol{x};\hat{W}) < 0) \leq \kappa\left\{c\beta_{max}^{-\frac{1}{2}}\left(\frac{n\|\boldsymbol{\mu}\|^4}{n\|\boldsymbol{\mu}\|^2 + R}\right)^{\frac{1}{2}}\right\},$$

*where $\kappa(t) = \mathbb{P}(|\xi_1| \geq t)$ and $\beta_{min}$ is the smallest eigenvalue of $\Sigma$ and $\beta_{max} = \|\Sigma\|$.*

*Proof.* Since we assume $\sqrt{n}\tilde{\varepsilon} \lesssim 1$ in all the regimes, we have $\tilde{\varepsilon} \leq \frac{q_\gamma}{2}$ for sufficiently large $n$.
By Lemma C.10 and C.11 with $\lambda = \frac{q_\gamma}{2\gamma(1-\gamma)}$, we have

$$\frac{\hat{c}_2^2}{\tilde{c}_1}\frac{n\|\boldsymbol{\mu}\|^4}{n\|\boldsymbol{\mu}\|^2 + R} \leq \left(\frac{\langle\bar{\boldsymbol{w}},\boldsymbol{\mu}\rangle}{\|\bar{\boldsymbol{w}}\|}\right)^2 \leq \frac{\tilde{c}_2^2}{\hat{c}_1}\frac{n\|\boldsymbol{\mu}\|^4}{n\|\boldsymbol{\mu}\|^2 + R}.$$

Then, the desired conclusion holds from Lemma C.12. $\qquad\square$

## D   PROOF OF THEOREM 6.2 AND THEOREM 6.3

First we establish connection between events $E(\theta_1, \theta_2)$ and $\tilde{E}_1(\tilde{\varepsilon}_1, R)$, $\tilde{E}_2(\varepsilon_2, R)$.

**Lemma D.1.** *Suppose $\tilde{E}_1(\tilde{\varepsilon}_1, R) \cap \tilde{E}_2(\varepsilon_2, R)$ hold with $\tilde{\varepsilon}_1 \leq \frac{1}{2}$, then event $E(\theta_1, \theta_2)$ holds with*

$$\theta_1 = \frac{\tilde{\varepsilon}_1}{2} \quad and \quad \theta_2 = \frac{\tilde{\varepsilon}_2\sqrt{R}}{2\|\boldsymbol{\mu}\|}.$$

*Proof.* By $\tilde{E}_1(\tilde{\varepsilon}_1, R)$, we have $|\|\boldsymbol{z}_i\|^2 - R| \leq \frac{\tilde{\varepsilon}_1}{3}R$ and $|\langle\boldsymbol{z}_i, \boldsymbol{z}_k\rangle| \leq \frac{\tilde{\varepsilon}_1}{3}R$ for any $i \neq k$. Then, we have

$$\left|\left\langle\frac{\boldsymbol{z}_i}{\|\boldsymbol{z}_i\|}, \frac{\boldsymbol{z}_k}{\|\boldsymbol{z}_k\|}\right\rangle\right| \leq \frac{\frac{\tilde{\varepsilon}_1}{3}R}{(1-\frac{\tilde{\varepsilon}_1}{3})R}$$
$$\leq \frac{\tilde{\varepsilon}_1}{2},$$

where the last line follows from the assumption $\tilde{\varepsilon}_1 \leq \frac{1}{2}$.

Similarly, by $\tilde{E}_1(\tilde{\varepsilon}_1, R) \cap \tilde{E}_2(\varepsilon_2, R)$ , we have

$$\left|\left\langle\frac{\boldsymbol{z}_i}{\|\boldsymbol{z}_i\|}, \frac{\boldsymbol{\mu}}{\|\boldsymbol{\mu}\|}\right\rangle\right| \leq \frac{\frac{\tilde{\varepsilon}_2}{3}R}{\sqrt{(1-\frac{\tilde{\varepsilon}_1}{3})R}\|\boldsymbol{\mu}\|} \leq \frac{\tilde{\varepsilon}_2\sqrt{R}}{2\|\boldsymbol{\mu}\|},$$

where the last inequality is due to $\tilde{\varepsilon}_1 \leq \frac{1}{2}$. $\qquad\square$

### D.1   PROOF OR THEOREM 6.2

For model (sG), we have

**Lemma D.2** (Lemma S4.1, S4.2, Hashimoto et al. (2025)). *Consider model (sG). There exist universal constants $C_i$, $i = 1, 2$ such that for any $\delta \in (0, \frac{1}{2})$, event $\tilde{E}_1 \cap \tilde{E}_2$ holds with probability at least $1 - 2\delta$, where*

$$R = \mathrm{tr}(\Sigma),$$
$$\tilde{\varepsilon}_1 := C_1 \max\left\{\sqrt{n\log\left(\frac{1}{\delta}\right)}\|\Sigma\|_F, n\log\left(\frac{1}{\delta}\right)\|\Sigma\|\right\}/\mathrm{tr}(\Sigma)$$
$$\tilde{\varepsilon}_2 := C_2\sqrt{n\log\left(\frac{1}{\delta}\right)}\|\Sigma^{\frac{1}{2}}\boldsymbol{\mu}\|/\mathrm{tr}(\Sigma).$$

**Lemma D.3.** *There exists a universal constant $C_3$ such that we have with probability at least $1 - \delta$*

$$|n_+ - n_-| \leq C_3 \sqrt{n \log(\tfrac{1}{\delta})}.$$

*Proof.* By independence of $y_i$ and $y_i \in \{\pm 1\}$, Hoeffding's inequality implies that

$$\left| \sum_i y_i \right| \leq C_3 \sqrt{n \log(\tfrac{1}{\delta})}$$

holds with probability at least $1 - \delta$, where $C_3$ is a universal constant.

The desired result follows by noting that $\sum_i y_i = n_+ - n_-$. $\qquad\square$

**Corollary D.4.** *Event $\tilde{E}_3$ holds with $\tilde{\varepsilon}_3 := C_3 \sqrt{\frac{\log(\frac{1}{\delta})}{n}}$ with probability at least $1 - \delta$.*

With these, we can rewrite assumptions of our main theorems, namely Theorem 4.8 and C.13, using parameters defining the model (sG).

From Theorem 4.8, we have the following result on (sG).

**Theorem D.5.** *Consider model (sG). Suppose*

$$\mathrm{tr}(\Sigma) \geq \max \left\{ 2C_1 \sqrt{n \log(\tfrac{1}{\delta})} \|\Sigma\|_F, 2C_1 n \log(\tfrac{1}{\delta}) \|\Sigma\|, C_2^2 n \log(\tfrac{1}{\delta}) \frac{\|\Sigma^{\frac{1}{2}} \boldsymbol{\mu}\|^2}{\|\boldsymbol{\mu}\|^2} \right\}, \qquad (99)$$

*and further suppose one of the following conditions:*

*(i)* $\|\boldsymbol{\mu}\|^2 \geq 2.18 \max \left\{ C_1 \sqrt{n \log(\tfrac{1}{\delta})} \|\Sigma\|_F, C_1 n \log(\tfrac{1}{\delta}) \|\Sigma\|, C_2 \sqrt{n \log(\tfrac{1}{\delta})} \|\Sigma^{\frac{1}{2}} \boldsymbol{\mu}\| \right\}$,
$\alpha \leq \left[ 1.2\{n\|\boldsymbol{\mu}\|^2 + \mathrm{tr}(\Sigma)\} \right]^{-1}$, *and* $\sigma \leq 0.2\gamma\alpha \sqrt{\frac{\|\boldsymbol{\mu}\|^2 + \mathrm{tr}(\Sigma)}{m}}$,

*(ii)* $\mathrm{tr}(\Sigma) \geq \dfrac{67.2e}{\gamma^2} n\|\boldsymbol{\mu}\|^2$,
$\mathrm{tr}(\Sigma) \geq \dfrac{453e}{\gamma^3} n \max \left\{ C_1 \sqrt{n \log(\tfrac{1}{\delta})} \|\Sigma\|_F, C_1 n \log(\tfrac{1}{\delta}) \|\Sigma\|, C_2 \sqrt{n \log(\tfrac{1}{\delta})} \|\Sigma^{\frac{1}{2}} \boldsymbol{\mu}\| \right\}$,
$\alpha \leq \{8\mathrm{tr}(\Sigma)\}^{-1}$, *and* $\sigma \leq 0.1\gamma\alpha \sqrt{\frac{\mathrm{tr}(\Sigma)}{m}}$.

*Then, we have with probability at least $1 - 2\delta$, the gradient descent iterate (2) keeps all the neurons activated for $t \geq 1$, satisfies $\mathcal{L}(W^{(t)}) = O(t^{-1})$, and converges in the direction given in Theorem 4.8.*

*Proof.* By Lemma D.1, D.2, and the assumption (99), all the conclusions of Lemma D.1 and D.2 hold with probability at least $1 - 2\delta$ with

$$R = \mathrm{tr}(\Sigma),$$

$$\theta_1 = \frac{\tilde{\varepsilon}_1}{2} = \frac{C_1 \max \left\{ \sqrt{n \log(\frac{1}{\delta})} \|\Sigma\|_F, n \log(\frac{1}{\delta}) \|\Sigma\| \right\}}{2\mathrm{tr}(\Sigma)} \leq \frac{1}{4},$$

$$\theta_2 = \frac{\tilde{\varepsilon}_2 \sqrt{R}}{2\|\boldsymbol{\mu}\|} = \frac{C_2 \sqrt{n \log(\frac{1}{\delta})} \|\Sigma^{\frac{1}{2}} \boldsymbol{\mu}\|}{2\sqrt{\mathrm{tr}(\Sigma)} \|\boldsymbol{\mu}\|} \leq \frac{1}{2}.$$

Then, we have $0.91\sqrt{\mathrm{tr}(\Sigma)} \leq R_{min}(\boldsymbol{z}) \leq R_{max}(\boldsymbol{z}) \leq 1.09\sqrt{\mathrm{tr}(\Sigma)}$.

From these, we have

$$
\begin{aligned}
& 2\theta_2 \|\boldsymbol{\mu}\| R_{max}(\boldsymbol{z}) + \theta_1 R_{max}^2(\boldsymbol{z}) \\
\leq\ & 1.09 C_2 \sqrt{n \log(\tfrac{1}{\delta})} \|\Sigma^{\frac{1}{2}} \boldsymbol{\mu}\| \\
& \quad + 0.6 C_1 \max\left\{ \sqrt{n \log(\tfrac{1}{\delta})} \|\Sigma\|_F, n \log(\tfrac{1}{\delta}) \|\Sigma\| \right\} \\
\leq\ & 2.18 \max\{ C_1 \sqrt{n \log(\tfrac{1}{\delta})} \|\Sigma\|_F, C_1 n \log(\tfrac{1}{\delta}) \|\Sigma\|, C_2 \sqrt{n \log(\tfrac{1}{\delta})} \|\Sigma^{\frac{1}{2}} \boldsymbol{\mu}\| \}.
\end{aligned}
\tag{100}
$$

We first show the desired conclusions under condition (i).

First, it is immediate from (100) to see $\|\boldsymbol{\mu}\|^2 \geq 2\theta_2 \|\boldsymbol{\mu}\| R_{max}(\boldsymbol{z}) + \theta_1 R_{max}^2(\boldsymbol{z})$ holds under condition (i).

Furthermore, condition (i) also implies that Assumption 4.3 is satisfied since

$$
\alpha\{n\|\boldsymbol{\mu}\|^2 + R_{max}^2(\boldsymbol{z})\} < 1.2\alpha\{n\|\boldsymbol{\mu}\|^2 + \mathrm{tr}(\Sigma)\} \leq 1.
$$

We note that $\exp(\rho) < 1.26$ holds under condition (i) since

$$
\begin{aligned}
\rho &\leq \sigma \sqrt{m \cdot \frac{3}{2} \cdot 1.2\{\|\boldsymbol{\mu}\|^2 + \mathrm{tr}(\Sigma)\}} \\
&\leq 1.35 \sigma \sqrt{m\{\|\boldsymbol{\mu}\|^2 + \mathrm{tr}(\Sigma)\}} \\
&\leq 0.27 \gamma \alpha\{\|\boldsymbol{\mu}\|^2 + \mathrm{tr}(\Sigma)\} \\
&\leq \frac{0.27}{1.2} \gamma \leq 0.225.
\end{aligned}
$$

where the first inequality follows from the definition of $\rho$ and $\theta_2 \leq \frac{1}{2}$ and $R_{max}(\boldsymbol{z}) \leq 1.2\sqrt{\mathrm{tr}(\Sigma)}$, the remaining inequalities follow from condition (i).

Noting $\exp(\rho) < 1.26$ and $\rho \leq 0.27\gamma\alpha\{\|\boldsymbol{\mu}\|^2 + \mathrm{tr}(\Sigma)\}$, we also see that Assumption 4.2 is satisfied since

$$
\begin{aligned}
\rho \exp(\rho) &< 1.26\rho \\
&\leq 1.26 \cdot 0.27 \gamma \alpha\{\|\boldsymbol{\mu}\|^2 + \mathrm{tr}(\Sigma)\} \\
&\leq \frac{1.26 \cdot 0.27}{0.8} \gamma \alpha\{\|\boldsymbol{\mu}\|^2 + R_{min}^2(\boldsymbol{z})\} \\
&\leq \gamma(1 - \theta_2)\alpha\{\|\boldsymbol{\mu}\|^2 + R_{min}^2(\boldsymbol{z})\}.
\end{aligned}
$$

Since all the assumptions of (i) of Theorem 4.8 are satisfied, the desired conclusions hold with probability at least $1 - 2\delta$ under condition (i).

Next we show the desired conclusions under condition (ii). To see this, first note that $\tilde{\varepsilon}_1 \leq \frac{1}{2}$ implies that $\frac{R_{max}^2(\boldsymbol{z})}{R_{min}^2(\boldsymbol{z})} \leq \frac{1 + \frac{\tilde{\varepsilon}_1}{3}}{1 - \frac{\tilde{\varepsilon}_1}{3}} \leq \frac{7}{5}$. So, we have $C_w$ in Theorem 4.8 with $C_w \leq \frac{168e}{5\gamma^2}$. So, by setting $\varepsilon_i = \frac{5\gamma^2}{368e}, i = 1,3$, $C_w \varepsilon_i \leq \frac{1}{2}$ is ensured. Moreover, by setting $\varepsilon_2 = \frac{1}{2}$, $\varepsilon_1 + \varepsilon_2 + \varepsilon_3 < 1$ is ensured.

We note that $\exp(\rho) < \frac{10}{9}$ holds under condition (ii) since

$$
\begin{aligned}
\rho &\leq 1.35 \sigma \sqrt{m\{\|\boldsymbol{\mu}\|^2 + \mathrm{tr}(\Sigma)\}} \\
&\leq 1.35 \sigma \sqrt{m(1.0055)\mathrm{tr}(\Sigma)} \\
&\leq 1.36 \sigma \sqrt{m\,\mathrm{tr}(\Sigma)} \\
&\leq 1.36 \cdot 0.1 \gamma \alpha \mathrm{tr}(\Sigma) \\
&\leq \frac{1.36 \cdot 0.1}{8} \gamma \leq 0.017 < \log(\tfrac{10}{9}).
\end{aligned}
$$

By noting $\rho \le 0.136\gamma\alpha\text{tr}(\Sigma)$ and $\exp(\rho) < \frac{10}{9}$, condition (ii) implies that Assumption 4.5 is satisfied since

$$
\begin{aligned}
\rho\exp(\rho) &\le \tfrac{10}{9} \cdot 0.136\gamma\alpha\text{tr}(\Sigma) \\
&\le \tfrac{10}{9} \cdot 0.136 \cdot 0.8^{-1}\gamma\alpha R_{min}^2(\boldsymbol{z}) \\
&\le 0.19\gamma\alpha R_{min}^2(\boldsymbol{z}) \\
&\le \varepsilon_2\gamma(1-\theta_2)\alpha R_{min}^2(\boldsymbol{z}).
\end{aligned}
$$

Similarly, by noting $\exp(\rho) < \frac{10}{9}$, condition (ii) also implies that Assumption 4.7 is satisfied since

$$
\begin{aligned}
6\alpha R_{min}^2(\boldsymbol{z})\exp(\rho) &< 7.2 \cdot \tfrac{10}{9}\alpha R_{min}^2(\boldsymbol{z}) \\
&= 8\alpha\text{tr}(\Sigma) \le 1.
\end{aligned}
$$

By (100), condition (ii) implies that Assumption 4.4 is satisfied since

$$
\begin{aligned}
&2\theta_2\|\boldsymbol{\mu}\|R_{max}(\boldsymbol{z}) + \theta_1 R_{max}^2(\boldsymbol{z}) \\
\le\,& 2.18\max\{C_1\sqrt{n\log(\tfrac{1}{\delta})}\|\Sigma\|_F, C_1 n\log(\tfrac{1}{\delta})\|\Sigma\|, C_2\sqrt{n\log(\tfrac{1}{\delta})}\|\Sigma^{\frac{1}{2}}\boldsymbol{\mu}\|\} \\
\le\,& \frac{\gamma^3\text{tr}(\Sigma)}{453en} \\
\le\,& \frac{\frac{5\gamma^2}{336e} \cdot \frac{1}{2} \cdot 0.8\gamma\text{tr}(\Sigma)}{n(10/9)^2} \\
\le\,& \frac{\varepsilon_1\gamma(1-\theta_2)R_{min}^2(\boldsymbol{z})}{n\exp(2\rho)}.
\end{aligned}
$$

Since all the assumptions of (ii) of Theorem 4.8 are satisfied, the desired conclusions hold with with probability at least $1 - 2\delta$ under condition (ii). $\qquad\square$

Application of Theorem C.13 on the model (sG) is given below:

**Theorem D.6.** *Consider model (sG). Suppose $n \ge \max\left\{4C_3^2\log(\tfrac{1}{\delta}), N(\gamma, q_+)\right\}$ and one of the following holds:*

*(i) $q_+ q_- = 0$, $n\|\boldsymbol{\mu}\|^2 < \frac{\gamma}{1+(1+\gamma)/2}\text{tr}(\Sigma)$,*

$$
\text{tr}(\Sigma) \ge 4\gamma^{-3}n\sqrt{\log(\tfrac{1}{\delta})}\max\left\{C_1\|\Sigma\|_F, C_1\|\Sigma\|\sqrt{n\log(\tfrac{1}{\delta})}, C_2\sqrt{n}\|\Sigma^{\frac{1}{2}}\boldsymbol{\mu}\|\right\}, \text{and}
$$

$$
\|\boldsymbol{\mu}\|^2 \ge \tilde{D}^{-1}C_2\|\Sigma^{\frac{1}{2}}\boldsymbol{\mu}\|\sqrt{\log(\tfrac{1}{\delta})},
$$

*(ii) $q_+ q_- \ne 0$, $n\|\boldsymbol{\mu}\|^2 \le \frac{q_\gamma}{2\gamma(1-\gamma)}\text{tr}(\Sigma)$,*

$$
\text{tr}(\Sigma) \ge 20q_\gamma^{-2}n\sqrt{\log(\tfrac{1}{\delta})}\max\left\{C_1\|\Sigma\|_F, C_1\|\Sigma\|\sqrt{n\log(\tfrac{1}{\delta})}, C_2\sqrt{n}\|\Sigma^{\frac{1}{2}}\boldsymbol{\mu}\|\right\}, \text{and}
$$

$$
\|\boldsymbol{\mu}\|^2 \ge \tilde{D}^{-1}C_2\|\Sigma^{\frac{1}{2}}\boldsymbol{\mu}\|\sqrt{\log(\tfrac{1}{\delta})},
$$

*(iii) $q_\gamma > \gamma$, $n\|\boldsymbol{\mu}\|^2 \ge \frac{q_\gamma}{2\gamma(1-\gamma)}\text{tr}(\Sigma)$,*

$$
\text{tr}(\Sigma) \ge \frac{2\sqrt{10}}{q_\gamma\sqrt{q_\gamma - \gamma}}n\|\boldsymbol{\mu}\|\{\log(\tfrac{1}{\delta})\}^{1/4}
$$

$$
\times \sqrt{\max\left\{C_1\|\Sigma\|_F, C_1\|\Sigma\|\sqrt{n\log(\tfrac{1}{\delta})}, C_2\sqrt{n}\|\Sigma^{\frac{1}{2}}\boldsymbol{\mu}\|\right\}}.
$$

*Then, we have with probability at least $1 - 3\delta$*

$$\mathbb{P}_{(\boldsymbol{x},y)}(yf(\boldsymbol{x};\hat{W}) < 0) \le \exp\left(-c\frac{n\|\boldsymbol{\mu}\|^4}{L^2\|\Sigma\|\{n\|\boldsymbol{\mu}\|^2 + \operatorname{tr}(\Sigma)\}}\right),$$

*where $c$ is a constant which depends only on $\gamma$ and $q_+$.*

*Proof.* By Lemma D.2 and Corollary D.4, event $\bigcap_{i=1,2,3}\tilde{E}_i$ holds with probability at least $1 - 3\delta$ with

$$R = \operatorname{tr}(\Sigma),$$

$$\tilde{\varepsilon}_1 = C_1 \max\left\{\sqrt{n\log\left(\frac{1}{\delta}\right)}\|\Sigma\|_F, n\log\left(\frac{1}{\delta}\right)\|\Sigma\|\right\}/\operatorname{tr}(\Sigma)$$

$$\tilde{\varepsilon}_2 = C_2\sqrt{n\log\left(\frac{1}{\delta}\right)}\|\Sigma^{\frac{1}{2}}\boldsymbol{\mu}\|/\operatorname{tr}(\Sigma),$$

$$\tilde{\varepsilon}_3 = C_3\sqrt{\frac{\log(\frac{1}{\delta})}{n}}.$$

Note that the assumption $n \ge 4C_3^2\log(\frac{1}{\delta})$ implies $\tilde{\varepsilon}_3 \le \frac{1}{2}$.

Then, the desired conclusion follows by simply rewriting assumptions of Theorem C.13. $\square$

We can now see that Theorem 6.2 follow from Theorem D.5 and D.6 by ensuring both assumptions are satisfied. Specifically, Theorem D.5 under condition (i) and Theorem D.6 under condition (i) and (ii) imply Theorem 6.2 under (a)-(i). Similarly, Theorem D.5 under condition (ii) and Theorem D.6 under condition (i) and (ii) imply Theorem 6.2 under condition (a)-(ii). Lastly, Theorem D.5 under condition (i) and Theorem D.6 under condition (iii) imply Theorem 6.2 under condition (b).

## D.2 Proof of Theorem 6.3

For model (PM), we have instead of Lemma D.2

**Lemma D.7** (Lemma S3.4, S3.5, Hashimoto et al. (2025)). *Consider model (PM). There exist a constant $C(r, K)$ such that for any $\delta \in (0, \frac{1}{2})$, event $\tilde{E}_1 \cap \tilde{E}_2$ holds with probability at least $1 - 2\delta$, where*

$$R = \operatorname{tr}(\Sigma),$$

$$\tilde{\varepsilon}_1 := C(r, K)\delta^{-\frac{2}{r}}\max\left\{n^{\frac{2}{r}}p^{\frac{2}{r}-\frac{1}{2}}, n^{\frac{4}{r}}\right\}\frac{\|\Sigma\|_F}{\operatorname{tr}(\Sigma)},$$

$$\tilde{\varepsilon}_2 := \sqrt{\frac{n}{\delta}}\frac{\|\Sigma^{\frac{1}{2}}\boldsymbol{\mu}\|}{\operatorname{tr}(\Sigma)}.$$

Similar arguments using D.7 instead of D.2 allows us to establish results analogous to Theorem D.5 and D.6.

**Theorem D.8.** *Consider model (PM). Let $\delta \in (0, \frac{1}{2})$. Suppose, for sufficiently large constant $C$ which depends on $\delta, \gamma, q_+, r, K$,*

$$\operatorname{tr}(\Sigma) \ge C\max\left\{n^{\frac{2}{r}}p^{\frac{2}{r}-\frac{1}{2}}\|\Sigma\|_F, n^{\frac{4}{r}}\|\Sigma\|_F, n\frac{\|\Sigma^{\frac{1}{2}}\boldsymbol{\mu}\|^2}{\|\boldsymbol{\mu}\|^2}\right\} \tag{101}$$

*and further suppose one of the following conditions:*

*(i)* $\|\boldsymbol{\mu}\|^2 \ge C\max\left\{n^{\frac{2}{r}}p^{\frac{2}{r}-\frac{1}{2}}\|\Sigma\|_F, n^{\frac{4}{r}}\|\Sigma\|_F, \sqrt{n}\|\Sigma^{\frac{1}{2}}\boldsymbol{\mu}\|\right\},$
$\alpha \le \left[1.2\{n\|\boldsymbol{\mu}\|^2 + \operatorname{tr}(\Sigma)\}\right]^{-1},$ *and* $\sigma \le 0.2\gamma\alpha\sqrt{\frac{\|\boldsymbol{\mu}\|^2 + \operatorname{tr}(\Sigma)}{m}},$

*(ii)* $\operatorname{tr}(\Sigma) \ge C\max\left\{n\|\boldsymbol{\mu}\|^2, n^{\frac{2}{r}+1}p^{\frac{2}{r}-\frac{1}{2}}\|\Sigma\|_F, n^{\frac{4}{r}+1}\|\Sigma\|_F, n^{\frac{3}{2}}\|\Sigma^{\frac{1}{2}}\boldsymbol{\mu}\|\right\},$
$\alpha \le \{8\operatorname{tr}(\Sigma)\}^{-1},$ *and* $\sigma \le 0.1\gamma\alpha\sqrt{\frac{\operatorname{tr}(\Sigma)}{m}}.$

*Then, we have with probability at least $1 - 2\delta$, the gradient descent iterate (2) keeps all the neurons activated for $t \geq 1$, satisfies $\mathcal{L}(W^{(t)}) = O(t^{-1})$, and converges in the direction given in Theorem 4.8.*

*Proof.* The theorem follows from Theorem 4.8, Lemma D.1, and Lemma D.7 by repeating the same argument from the proof of Theorem D.5 by replacing Lemma D.2 with Lemma D.7. $\square$

Similarly, we obtain a result analogous to Theorem D.6.

**Theorem D.9.** *Consider model (PM). Let $\delta \in (0, \frac{1}{3})$. Suppose $n \geq \max\left\{4C_3^2 \log(\frac{1}{\delta}), N(\gamma, q_+)\right\}$ and one of the following holds for sufficiently large constant $C$ which depends on $\delta, \gamma, q_+, r, K$:*

*(i)* $\mathrm{tr}(\Sigma) \geq C \max\{n\|\boldsymbol{\mu}\|^2, n^{\frac{2}{r}+\frac{1}{2}} p^{\frac{2}{r}-\frac{1}{2}} \|\Sigma\|_F, n^{\frac{4}{r}+\frac{1}{2}} \|\Sigma\|_F, n^{\frac{3}{2}} \|\Sigma^{\frac{1}{2}}\boldsymbol{\mu}\|\}$ *and*

$\|\boldsymbol{\mu}\|^2 \geq C\|\Sigma^{\frac{1}{2}}\boldsymbol{\mu}\|,$

*(ii)* $q_\gamma > \gamma$, $n\|\boldsymbol{\mu}\|^2 \geq \frac{q_\gamma}{2\gamma(1-\gamma)}\mathrm{tr}(\Sigma)$, *and*

$$\mathrm{tr}(\Sigma) \geq C\|\boldsymbol{\mu}\| \max\left\{n^{\frac{1}{r}+\frac{3}{4}} p^{\frac{1}{r}-\frac{1}{4}} \|\Sigma\|_F^{\frac{1}{2}}, n^{\frac{2}{r}+\frac{3}{4}} \|\Sigma\|_F^{\frac{1}{2}}, n^{\frac{5}{4}} \|\Sigma^{\frac{1}{2}}\boldsymbol{\mu}\|^{\frac{1}{2}}\right\}.$$

*Then, we have with probability at least $1 - 3\delta$*

$$\mathbb{P}_{(\boldsymbol{x},y)}(yf(\boldsymbol{x};\hat{W}) < 0) \leq \exp\left(-c\frac{n\|\boldsymbol{\mu}\|^4}{L^2\|\Sigma\|\{n\|\boldsymbol{\mu}\|^2 + \mathrm{tr}(\Sigma)\}}\right),$$

*where $c$ is a constant which depends only on $\gamma$ and $q_+$.*

*Proof.* The theorem follows from Theorem C.13, Lemma D.1, Lemma D.7, and Lemma D.4 by repeating the same argument from the proof of Theorem D.6 by replacing Lemma D.2 with Lemma D.7. $\square$

Putting these together with $\Sigma = I_p$, we obtain Theorem 6.3. Specifically, Theorem 6.3 under condition (i) follows from Theorem D.8 under (i) and Theorem D.9 under (i), Theorem 6.3 under condition (ii) follows from Theorem D.8 under (i) and Theorem D.9 under (ii), and Theorem 6.3 under condition (iii) follows from Theorem D.8 under (ii) and Theorem D.9 under (i).

