# OpenReview forum: "Directional Convergence, Benign Overfitting of Gradient Descent in leaky ReLU two-layer Neural Networks"
_ICLR.cc/2026/Conference — ICLR 2026 Poster_

### Official Review · Reviewer_i5vN · 2025-10-23

**Soundness:** 2
**Presentation:** 3
**Contribution:** 3
**Rating:** 4
**Confidence:** 4

**Summary:**

This work proves directional convergence of two-layer neural networks with leaky ReLU activation trained on the linearly separable data without label flipping via gradient descent. The data distribution are either mixture sub-Gaussian or mixture of polynomially tailed distribution.

**Strengths:**

The derivation is very solid and the error bounds look correct. The writing is clear and easy to follow.

**Weaknesses:**

1. The studied data distribution is restricted to two linearly separable mixture models.

2. Previous work on benign overfitting need $d >> n$ or near-orthogonality because of the existence of the flipped labels. It's intuitive and very expected that near-orthogonality is no longer needed if there is no flipped labels.

3. Theorem 4.8 states the convergent direction is the solution of an optimization problem (5), but does not solve (5).

4. In your Assumption 6.1 (a), in the special case where $\Sigma = I_d$, it reduces to $d \geq C \max (n \|\mu\|^2, n^2)$, which is the same as Assumption (CL2) in [1], why you do not have near-orthogonality under such assumption?

5. typos: The latex does not compile correctly at Assumption 6.1 (a)

[1] Benign Overfitting in Linear Classifiers and Leaky ReLU Networks from KKT Conditions for Margin Maximization

**Questions:**

What's your definition of benign overfitting? Why there exists benign overfitting if all labels are correct/unflipped?

---

> ### Author Response · Authors · 2025-11-22
>
> We sincerely thank the reviewer for the time and effort devoted to evaluating our work. We appreciate the thoughtful comments, and we are grateful for the opportunity to clarify several points regarding our contribution. All line numbers below are from original submission.
>
> We would first like to emphasize that benign overfitting has been studied along multiple lines, including both with and without label-flipping noise.
>
> For example, Cao et al (2021) and [Minsker et al. (2021)](https://arxiv.org/abs/2111.07041) studied binary linear classification without label-flipping noise. Chatterji and Long (2021), Wang and Thrampoulidis (2022), Hashimoto et al. (2025) and [Tsigler et al. (2025)](https://arxiv.org/abs/2503.07966]) studied benign overfitting in binary linear classification both with and without label-flipping noise. Notably, Hashimoto et al. (2025) showed that generalization behavior differs in the strong signal regime depending on whether label-flipping noise is present, while the behavior is similar in the weak-signal (nearly-orthogonal) regime. In addition, Frei et al. (2022) and Xu and Gu (2023) studied benign overfitting in two-layer neural networks both with and without label-flipping noise.
>
> We also emphasize that the mixture model considered here—as well as in the above works—naturally includes a second source of noise, represented by the random vector $\boldsymbol{z}$. For intuition, consider the isotropic mixture $x = y\boldsymbol{\mu} + \boldsymbol{z}$ with $\boldsymbol{z}\sim \mathcal{N}(0, I_p)$. Since $\|\boldsymbol{z}\|$ is typically of order $\sqrt{p}$, the two class-conditional distributions overlap significantly unless $\|\boldsymbol{\mu}\|$ is substantially larger than $\sqrt{p}$. Thus, even without label-flipping noise, overfitting phenomena remain non-trivial.
>
> **Responses to questions:**
>
> *1. The studied data distribution is restricted to two linearly separable mixture models.*
> - We do not assume linear separability a priori. Rather, we show that under model (sG) or (PM), linear separability arises with high probability due to over-parameterization. Even in the isotropic Gaussian mixture, such separability does not occur with high probability when $n < p$; it emerges specifically if $p > n$.
>
> *2. Previous work on benign overfitting need $d\gg n$ or near-orthogonality because of the existence of the flipped labels. It's intuitive and very expected that near-orthogonality is no longer needed if there is no flipped labels.*
> - As noted earlier, even without label flips, overlap between the two mixture components persists unless the signal is substantially strong. Thus, the absence of label-flipping noise does not automatically eliminate the need for near-orthogonality.
> - Prior work required $p \gg n$ primarily to control concentration properties of $\boldsymbol{z}_i$ (e.g. $\|\boldsymbol{z}_i\|$, $\langle \boldsymbol{z}_i, \boldsymbol{z}_j \rangle$), which is standard in benign overfitting proofs (including above references).
> - Our test-error bounds (Theorems 5.1 and 6.2) reveal a phase transition between weak-signal and strong-signal regimes, occurring around $\|\boldsymbol{\mu}\|\approx \sqrt{\frac{p}{n}}$ in the isotropic case. This transition—rather than the presence or absence of label flips—explains when near-orthogonality becomes necessary.
>
> *3. Theorem 4.8 states the convergent direction is the solution of an optimization problem (5), but does not solve (5).*
> - We do in fact characterize the solution. Optimization problem (5) corresponds to the maximum-margin linear classifier, which generally lacks a closed-form formula. However, in the over-parameterized regime, the closed-form solution
> $$\boldsymbol{\hat w} = \tilde X^\top (\tilde X \tilde X^\top)^{-1}\boldsymbol{y}$$
> is obtained, as noted in the proof sketch (line 448). This gives an explicit representation of the convergent direction.
>
> *4. In your Assumption 6.1 (a), in the special case where $\Sigma = I_d$, it reduces to $d \geq C\max\{n\|\boldsymbol{\mu}\|^2, n^2\}$, which is the same as Assumption (CL2) in [1], why you do not have near-orthogonality under such assumption?*
> - As stated in lines 361–362, Assumption 6.1(a) corresponds to the weak-signal (nearly-orthogonal) regime, while Assumption 6.1(b) corresponds to the strong-signal regime. Thus the distinction is explicit in our formulation: near-orthogonality arises in regime (a) but not in (b).
>
> *5. What's your definition of benign overfitting? Why there exists benign overfitting if all labels are correct/unflipped?*
> - We say benign overfitting holds when arbitrarily small test error is achieved while the training data is perfectly classified despite the presence of noise (here arising from $\boldsymbol{z}$). We have added this definition at the end of Section 3. Even with correct labels, the feature-level noise can make the problem sufficiently challenging that overfitting is still meaningful.

---

### Official Review · Reviewer_RXkj · 2025-10-31

**Soundness:** 3
**Presentation:** 3
**Contribution:** 3
**Rating:** 8
**Confidence:** 3

**Summary:**

The paper studies two-layer leaky ReLU networks trained by gradient descent in the context of binary classification using an exponentially tailed loss. It establishes two main theoretical results. First, it proves that gradient descent, with a fixed step size, exhibits convergence in direction for the network weights, approaching the max-margin classifier. Second, it characterizes the generalization behavior of such networks under a mixtures data model with mean $\mu$ and covariance $\Sigma$. The analysis identifies a phase transition between benign and harmful overfitting,
$
n \|\mu\|^4 \gg \|\Sigma\| \,Tr(\Sigma).
$
The results apply to data distributions that may be correlated and heavy-tailed (need finite fourth moments), they don't require the common assumptions of isotropy or sub-Gaussian tails.

**Strengths:**

The two key technical contributions of this paper seem to me to be as follows.
- convergence in direction for gradient descent, not just gradient flow. Although this might feel intuitive this is not I would imagine easy to prove,  you need to show some uniform control on the angle between successive gradients which is complicated by the fact that the gradient angle can jump around discontinuously due to the non-smoothness of leaky-ReLU.
- identifies benign versus harmful overfitting regimes without isotropic / nearly orthogonal data: in many former works this was a requirement for proving benign overfitting as it allows one to treat network activations of different points almost independently, thereby simplifying the dynamics greatly. It is worth noting that another work tackles this issue in a slightly different setting (hinge loss rather than exponential tailed), see questions below.
- the paper extends benign-overfitting and directional-convergence results to heavy-tailed mixtures.

**Weaknesses:**

The main weaknesses of the paper I think are as follows
- Scope / relevance of the setup, namely shallow leaky-relu network, only training the inner layer weights, data which is linearly separable / drawn from a relatively simple distribution. Analysis even in this setting is pretty challenging though so I do not think this is a major issue.
- A flip side of one of the paper's strengths is that it does not provide finite-sample or high-probability generalization guarantees. The absence of sub-Gaussian or independence assumptions I think precludes standard concentration arguments, so the results provided are weaker in the sense that they hold asymptotically and in expectation. In short, compared to prior results, while the theory applies to broader data distributions, it does not yield as explicit rates or probability bounds. This really is more of a trade-off though than a weakness.
- I think some relevant work has been missed in the literature review, see questions below.

**Questions:**

- On convergence in direction: for what range of step sizes do we get convergence versus oscillation / divergence? At the moment Theorem 4.8 seems to say for sufficiently small step-size we get convergence, I am wondering if you have any insight into when this convergence breaks down?

- Can you sketch the convergence proof idea out perhaps? I am imagining that you try to fix the activation pattern eventually, allowing you to treat the network as a piecewise linear model? How do you control switching between activations regions and the contraction in angle?

- Have you thought about the non linearly separable setting? In this setting I think perhaps a finite solution would exist and as a result the analysis of convergence in direction would no longer be relevant but one could still think about a soft margin bias say.

- I think this work has some similarities (particularly in regard to relaxing the orthogonality conditions on the data) with "Benign overfitting in leaky relu networks with moderate input dimension", by Karhadkar et al, a link to which is given below,
https://proceedings.neurips.cc/paper_files/paper/2024/hash/4054556fcaa934b0bf76da52cf4f92cb-Abstract-Conference.html. Would you possibly comment? I think this and maybe some of the references therein might be worth considering adding to your prior work section.

**Details Of Ethics Concerns:**

Not applicable.

---

> ### Author Response · Authors · 2025-11-22
>
> We sincerely thank the reviewer for the time and effort devoted to evaluating our work. We appreciate the thoughtful comments, and we are grateful for the opportunity to clarify several points regarding our setting and contribution.
>
> Before addressing the specific questions, we would like to clarify a few points where our presentation may have caused confusion. All line numbers below refer to the original submission.
>
> **Clarifications:**
>
> *"The analysis identifies a phase transition between benign and harmful overfitting, $n\|\mu\|^4 \gg |\Sigma|, \mathrm{Tr}(\Sigma)$."*
>
> - In our work, the “phase transition” refers to the shift between the weak-signal and strong-signal regimes that appears explicitly in the test-error bounds of Theorem 5.1 (lines 322–325). The key amount in the bound $\frac{n\|\mu\|^4}{n\|\mu\|^2 + \mathrm{Tr}(\Sigma)}$ exhibits a transition depending on whether $n\|\mu\|^2$ or $\mathrm{Tr}(\Sigma)$ dominates.
>
> - The distinction between benign and harmful overfitting is discussed after Theorem 6.2 (lines 376–377).
>
> *"The results apply to data distributions that may be correlated and heavy-tailed (need finite fourth moments)."*
>
> - We would like to emphasize that our assumptions are even weaker: we only assume finite $r$th moments where $r \in (2, 4]$; see the definition of model (PM) in Section 3 (line 180-181).
>
> *"A flip side of one of the paper's strengths is that it does not provide finite-sample or high-probability generalization guarantees.*
>
> - Our high-probability guarantees are provided in Theorem 6.2 for model (sM) and Theorem 6.3 for model (PM). Sections 4 and 5 present the core results in deterministic form, and as noted at the end of each section (lines 277–288 and 329–331), Section 6 shows that all the assumptions hold with high probability. We would like to highlight that this separation between deterministic and probabilistic arguments is what allows us to relax distributional assumptions far beyond prior work.
>
> **Responses to questions**
>
> *1. On convergence in direction: for what range of step sizes do we get convergence versus oscillation / divergence? ... I am wondering if you have any insight into when this convergence breaks down?*
>
> - Thank you for raising this. Our focus in this paper is to understand how implicit bias interacts with generalization rather than to derive the most general possible convergence conditions. Theorem 4.8 therefore presents sufficient–but not fully necessary–conditions. We agree that characterizing the exact boundary between convergence and oscillation/divergence is an interesting direction.
>
> *2. Can you sketch the convergence proof idea out perhaps? I am imagining that you try to fix the activation pattern eventually, allowing you to treat the network as a piecewise linear model? ...*
>
> - A proof sketch is provided in Section 7. The key idea is to fix the activation pattern. The event $E(\theta_1, \theta_2)$ (equation (4)) is introduced precisely to control the gradient descent trajectory. The assumptions on $\theta_1, \theta_2$ combined with our step-size and initialization assumptions allow us to fix the activation pattern and treat the network as effectively linear.
>
> *3. Have you thought about the non linearly separable setting? ...*
>
> - In the current work we reduce implicit bias in leaky ReLU networks to that in linear classification under linear separability. Extending the results to nonseparable data is an appealing direction. [Zi & Telgarsky (2019)](https://proceedings.mlr.press/v99/ji19a/ji19a.pdf) study implicit bias for nonseparable data in linear models, and it would indeed be interesting to explore analogous behavior for ReLU networks.
>
> *4. I think this work has some similarities (...) with "Benign overfitting in leaky relu networks with moderate input dimension", by Karhadkar et al. Would you possibly comment? ...*
>
> - Thanks for your pointer. We added notes in prior work section and after Theorem 6.2. Let us comment below on similarities and differences.
>
> - **Implicit bias:** Their work uses hinge loss and therefore does not observe directional convergence, making detailed comparison difficult.
>
> - **Distributional assumptions:** Their setting can be viewed as a special case of ours, except that they include label-flipping noise. In our notation, their model corresponds to a Gaussian mixture with $\|\boldsymbol{\mu}\| \approx 1$, $\mathrm{Tr}(\Sigma) \approx 1$, and $\boldsymbol{\mu} \perp \boldsymbol{z}$, which places them in the strong-signal regime $n\|\boldsymbol{\mu}\|^2 \gg \mathrm{Tr}(\Sigma)$. Prior work (e.g., [Hashimoto et al. 2025](https://www.arxiv.org/abs/2501.10538)) shows that the behavior with label-flipping noise is quite different from the noiseless setting, which explains why the test-error bounds are not directly comparable. Moreover, since their work keeps $\|\boldsymbol{\mu}\|$ and $\mathrm{Tr}(\Sigma)$ fixed, the phase transition between weak and strong signal regimes is not observed.

---

> ### Comment · Reviewer_RXkj · 2025-11-25
>
> Thanks for your feedback.
>
> On the benign overfitting contributions  I am now confused and probably missed this in the first pass. When most people have done analysis on benign overfitting before they explicitly consider label noise, i.e., flipping a number of the labels of points so that some fraction of the training sample is mislabeled. I implicitly assumed this was the setting you were studying but after double checking I can't see where you incorporate label noise or more generally mislabelling into your model / setup? If wrongly labelled data isn't incorporated then I don't think you are really providing results on benign overfitting in the mainstream sense. In particular, I would loosely define benign overfitting as "the predictor interpolates noisy / incorrect labels yet the predictor remains near Bayes-optimal" and I am not aware of any theory work which studies benign overfitting without label noise (i.e., only feature noise). Currently then, although I think the directional convergence of GD is a nice contribution I am worried that the benign overfitting contribution has been overstated and the comparison with prior work needs to be much better clarified and have such have had to lower my score. Happy to discuss further.

---

> > ### Author Response · Authors · 2025-11-25
> >
> > Thanks for your additional comment. We are happy to make further clarifications.
> >
> > We would like to emphasize that  1) benign overfitting has been studied for the case without label-flipping noise, and 2) the training data is still noisy even without label-flipping noise.
> >
> > 1. Prior work
> >
> > - While some works emphasized label-flipping noise, a parallel line of research has studied benign overfitting in settings without label-flipping noise. This includes some of the earliest theoretical results in the area.
> >
> > - For example, [Cao et al (2021)](https://proceedings.neurips.cc/paper/2021/file/46e0eae7d5217c79c3ef6b4c212b8c6f-Paper.pdf) and [Minsker et al. (2021)](https://arxiv.org/abs/2111.07041) studied binary linear classification without label-flipping noise.
> >
> > - [Wang and Thrampoulidis (2022)](https://epubs.siam.org/doi/10.1137/21M1415121) studied benign overfitting in binary linear classification for both with/without label-flipping noise (Section 2-6 for noise-free label model and Section 7 for noisy label model). More recently, [Tsigler et al. (2025)](https://arxiv.org/abs/2503.07966) studied benign overfitting of the minimum norm interpolating solution for both with/without label-flipping noise, and showed that their geometric pictures are quite different (Section 2 for the geometric argument and Section 4 for benign overfitting). Lastly, [Hashimoto et al. (2025)](https://www.arxiv.org/abs/2501.10538) showed that generalization behavior differs in the strong signal regime depending on whether label-flipping noise is present, while the behavior is similar in the weak-signal (nearly-orthogonal) regime (Section 3).
> >
> > - Prior work which include studies of benign overfitting without label-flipping noise is not limited to above, e.g. Chatterji and Long (2021) studied benign overfitting in binary linear classification both with and without label-flipping noise, Frei et al. (2022) and Xu and Gu (2023) studied benign overfitting in ReLU type two-layer neural networks both with and without label-flipping noise.
> >
> > 2) Noise in the training data:
> >
> > - The mixture model considered here—as well as in previous works above—naturally includes a source of noise, represented by the random vector $\boldsymbol{z}$. The training data $(\boldsymbol{x}_i, y_i)$ is still noisy in a sense $\boldsymbol{x}_i$ may be located near $-\boldsymbol{\mu}$ while $y_1 = 1$ (recall we defined $\boldsymbol{x}_i = y_i \boldsymbol{\mu} + \boldsymbol{z}_i$).
> >
> > - For intuition, consider the isotropic mixture $x = y\boldsymbol{\mu} + \boldsymbol{z}$ with $\boldsymbol{z}\sim \mathcal{N}(0, I_p)$. Since $\|\boldsymbol{z}\|$ is typically of order $\sqrt{p}$, the two class-conditional distributions overlap significantly unless $\|\boldsymbol{\mu}\|$ is substantially larger than $\sqrt{p}$. Thus, even without label-flipping noise, the training data is still noisy and overfitting phenomena remain non-trivial.
> >
> > Actually, extending our work to noisy label setting is not difficult for the weak signal regime. However, as noted in Hashimoto et al. (2025), incorporating label noise in the strong-signal regime would lead to qualitatively different phenomena and requires substantially more technical work. For this reason, we chose to limit our results to noise-free label case for better presentation, and the noisy-label setting including the strong signal regime is left for future work.

---

> > > ### Comment · Reviewer_RXkj · 2025-11-25
> > >
> > > Thanks for your response.
> > >
> > > 1) Sure prior works study both the noisy and clean label cases, but I don't think any of these prior works state that they are studying benign overfitting when there is no label noise... Feel free to point me at a specific section in a paper / theorem or lemma contrary to that but take the paper by Wang et al, section 7 is on benign overfitting and here they specifically introduce label noise.
> > >
> > > 2) I understand that feature noise is also a source of complication but I struggle to see how feature noise and label noise are equivalent in this setting. In particular, label noise will result in contributions to the aggregate gradient which point in the wrong direction in the subspace spanned by $\mu$, moreover one will typically have points which can only be fitted at the expense of the others which in turn makes the activation patterns less stable. It isn't clear to me why you have the same issues with feature noise alone.
> > >
> > > 3) What exactly do you mean by the weak signal regime? Can you explain why you think it would be easy / trivial to extend your results to this setting? I would expect it to complicate your convergence results.

---

> > > > ### Author Response · Authors · 2025-11-25
> > > >
> > > > Thanks for your additional questions. We are happy to provide further clarifications. We split our clarification into two comments due to the character limit.
> > > >
> > > > 1. Prior work
> > > >
> > > > - We would like to emphasize the following papers explicitly states that they study benign overfitting in no label-flipping noise models.
> > > >
> > > > - Cao et al (2021): As noted in our previous comment, this work only considers no label-flipping noise model. In the abstract they states *'In this paper, we study this “benign overfitting” phenomenon of the maximum margin classifier for linear classification problems.'* Moreover, they state in the beginning of Section 3 *'We also showcase the application of our results to isotropic and anisotropic sub-Gaussian mixture models to study the conditions under which benign overfitting occurs.'* Their benign overfitting result is given by Corollary 3.5 as an application of Theorem 3.1 (their main theorem). The term benign overfitting is introduced as *'capable of fitting noisy training data sets, while at the same time achieving small test errors'* in the beginning of the introduction.
> > > >
> > > > - Minsker et al. (2021): As noted in our previous comment, this work only considers no label-flipping noise model. In the abstract they states *'We conclude that interpolation is not only benign but can also be optimal, and in some cases robust.'* The term benign overfitting is introduced as *"estimators that interpolate the training data also yield good generalization error bounds when the number of covariates exceeds the sample size"* in the beginning of the introduction.
> > > >
> > > > - Wang and Thrampoulidis (2022):  As noted in our previous comment, this work studies no label-flipping noise model in Section 2 - 6. Benign overfitting result is discussed in Section 5, specifically in Corollary 5.1, Remark1, and the associated figure 3, whose title is *'Numerical demonstration of benign overfitting for the GMM'*. The term benign overfitting is introduced as *'the risk asymptotically approaches the optimal Bayes error despite perfectly fitting to noisy data'* at the end of the first paragraph of section 1.1.
> > > >
> > > > - To the best of our knowledge, the term benign overfitting first appeared in [Bartlett et al. (2020)](https://proceedings.neurips.cc/paper_files/paper/2022/file/a12c999be280372b157294e72a4bbc8b-Supplemental-Conference.pdf), in which they study benign overfitting in linear regression. In its introduction, the term is introduced as *'The phenomenon of benign overfitting is one of the key mysteries uncovered by deep learning methodology: deep neural networks seem to predict well, even with a perfect fit to noisy training data. Motivated by this phenomenon, we consider when a perfect fit to training data in linear regression is compatible with accurate prediction.'* It does not specify how noise should be incorporated in the training data. For binary classification, it has been used for both with/without label-flipping noise models as noted earlier.
> > > >
> > > > 2. Intuition behind feature noise
> > > >
> > > > - First, we would like to clarify that we do not consider feature noise and label noise are equivalent. They are distinct sources of noise that can be introduced in the training data $(\boldsymbol{x}_i, y_i)$. Recalling that our data is generated by $\boldsymbol{x} = y\boldsymbol{\mu} + \boldsymbol{z}$, random vector $\boldsymbol{z}$ introduce noise to feature alone, whereas label-flipping noise add noise to both the feature and label. For both cases, the training data $(\boldsymbol{x}_i, y_i)$ is noisy and the benign overfitting remains non-trivial.
> > > >
> > > > - For the intuition, consider isotropic gaussian case $\boldsymbol{x}_i = y_i\boldsymbol{\mu} + \boldsymbol{z}_i$, where $\boldsymbol{z}_i \sim \mathcal{N}(0, I_p)$. The two class conditional distributions (one centered at $\boldsymbol{\mu}$ and the other at $-\boldsymbol{\mu}$) would have significant overlap unless the signal $\boldsymbol{\mu}$ is significantly stronger than the strength of the noise $\boldsymbol{z}_i$, typically $\approx \sqrt{p}$.  In other words, you may not be able to identify whether $\boldsymbol{x}_i$ belongs to the class $y = 1$ or $y = -1$. Such situation would indeed complicate the activation pattern during training via gradient descent and also affect the generalization performance.

---

> > > > > ### Author Response · Authors · 2025-11-25
> > > > >
> > > > > 3. Extension to noisy label models under the weak signal regime
> > > > >
> > > > > - For model (sG) and (PM), we call the model is in the weak signal regime when $n\|\boldsymbol{\mu}\|^2 \lesssim \mathrm{Tr}(\Sigma)$. The more general definition is given shortly after Theorem 5.1 (line 332-333 in the revision). In the case of the isotropic case, this is when $n\|\boldsymbol{\mu}\|^2 \lesssim p$.
> > > > >
> > > > > - We consider that extending our results to the weak signal case is not difficult (but not necessarily trivial) for two reasons:
> > > > >
> > > > > 1) Analyzing the generalization performance is much simpler than the strong signal case since the gram matrix $\tilde X\tilde X^\top$ (see eqn 13) can be approximated by a diagonal matrix as noted at the end of the proof sketch (line 489-490 in the revision). Thus, the behavior of the gram inverse $(\tilde X\tilde X^\top)^{-1}$ can be analyzed easily even with the presence of label-flipping noise. The situation is significantly different in the strong signal regime. The gram matrix $\tilde X\tilde X^\top$ can no longer be approximated by a diagonal matrix and hence understanding the behavior of the gram inverse requires substantially more complex technical development.
> > > > >
> > > > > 2) It requires more involved arguments to show directional convergence. However, this can be dealt with concentration of the number of labels flipped (approximately $\eta n$ labels get flipped for sufficiently large $n$). With the concentration, we can control gradient descent updates in a similar manner as in our work.
> > > > >
> > > > > Hope the above responses clarifies your questions. Please let us know if you have further questions/comments.

---

> > > > > > ### Comment · Reviewer_RXkj · 2025-11-26
> > > > > >
> > > > > > 3. Ok thanks, looking at 13 I would expect that as you introduce label noise the sets J_{-} and J_{+} could change though? In particular, under label noise I think these sets could change in time (I think these are the neuron indices with a negative versus positive output weight? Correct me if I am wrong). Switches of sign provably occur for instance in https://arxiv.org/pdf/2306.09955 (albeit in the case of hinge loss, I would be surprised if something similar didn't happen here as well though). To be clear, I am not saying the current approach couldn't be adapted, only that as things stand it doesn't seem trivial to extend to (at least a non vanishing fraction in some limit) amount of label noise?

---

> > > > > > > ### Author Response · Authors · 2025-11-26
> > > > > > >
> > > > > > > Thanks for your additional question. We are happy to elaborate on why our approach still works with label-flipping noise under the weak signal regime.
> > > > > > >
> > > > > > > - It is not $J_+$ or $J_-$ which is impacted by introducing label noise since $J_+$ and $J_-$ are the set of indices for the second layer weights $a_j$ (recall that $a_j$ are set at $\pm 1/\sqrt{m}$). It is the matrix $B_j$ in equation (13) that is going to be contaminated by label noise as $B_j$ is defined by $(B_j)_{kk} = \zeta(a_jy_k)$ (defined right before eqn (13)). Here $\zeta$ is a function defined by $\zeta(x) = 1$ if $x \geq 0$ and $=\gamma$ if $x<0$.
> > > > > > >
> > > > > > > - Nevertheless, the gram inverse $(\tilde X \tilde X^\top)^{-1}$ can be analyzed in a similar manner under the weak signal regime since $XX^\top$ is not affected by label noise. As noted in the last paragraph of Section 7 (after eqn 14), $XX^\top$ can be approximated by $RI_n$ under the weak signal regime, and hence $\tilde X\tilde X^\top \approx R(q_+B_+^2 + q_- B_-^2)$, where $q_+ = |J_+|/m$ and $q_- = |J_-|/m$, respectively. Thus, $\tilde X \tilde X^\top$ is still approximated by a diagonal matrix even with label noise, so the gram inverse can be approximated by the inverse of the diagonal matrix.
> > > > > > >
> > > > > > > - We would like to add that this is not the only change that would be caused by label noise. As you can see from the explicit form of the convergent direction $\hat w = \tilde X^\top (\tilde X \tilde X^\top)^{-1}\boldsymbol{y}$ (line 471 in the revision), not only $\tilde X$ but also $\boldsymbol{y}$ is going to be contaminated by label noise. Thus, we need to make some minor adjustments to our argument for a test error bound. However, the adjustments required here can be mostly handled by a standard concentration argument (approximately $\eta n$ labels get flipped for sufficiently large $n$), and not more complicated than analyzing the gram inverse.
> > > > > > >
> > > > > > > - Lastly we would like to emphasize once again that the situation is very different for the strong signal regime. Even without label flip, the analysis of the gram inverse required highly technically involved arguments, which led to one of our main contributions, i.e. establishing a test error bound beyond the weak signal regime (=nearly orthogonal data regime) which shows a phase transition between regimes. While our argument for directional convergence can be adapted for the strong signal regime with minor changes, we believe that establishing a test error bound would require far more involved argument as was the case for binary linear classification.

---

### Official Review · Reviewer_SjjM · 2025-11-01

**Soundness:** 4
**Presentation:** 4
**Contribution:** 3
**Rating:** 6
**Confidence:** 4

**Summary:**

This work studies 2 layer (1-hidden layer) leaky relu nets. The authors give result on the limiting behavior of gradient descent and generalization bound.

**Strengths:**

Main result on convergence of the parameters also gives the geometry of the decision boundary (which is linear).

Goes beyond the nearly orthogonal setting that many previous work assumes.

Is well-written.

**Weaknesses:**

The term "benign overfitting" is used throughout and in theorem 6.3, but I could not find the mathematical definition of the term.

Is interpolation implied by result on Line 258?

Also would be helpful to explicitly to write down the Bayes error rate (perhaps it's obvious and I just missed it, but I can't figure it out while reading).

**Questions:**

How is w_bar on Line 268 related to the mu from eqn (3)?

Line 257: does a neuron being "activated" means the same thing for leaky-relu as it does for the standard relu? the term is a bit confusing since leaky-relu doesn't have a "dead spot".

Line 259 says "the convergent direction [...] can also be given by ...". Is the convergent direction unique, or is the direction defined in eqn (5) just one possible convergent direction?

---

> ### Author Response · Authors · 2025-11-22
>
> We sincerely thank the reviewer for the time and effort devoted to evaluating our work. We appreciate the thoughtful comments, and we are grateful for the opportunity to clarify several points regarding our setting and contributions. We note that all line numbers below are from original submission.
>
> **Responses to questions:**
>
> *1. The term "benign overfitting" is used throughout and in theorem 6.3, but I could not find the mathematical definition of the term.*
>
> - We say benign overfitting holds if the training data is classified perfectly while the test error on a new observation is arbitrarily close to zero. This is added at the end of Section 3.
>
> *2. Is interpolation implied by result on Line 258?*
>
> - Yes, as you pointed out line 258 implies the training loss $\mathcal{L}(W^{(t)})$ is driven to zero at the order of $t^{-1}$, thus implies interpolation. We have added a brief clarification about this point after Theorem 6.2. when we discuss sufficient conditions of benign overfitting under model (sG).
>
> *3. Also would be helpful to explicitly to write down the Bayes error rate (perhaps it's obvious and I just missed it, but I can't figure it out while reading).*
>
> - The Bayes error rate depends on the distribution of a random vector $\boldsymbol{z}$ in model (M), and there is no closed form in general. We can obtain an exact amount if $\boldsymbol{z}$ is Gaussian $\sim \mathcal{N}(0, \Sigma)$ and $\Sigma$ is invertible, which is given by $\kappa(\|\Sigma^{-1/2} \boldsymbol{\mu}\|)$. Here $\kappa (t) = \mathbb{P}(\xi \geq t)$, where $\xi \sim \mathcal{N}(0, 1)$. Comparing to our error bounds in Theorem 6.2 for Gaussian mixture, our bounds in the case of isotropic Gaussian mixture becomes $\kappa\left[c^{-1}\left(\frac{n\|\boldsymbol{\mu}\|^4}{n\|\boldsymbol{\mu}\|^2 + \mathrm{Tr}(\Sigma)}\right)^\frac{1}{2}\right]$ and $\kappa\left[c\left(\frac{n\|\boldsymbol{\mu}\|^4}{n\|\boldsymbol{\mu}\|^2 + \mathrm{Tr}(\Sigma)}\right)^\frac{1}{2}\right]$ while the Bayes error rate becomes $\kappa(\|\boldsymbol{\mu}\|)$. Therefore, we see that our bounds are tight in the strong signal regime, i.e. when $n\|\boldsymbol{\mu}\|^2 \gg p$, up to a constant factor in $\kappa$. This indicates that the implicit bias of gradient descent leads to the direction which achieves the Bayes optimal rate in the strong signal regime while not in the weak signal regime. We have added this observation briefly after Theorem 6.2.
>
>
> *4. How is $\bar w$ on line 268 related to the mu from eqn (3)?*
>
> - $\boldsymbol{\bar w}$ is a function of $\boldsymbol{x}_i$, and $\boldsymbol{x}_i$ are a function of $y_i$, $\boldsymbol{\mu}$, and $\boldsymbol{z}_i$ as specified in eqn(3). Therefore, $\boldsymbol{\bar w}$ depends on $\boldsymbol{\mu}$. A closed form expression of $\boldsymbol{\bar w}$ can be obtained under over-parametrization as noted in Section 7 (line 448).
>
> *5. Line 257: does a neuron being "activated" means the same thing for leaky-relu as it does for the standard relu? the term is a bit confusing since leaky-relu doesn't have a "dead spot".*
>
> - We say that $j$-th neuron is activated if $a_jy_i\phi(\langle \boldsymbol{x}_i, \boldsymbol{w}_j\rangle) >0$ for all $i\in I$. We have added this explanation in the beginning of section 4, previously it appeared later in the beginning of section 7.
>
> *6. Line 259 says "the convergent direction [...] can also be given by ...". Is the convergent direction unique, or is the direction defined in eqn (5) just one possible convergent direction?*
>
> - Yes, it is unique since the optimization problem eqn(5) is strictly convex optimization problem. We added the clarification after Theorem 4.8.

---

### Official Review · Reviewer_4YXg · 2025-11-02

**Soundness:** 4
**Presentation:** 3
**Contribution:** 3
**Rating:** 8
**Confidence:** 2

**Summary:**

The authors consider a natural simplified model of a two layer network with leaky ReLU activations.  The first-layer weights are trainable, and the second layer weights are simply fixed and set to $\pm 1$ up to rescaling.  The authors use an exponential loss function. This is all a fairly standard model for this type of analysis. The goal is to solve a classification problem, where the two classes are centered at $\pm \mu$ and iid noise either with subgaussian or polynomial tails is added to each data point.

The first main contribution of the paper is to work out the convergent direction of the network weights.  Notably, this convergent direction is worked out explicitly under deterministic conditions, and is presented as the solution to a simple optimization problem, which has appeared in prior work under stronger conditions.  This allows the paper to separate the deterministic and probabilistic components of the analysis.  Then in Theorem 5.1 they turn to understanding the classification error under a random data-generation model.  Interestingly, this work gives both an upper and a lower bound on the probability of misclassification, so the authors are able to observe a phase transition between the weak versus strong signal regime, with the latter being associated with higher levels of misclassification.  The authors apply their result to random data models (with subgaussian and polynomial tails) in the overparameterized regime and thereby give new results on the benign overfitting phenomenon.  The main improvements are that the data need not be nearly orthogonal, and that they can handle leaky ReLU activations.

**Strengths:**

The paper seems technically sound and gives legitimate improvements over prior work.

The first improvement is that the analysis holds beyond the nearly orthogonal setting, which is necessary for understanding the types of data that occur in practice, particularly when the means of the two classes are far apart.  They also understand the convergent direction of the weights in a data-dependent way, which may make these results easier to apply as a black box in other settings.  In addition to obtaining an upper bound on the probability of misclassification, they also obtain a lower bound.  This allows for a precise characterization of when overfitting occurs, which appears to be an improvement over prior work.

The observation of a phase transition between the weak and strong signal regime is also interesting, and gives an explanation as to way previous work required an orthogonality assumption.  This seems to be a substantially novel result compared to prior work.

**Weaknesses:**

Some of the actual results are perhaps a bit incrememental.  For example the convergent direction is precisely that of Frei et al. as should be expected. On the other hand, relaxing the orthogonality assumption seems like a nice direction, as previous work in the area was unable to do this.

The actual model is fairly weak compared to modern DL models, and so may be limited in terms of understanding why benign overfitting occurs in practice.  Benign overfitting in two-layer networks is an established research direction though, and think this would be broadly interesting to that community.

**Questions:**

What is the difficulty in move from gradient flow results to gradient descent?  Can gradient flow results not be converted to gradient descent results just by taking the step size small enough? Or is the interest in getting specific bounds on the step size?

“We confirm in the next section that these assumptions are satisfied with high probability under model (sG) and (PM), and conjecture that they can also be verified in other settings.”
What settings are conjectured to work?

The two-sided bounds e.g. in Theorem 6.2 are given for Gaussian z.  Are these bounds expected to hold more generally?

— Minor —

Assumption 6.1 (a) – latex compilation issue
Beginning of section 3.  $|a_j| = \pm \frac{1}{\sqrt{m}}$ – shouldn’t have the absolute values

---

> ### Author Response · Authors · 2025-11-22
>
> We sincerely thank the reviewer for the time and effort devoted to evaluating our work. We appreciate the thoughtful comments, and we are grateful for the opportunity to clarify several points regarding our contributions. We note that all line numbers below are from original submission.
>
> Before addressing the specific questions, we would like to comment on the following point:
>
> *"The observation of a phase transition between the weak and strong signal regime is also interesting, and gives an explanation as to way previous work required an orthogonality assumption. This seems to be a substantially novel result compared to prior work."*
>
> - We thank the reviewer for recognizing one of our contributions. We would like to emphasize that establishing our classification error bound in strong signal regime in fact requires more technically complex development than nearly orthogonal data regime as noted in proof sketch (line 468-469).
>
>
> Let us answer your questions in the following.
>
>
> **Responses to questions:**
>
> *Q1. What is the difficulty in move from gradient flow results to gradient descent? Can gradient flow results not be converted to gradient descent results just by taking the step size small enough? Or is the interest in getting specific bounds on the step size?*
>
> - We would like to first note that convergence of gradient flow does not in general imply convergence in gradient descent. Even for a small step size, gradient descent could diverge while gradient flow converge if the objective function is non-convex. Therefore, proving convergence of gradient descent requires more than simple discretization. Besides, the type of convergence in this work is directional convergence. To the best of our knowledge, there is no standard framework established for extending directional convergence from gradient flow to gradient descent. In fact, establishing directional convergence of gradient descent for neural networks has been an open problem since Ji and Telgarsky (2020) established it for gradient flow. There was not even a partial result for a while until Cai et al. (2025) showed under smoothness assumption (they require twice differentiability of the network in parameter space). It seems to us a novel approach is required to remove the smoothness assumption so that directional convergence of gradient descent can be established rigorously for ReLU type deep neural networks.
>
> *Q2. “We confirm in the next section that these assumptions are satisfied with high probability under model (sG) and (PM), and conjecture that they can also be verified in other settings.” What settings are conjectured to work?*
>
> - Both model (sG) and (PM) assumes that entries of $\boldsymbol{\xi}$ are independent (line 177-181), which means $\boldsymbol{z}$ has independent entries up to linear transformation. We think that this assumption may be weakened. For example, Hashimoto et al. (2025) considers a more involved model for binary linear classification. In their model, $\boldsymbol{z}$ has extra random scaling factor, i.e. $\boldsymbol{z} = g\Sigma^{1/2}\boldsymbol{\xi}$, where $g\in (0, \infty)$ is a positive random variable with certain moment conditions. With the random scaling $g$, $\boldsymbol{z}$ does no longer have independent entries after any linear transformation. However, their proof technique is quite different from ours and does not seem to work for our setting.
>
>
> *Q3. The two-sided bounds e.g. in Theorem 6.2 are given for Gaussian z. Are these bounds expected to hold more generally?*
>
> - Yes. Actually, we chose to present this result for Gaussian $\boldsymbol{z}$ since this is when we can give very explicit bounds. We can generalize this part in the following way:
>
> - If there exist decreasing functions $\phi_i : \mathbb{R} \to \mathbb{R}, i = 1, 2$ such that
> $$
> \phi_1(t) \leq \mathbb{P}(\langle \boldsymbol{z}, \boldsymbol{v} \rangle \leq -t) \leq \phi_2(t)
> $$
> for any $\boldsymbol{v}$ with $\|\boldsymbol{v}\| = 1$, then we have
> $$
> \phi_1 \left[ c \beta_{min}^{-\frac{1}{2}} \left(\frac{n \|\boldsymbol{\mu} \|^4}{n \|\boldsymbol{\mu} \|^2 + Tr (\Sigma)} \right)^\frac{1}{2}  \right] \leq P_{(\boldsymbol{x}, y)}(yf(\boldsymbol{x}; \hat W)<0) \leq \phi_2\left[c^{-1}\beta_{max}^{-\frac{1}{2}}\left(\frac{n\|\boldsymbol{\mu}\|^4}{n\|\boldsymbol{\mu}\|^2 + \rm{Tr}(\Sigma)}\right)^\frac{1}{2} \right].
> $$
> This is possible since our bounds on the key amount $\frac{\langle \boldsymbol{\hat w}, \boldsymbol{\mu} \rangle}{\|\boldsymbol{\hat w}\|}$ established in Lemma C.10 and C.11 are obtained without assuming Gaussianity. In the Gaussian case functions $\phi_i(t)$ are given by $\kappa(t) = \mathbb{P}(\xi_1 \geq t)$, which is exactly the conclusion of Theorem 6.2.
>
>
> *4. Assumption 6.1 (a) – latex compilation issue. Beginning of section 3. – shouldn’t have the absolute values.*
>
> - Thanks for pointing out the latex compilation issue and typo. We have corrected them.

---

### Author Response · Authors · 2025-12-03

We thank all the reviewers for the time and effort devoted to provide us with constructive feedbacks. The feedbacks in fact led us to the revision of our paper, which now highlights our main contributions with better clarity and readability. It was very unfortunate that we were unable to hear back from all of the reviewers before the discussion was discontinued due to the identity leakage incident.

Nevertheless, we believe that almost all comments/questions have been addressed successfully by our rebuttal which was made about a week before the discussion was discontinued: 1) most of them were constructive clarification questions which we have incorporated in the revision, and 2) only few concerns by reviewers are actually based on the factual misunderstanding which we believe should be clear from our rebuttal. We appreciate the reviewer RXkj for engaging in discussion right after our posting rebuttal so that we were able to answer all the questions/comments by RXkj before the closure of the discussion.

Lastly, we would like to give a high level summary of our contributions to help support new area chair.

**Summary of our main contributions**
- Established sufficient conditions of benign overfitting in leaky ReLU two-layer neural networks beyond nearly-orthogonal data regime in previous work, which resulted in the discovery of a newly identified phase transition (weak signal regime vs. strong signal regime).
- The analysis is done through first establishing directional convergence of gradient descent via exponential loss (Theorem 4.8) and studying the classification error bound of the convergent direction (Theorem 5.1). Previously directional convergence of ReLU type networks was only established for gradient flow. We also note that sufficient conditions are given in a deterministic manner. In this way, we separated deterministic and probabilistic components of the analysis.
- The separation led us to relax distributional assumptions significantly from previous work, i.e. we show that the deterministic conditions are met with high probability not only under sub-Gaussian mixtures (Theorem 6.2), but also under polynomially tailed mixtures (Theorem 6.3). For polynomially tailed mixtures, we only assume a very mild moment assumption.
- We further established a lower bound in the case of Gaussian mixtures, which shows our bound is tight and validates the phase transition in the classification error. The lower bound also tells us when benign overfitting fails (harmful overfitting occurs).

**Summary of our minor contributions**
- Our analysis do not require the width of networks to grow to infinity. In fact, the width $m$ is fixed in all of our analysis, which was not the case in previous work.
- Even in nearly orthogonal data regime (= the weak signal regime), we identified a new regime (Assumption 6.1 (a) (i)) which was not studied in previous work.

We would like to thank again all the reviewers for their dedication to make constructive feedbacks.

---

### Meta-Review · Area_Chair_bRF2 · 2026-01-02

**Summary:**

The majority of the reviewers were strongly supportive of this paper: they appreciated the technical achievements in understanding implicit regularization and benign overfitting of nonlinear neural nets trained by GD, under less restrictive assumptions than prior work.  There was a productive and significant back and forth between the authors and reviewers, and I would recommend that the authors revise their manuscript to take these into account.  The largest concern appears to be the usage of the phrase "benign overfitting" for settings where there is no label noise; I agree that this is not ideal, but not significant enough to prevent recommendation for acceptance.  I think the clear delineation of the distinct behaviors under different SNR regimes in the noise-free setting is sufficient for acceptance, but strongly recommend the authors make more explicit the lack of label noise and the limitation this imposes on the work.

**Reviewer Concerns:**

4YXg and RXkj each had questions regarding why the GF proof does not immediately yield the GD proof, which I think were reasonably addressed by the authors (non-convexity means such reductions are not straightforward/guaranteed)

Multiple reviewers asked for clarifications on the definition of benign overfitting.  I agree with the reviewers that calling it benign overfitting is a little inappropriate, although the author is correct that there have been multiple prior published works where such a thing was done.

**Reviewer Scores:**

4YXg would have maintained their 8

SjjM likely would have increased their score to an 8, as all concerns were fairly well addressed

RXkj mentioned they would lower their score, I am not sure if they would have maintained at a 6 or re-revised up to 8

i5vn likely would have increased to a 6

---

### Decision · Program_Chairs · 2026-01-26

Accept (Poster)